# Sampling 3D Molecular Conformers with Diffusion Transformers

**J. Thorben Frank**[1,2*]   **Winfried Ripken**[1,2*]   **Gregor Lied**[1*]
**Klaus-Robert Müller**[1,2,3,4,5]   **Oliver T. Unke**[3]   **Stefan Chmiela**[1,2]

[1]Technical University Berlin   [2]BIFOLD Berlin   [3]Google DeepMind
[4]MPI for Informatics, Saarbrücken   [5]Department of Artificial Intelligence, Korea University

thorbenjan.frank@gmail.com, oliverunke@google.com, stefan@chmiela.com

## Abstract

Diffusion Transformers (DiTs) have demonstrated strong performance in generative modeling, particularly in image synthesis, making them a compelling choice for molecular conformer generation. However, applying DiTs to molecules introduces novel challenges, such as integrating discrete molecular graph information with continuous 3D geometry, handling Euclidean symmetries, and designing conditioning mechanisms that generalize across molecules of varying sizes and structures. We propose DiTMC, a framework that adapts DiTs to address these challenges through a modular architecture that separates the processing of 3D coordinates from conditioning on atomic connectivity. To this end, we introduce two complementary graph-based conditioning strategies that integrate seamlessly with the DiT architecture. These are combined with different attention mechanisms, including both standard non-equivariant and SO(3)-equivariant formulations, enabling flexible control over the trade-off between between accuracy and computational efficiency. Experiments on standard conformer generation benchmarks (GEOM-QM9, -DRUGS, -XL) demonstrate that DiTMC achieves state-of-the-art precision and physical validity. Our results highlight how architectural choices and symmetry priors affect sample quality and efficiency, suggesting promising directions for large-scale generative modeling of molecular structures. Code is available at https://github.com/ML4MolSim/dit_mc.

## 1   Introduction

The three-dimensional arrangement of atoms in a molecule, known as conformation, determines its biological activity and physical properties, making it fundamental to applications such as computational drug discovery and material design. Accurately predicting the most energetically favorable conformers (i.e., stable conformations) for large molecular systems is a highly non-trivial task. Traditional techniques, such as Molecular Dynamics and Markov Chain Monte Carlo, attempt to explore the conformational space by simulating physical movement or probabilistic sampling. However, these methods often require many simulation steps to move from one conformer to another, making them computationally expensive. Generative machine learning (ML) models offer a more targeted approach by allowing to directly sample from the space of promising conformations.

Recent years have seen significant progress, enabled by the development of specialized architectures for the generation of molecules [1–9] and materials [10–12]. This is in contrast to image and video synthesis, where the more generalized diffusion transformer (DiT) architecture [13] has

---

*Equal contribution.

39th Conference on Neural Information Processing Systems (NeurIPS 2025).

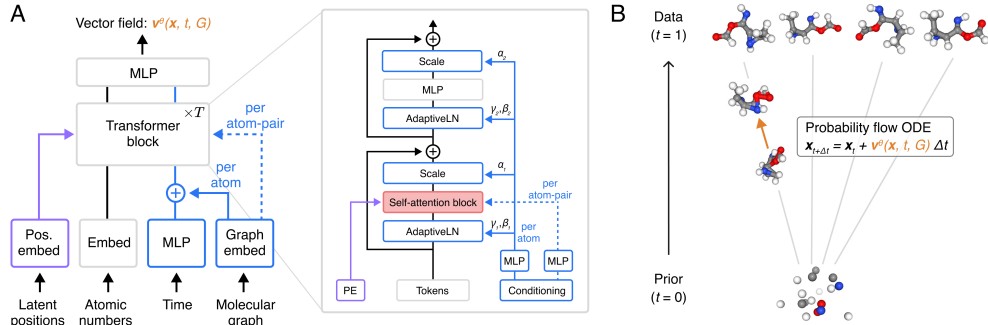

Figure 1: **(A)** Diffusion transformer for molecular conformer generation (DiTMC), with interchangeable self-attention blocks and positional embeddings (PEs); we evaluate various combinations as detailed in the main text. **(B)** DiTMC predicts a velocity per atom, used to model a probability flow ODE, which samples from the probability distribution $p(\boldsymbol{x}|\mathcal{G})$, where $\mathcal{G}$ is a molecular graph.

become a leading model, consistently delivering strong performance and efficiency across diverse applications [14–17]. Adapting DiTs, which were originally developed for grid-structured image data, to continuous, irregular molecular geometries poses unique challenges, which need to be addressed to unlock the potential of this powerful architecture for molecular conformer generation. Key design questions include how to encode molecular connectivity and incorporate Euclidean symmetries, such as translational and rotational invariance/equivariance.

In this work, we address these conceptual challenges and propose DiTMC, a new DiT-style architecture for molecular conformer generation. We introduce novel conditioning strategies based on molecular graphs, enabling the generation of 3D structures. Our modular architectural design allows us to systematically investigate the impact of different self-attention mechanisms within the DiT architecture on conformer generation quality and efficiency. We conduct a comparative study including standard (non-equivariant) self-attention with both absolute and relative positional embeddings and an explicitly $SO(3)$-equivariant variant. While exact equivariance can positively impact performance, it also incurs significant computational costs. We find that simpler attention mechanisms are highly scalable and still perform competitively. Multiple of the tested DiTMC variants achieve state-of-the-art precision on established conformer generation benchmarks. Moreover, the molecular structure ensembles generated by our models align more closely with physical reality, as evidenced by the high accuracy of physical properties extracted from them. To summarize, our work contains the following main contributions:

- We propose two complementary conditioning strategies based on trainable conditioning tokens for (pairs of) atoms extracted from molecular graphs, which are designed to align with the architectural principles of DiTs. We propose to condition our self-attention formulation on geodesic graph distances extracted from molecular graphs and demonstrate that it significantly increases performance of our model.

- We investigate the impact of different self-attention mechanisms, including standard (non-equivariant) and $SO(3)$-equivariant formulations, on model accuracy and performance. We find that including symmetries can improve the fidelity of generated samples at the price of increased computational cost during training and inference.

- Based on our insights, we present a simple, non-equivariant, yet expressive DiT architecture that achieves state-of-the-art precision and physical validity on established benchmarks. Its performance improves with model scaling, making it a promising candidate for large-scale molecular conformer generation.

## 2 Related Work

**Generative Modeling** Generative models create diverse, high-quality samples from an unknown data distribution. They are widely used in image generation [18, 19], text synthesis [20] and the natural sciences [21–23]. Most recent approaches learn a probabilistic path from a simple prior to the data

distribution, enabling both efficient sampling and likelihood estimation [24–28]. This is achieved by different modeling paradigms, including flow matching [26] or denoising diffusion [24].

**(Diffusion) Transformers** Originally introduced for natural language processing [29], transformer architectures have become state-of-the-art in computer vision [18], and recently also found widespread adoption in quantum chemistry, e.g., for protein prediction [30, 31], 3D molecular generation [32, 33] or molecular dynamics simulations [34–36]. In the context of generative modeling, diffusion transformers [13] (DiTs) have emerged as powerful tools incorporating conditioning tokens, e.g., to prompt image generation [13] or to design molecules and materials with desirable properties [37]. This work applies prototypical DiTs to molecular conformer generation.

**Molecular Conformer Generation** Molecular conformer generation aims to find atomic arrangements (Cartesian coordinates) consistent with a given molecular graph. Numerous ML approaches have been proposed for sampling conformers, all aiming to improve upon conventional methods.

While early approaches were based on RDKit [38] or variational auto encoders [39–41], more recent advances employ diffusion and flow-based models [42, 43], including E-NFs [9], CGCF [44], GeoDiff [45], TorsionDiff [46], MCF [47], ET-Flow [32], and DMT [48].

**De-Novo Molecular Generation** Instead of conditioning on a given molecular graph, several existing approaches tackle the problem of finding novel and stable molecules given only a set of atoms [8, 49, 50]. Bridging the gap between unconditional and conformer generation, universal models have been proposed that can solve a multitude of tasks. In particular, models within the Uni-Mol model family [51, 52], make use of graph geodesic distances. However, their design requires tracking high-dimensional pair-representations throughout the architecture, which makes it less efficient.

## 3 Preliminaries

### 3.1 Molecular Conformers

A molecule can be represented as a molecular graph $\mathcal{G} = (\mathcal{V}, \mathcal{E})$, where the nodes $\mathcal{V}$ correspond to atoms and the edges $\mathcal{E}$ represent chemical bonds between them. The nodes and edges contain information about their types, bond orders, and additional structural features such as branches, rings, and stereochemistry. The missing component is the exact spatial arrangement of the $N = |\mathcal{V}|$ atoms, represented as a 3D point cloud $\boldsymbol{x} \in \mathbb{R}^{N \times 3}$ in Euclidean space. Only the relative distances between atoms are relevant, as translating or rotating the entire point cloud $\boldsymbol{x}$ does not change the identity of the conformer. We frame conformer prediction as sampling from the SE(3)-invariant conditional probability distribution $p(\boldsymbol{x} \mid \mathcal{G})$, which will guide the design of our model architectures in Sec. 4.

### 3.2 Conditional Flow Matching

Starting from an easy-to-sample base distribution $q_0 : \mathbb{R}^d \mapsto \mathbb{R}_{\geq 0}$, a generative process creates samples from a target distribution $q_1 : \mathbb{R}^d \mapsto \mathbb{R}_{\geq 0}$ [27]. Here, $q_1$ models the molecular conformer data with $d = N \times 3$. We aim to learn a *time-dependent vector field* $u_t(\boldsymbol{x}) : [0, 1] \times \mathbb{R}^{N \times 3} \mapsto \mathbb{R}^{N \times 3}$, which defines an ordinary differential equation (ODE) whose solution pushes samples $\boldsymbol{x}_0 \in \mathbb{R}^{N \times 3}$ from the prior to samples $\boldsymbol{x}_1 \in \mathbb{R}^{N \times 3}$ from the data distribution. We describe this transformation in terms of a *stochastic interpolant* $\boldsymbol{x}_t$ [26, 53, 54]. A noisy sample at time $t \in [0, 1]$ is defined as

$$\boldsymbol{x}_t = (1 - t) \cdot \boldsymbol{x}_0 + t \cdot \boldsymbol{x}_1 + \sigma \cdot \boldsymbol{\epsilon}, \tag{1}$$

where $\boldsymbol{\epsilon} \in \mathbb{R}^{N \times 3}$ is drawn from the standard normal distribution $\mathcal{N}(0, \boldsymbol{I})$ and scaled by a constant $\sigma \in \mathbb{R}_{\geq 0}$. We remark that $t$ represents progress along this interpolation path, not physical time. Notably, stochastic interpolants enable transformations between arbitrary distributions and allow us to assess the performance of the generative process under varying prior distributions $q_0$. This contrasts with, e.g., score based diffusion methods [24, 55], which typically assume an isotropic Gaussian prior.

The stochastic interpolant induces a deterministic trajectory of densities $p_t(\boldsymbol{x})$, governed by an ODE known as the probability flow:

$$\mathrm{d}\boldsymbol{x} = u_t(\boldsymbol{x}) \, \mathrm{d}t. \tag{2}$$

If the vector field $u_t(\boldsymbol{x})$ was tractable to sample, the weights of a neural network (NN) $\boldsymbol{v}^\theta(\boldsymbol{x}, t):$ $[0, 1] \times \mathbb{R}^d \mapsto \mathbb{R}^d$ could be optimized directly by minimizing

$$\mathcal{L}_{\text{FM}}(\theta) = \mathbb{E}_{t \sim \mathcal{U}(0,1), \boldsymbol{x} \sim p_t(\boldsymbol{x})} \left\| u_t(\boldsymbol{x}) - \boldsymbol{v}^\theta(\boldsymbol{x}, t) \right\|. \tag{3}$$

The learned vector field $\boldsymbol{v}^\theta$ could then be used to generate new samples from the target distribution by starting from $\boldsymbol{x}_0 \sim q_0$ and integrating the probability flow ODE (Eq. 2), for example, using a numerical scheme such as Euler's method, i.e., $\boldsymbol{x}_{t+\Delta t} = \boldsymbol{x}_t + \boldsymbol{v}^\theta(\boldsymbol{x}_t, t)\Delta t$ for time step $\Delta t$.

However, for arbitrary distributions $q_0$ and $q_1$, the objective in Eq. 3 is computationally intractable [56]. Instead, we consider the expectation over interpolated point pairs from the two distributions. Eq. 1 defines a *conditional probability distribution* $p_t(\boldsymbol{x}|\boldsymbol{x}_0, \boldsymbol{x}_1) = \mathcal{N}(\boldsymbol{x}|(1 - t) \cdot \boldsymbol{x}_0 + t \cdot \boldsymbol{x}_1, \sigma^2)$, with *conditional vector field* $\boldsymbol{u}_t(\boldsymbol{x}|\boldsymbol{x}_0, \boldsymbol{x}_1) = \boldsymbol{x}_1 - \boldsymbol{x}_0$ [27]. The ability to directly sample from the conditional probability via Eq. 1 allows formulating the conditional flow matching (CFM) objective

$$\mathcal{L}_{\text{CFM}}(\theta) = \mathbb{E}_{t \sim \mathcal{U}(0,1), \boldsymbol{x}_0 \sim q_0, \boldsymbol{x}_1 \sim q_1, \boldsymbol{x} \sim p_t(\boldsymbol{x}|\boldsymbol{x}_0, \boldsymbol{x}_1)} \left\| \boldsymbol{u}_t(\boldsymbol{x}|\boldsymbol{x}_0, \boldsymbol{x}_1) - \boldsymbol{v}^\theta(\boldsymbol{x}, t) \right\|^2. \tag{4}$$

As shown in Ref. [26], the gradients of the two losses coincide, $\nabla_\theta \mathcal{L}_{\text{FM}} = \nabla_\theta \mathcal{L}_{\text{CFM}}$, thereby recovering the vector field that defines the probability flow ODE in Eq. 2. Following prior work [37, 57], we reparametrize the training objective to predict noise-free data $\boldsymbol{x}_0^\theta$ directly. During inference, we invert the reparametrization to obtain $\boldsymbol{v}^\theta$ for sampling (see Appendix C for details).

# 4 A New Diffusion Transformer for Molecular Conformer Sampling

We now describe the methodological advances of our work. We propose DiTMC, a new DiT-style architecture for conformer generation by learning a vector field using the loss in Eq. 4. As outlined in Sec. 3.1, this involves sampling from a conditional probability $p(\boldsymbol{x} \,|\, \mathcal{G})$, where $\mathcal{G}$ is the molecular graph representing atomic connectivity. Therefore, we choose our model to be a function $\boldsymbol{v}^\theta(\boldsymbol{x}, t, \mathcal{G})$, where the final output is a 3D velocity vector per atom, which is extracted from a readout layer.

Next, we outline the key components of the DiTMC architecture, with an overview shown in Fig. 1A. Training and architectural details are provided in Appendix B and Appendix D, respectively.

## 4.1 Conditioning Tokens

We begin by defining the conditioning tokens in DiTMC. Each DiTMC block receives a time conditioning token $\boldsymbol{c}^t \in \mathbb{R}^H$, as well as atom-wise conditioning tokens $\mathcal{C}_{\text{atom}}^\mathcal{G} = \{\boldsymbol{c}_i^\mathcal{G} \in \mathbb{R}^H \,|\, i \in [N]\}$, and pair-wise conditioning tokens $\mathcal{C}_{\text{pair}}^\mathcal{G} = \{\boldsymbol{c}_{ij}^\mathcal{G} \in \mathbb{R}^H \,|\, i, j \in [N]\}$ derived from $\mathcal{G} = (\mathcal{V}, \mathcal{E})$.

**Time conditioning** The current time $t$ of the latent state $\boldsymbol{x}_t$ is encoded via a two-layer MLP as

$$\boldsymbol{c}^t = \text{MLP}(t). \tag{5}$$

**Atom-wise conditioning** Atom-wise graph conditioning tokens are obtained from a GNN inspired by the processor module of the MeshGraphNet (MGN) framework [58] as

$$\boldsymbol{c}_i^\mathcal{G} = \text{GNN}_{\text{node}}(\mathcal{V}, \mathcal{E}), \tag{6}$$

where $\boldsymbol{c}_i^\mathcal{G}$ denotes the final node representation for atom $i$. See Appendix D.3 for details on the GNN.

**Pair-wise conditioning** We define pair-wise graph conditioning tokens inspired by the Graphormer architecture [59] as

$$\boldsymbol{c}_{ij}^\mathcal{G} = \text{MLP}(s(i, j)), \tag{7}$$

where $s(i, j)$ denotes the graph geodesic, i.e., the shortest path between atoms $i$ and $j$ in $\mathcal{G}$. This formulation allows conditioning on all atom pairs, including those not directly connected by a bond.

## 4.2 Positional Embeddings

To encode the atomic positions $\mathcal{R} = \{\vec{r}_1, \ldots, \vec{r}_N \,|\, \vec{r}_i \in \mathbb{R}^3\}$ of the latent state $\boldsymbol{x}_t$, we use positional embeddings (PEs). We examine a representative range of positional embeddings that vary in the

number of Euclidean symmetries they respect by construction, which affects how the latent representations transform under translations and rotations. We denote the set of positional embeddings by $\mathcal{P}$, where $\mathcal{P} = \{\boldsymbol{p}_i \,|\, i \in [N]\}$ for the atom-wise (aPE) embeddings, and $\mathcal{P} = \{\boldsymbol{p}_{ij} \,|\, i, j \in [N], \, i \neq j\}$ for pair-wise (rPE or PE(3)) embeddings. Without loss of generality, we assume that the positions are centered such that the center of mass vanishes (see Appendix A).

**Absolute Positional Embeddings** Following Refs. [31, 37], atom-wise absolute Positional Embeddings (aPE) are calculated as

$$\boldsymbol{p}_i^{\text{aPE}} = \text{MLP}(\vec{r}_i) \,, \tag{8}$$

such that $\boldsymbol{p}_i^{\text{aPE}} \in \mathbb{R}^H$. This kind of positional embedding does not preserve rotational nor translational invariance, and serves as a baseline without any symmetry constraints.

**Relative Positional Embeddings** We use displacements vectors $\vec{r}_{ij} = \vec{r}_i - \vec{r}_j$ to build pairwise relative Positional Embeddings (rPE) as

$$\boldsymbol{p}_{ij}^{\text{rPE}} = \text{MLP}(\vec{r}_{ij}) \,, \tag{9}$$

such that $\boldsymbol{p}_{ij}^{\text{rPE}} \in \mathbb{R}^H$. This formulation ensures translational invariance but not rotational invariance.

**Euclidean Positional Embeddings** Adapting ideas from equivariant message passing neural networks like PaiNN [60] or NequIP [61], we construct SO(3)-equivariant pairwise Euclidean Positional Embeddings (PE(3)) as a concatenation of $L + 1$ components

$$\boldsymbol{p}_{ij}^{\text{PE(3)}} = \bigoplus_{\ell=0}^{L} \boldsymbol{\phi}_\ell(r_{ij}) \odot \boldsymbol{Y}_\ell(\hat{r}_{ij}) \,, \tag{10}$$

where $\boldsymbol{\phi}_\ell : \mathbb{R} \mapsto \mathbb{R}^{1 \times H}$ is a radial filter function, $\hat{r} = \vec{r}/r$, and $\boldsymbol{Y}_\ell \in \mathbb{R}^{(2\ell+1) \times 1}$ are spherical harmonics of degree $\ell = 0, \ldots, L$. The element-wise multiplication '$\odot$' between radial filters and spherical harmonics is understood to be "broadcasting" along axes with size 1, such that $(\boldsymbol{\phi}_\ell \odot \boldsymbol{Y}_\ell) \in \mathbb{R}^{(2\ell+1) \times H}$ and (after concatenation) $\boldsymbol{p}_{ij}^{\text{PE(3)}} \in \mathbb{R}^{(L+1)^2 \times H}$. Under rotation of the input positions, these positional embeddings transform equivariantly (see Appendix E). Moreover, because displacement vectors are used as inputs, the embeddings are also invariant to translations. As a result, they respect the full set of Euclidean symmetries relevant to molecular geometry.

## 4.3 DiTMC Block

Based on the positional embedding strategies introduced in Sec. 4.2, we can define different DiTMC blocks that preserve the extent of Euclidean symmetries encoded in the embeddings throughout the model.

Each DiTMC block transforms a set of input tokens $\mathcal{H}$ into a set of output tokens $\mathcal{H}'$, which serve as input for the next block. For aPE and rPE, we have $\mathcal{H} = \{\boldsymbol{h}_1, \ldots, \boldsymbol{h}_N \,|\, \boldsymbol{h}_i \in \mathbb{R}^H\}$, and for PE(3), we have $\mathcal{H} = \{\boldsymbol{h}_1, \ldots, \boldsymbol{h}_N \,|\, \boldsymbol{h}_i \in \mathbb{R}^{(L+1)^2 \times H}\}$.

In each DiTMC block, we inject time-based, as well as graph-based atom-wise and pair-wise conditioning information via the conditioning tokens introduced in Sec. 4.1. We use the pair-wise graph conditioning tokens $\mathcal{C}_{\text{Pair}}^{\mathcal{G}}$ during the self-attention update,

$$\boldsymbol{h}_i = \boldsymbol{h}_i + \text{ATT}(\mathcal{H}, \mathcal{P}, \mathcal{C}_{\text{Pair}}^{\mathcal{G}})_i \,, \tag{11}$$

where we employ a standard self-attention mechanism for aPE and rPE and an SO(3)-equivariant self-attention mechanism for PE(3). Additionally, we use the time conditioning token $\boldsymbol{c}^t$ and the atom-wise graph conditioning tokens $\mathcal{C}_{\text{atom}}^{\mathcal{G}} = \{\boldsymbol{c}_i^{\mathcal{G}} \in \mathbb{R}^H \,|\, i \in [N]\}$ to obtain per-atom bias and scaling parameters,

$$\alpha_{1i}, \beta_{1i}, \gamma_{1i}, \alpha_{2i}, \beta_{2i}, \gamma_{2i} = \text{MLP}(\boldsymbol{c}^t + \boldsymbol{c}_i^{\mathcal{G}}) \,, \tag{12}$$

and apply adaptive layer norm (AdaLN) and adaptive scale (AdaScale) [13] for conditioning (see Fig. 1A) similar to applications of DiTs in image synthesis.

We provide details on the non-equivariant DiTMC blocks based on aPE and rPE in Appendix D.1, and on the SO(3)-equivariant DiTMC block based on PE(3) in Appendix D.2.

Table 1: Results on GEOM-QM9 for different generative models (parameter counts in parentheses). -R indicates Recall, -P indicates Precision. Best results in **bold**, second best underlined; our models are marked with an asterisk (∗). Our results are averaged over three random seeds. See Appendix Tab. A9 for results including standard deviations.

| Method | COV-R [%]↑ | | AMR-R [Å]↓ | | COV-P [%]↑ | | AMR-P [Å]↓ | |
|---|---|---|---|---|---|---|---|---|
| | Mean | Median | Mean | Median | Mean | Median | Mean | Median |
| GeoMol (0.3M) | 91.5 | **100.0** | 0.225 | 0.193 | 86.7 | **100.0** | 0.270 | 0.241 |
| GeoDiff (1.6M) | 76.5 | **100.0** | 0.297 | 0.229 | 50.0 | 33.5 | 0.524 | 0.510 |
| Tors. Diff. (1.6M) | 92.8 | **100.0** | 0.178 | 0.147 | 92.7 | **100.0** | 0.221 | 0.195 |
| MCF-B (64M) | 95.0 | **100.0** | 0.103 | 0.044 | 93.7 | **100.0** | 0.119 | 0.055 |
| DMT-B (55M) | 95.2 | **100.0** | 0.090 | 0.036 | 93.8 | **100.0** | 0.108 | 0.049 |
| ET-Flow (8.3M) | **96.5** | **100.0** | 0.073 | 0.030 | 94.1 | **100.0** | 0.098 | 0.039 |
| ∗DiTMC+aPE-B (9.5M) | 96.1 | **100.0** | 0.073 | 0.030 | 95.4 | **100.0** | 0.085 | 0.037 |
| ∗DiTMC+rPE-B (9.6M) | 96.3 | **100.0** | 0.070 | 0.027 | **95.7** | **100.0** | **0.080** | 0.035 |
| ∗DiTMC+PE(3)-B (8.6M) | 95.7 | **100.0** | **0.068** | **0.021** | 93.4 | **100.0** | 0.089 | **0.032** |

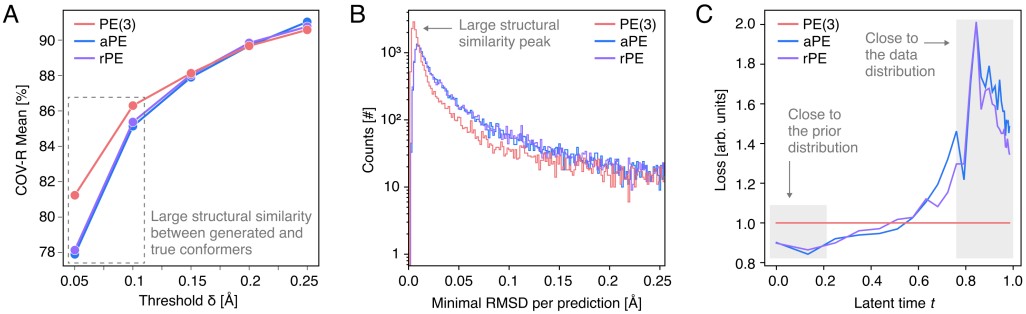

Figure 2: Analysis of SO(3)-equivariant (PE(3)) and non-equivariant (aPE, rPE) model formulations on GEOM-QM9. **(A)** Mean Coverage Recall (COV-R) versus root mean square deviation (RMSD) threshold $\delta$ to any reference conformer. **(B)** Histogram of the minimal RMSD per generated sample. **(C)** Loss as a function of latent time $t$ relative to PE(3) loss (see Appendix J for details).

## 5 Experiments

**Datasets and Metrics** We conduct our experiments on the GEOM dataset [62], comprising QM9 (133,258 small molecules) and AICures (304,466 drug-like molecules). Reference conformers are generated using CREST [63]. Drug-like molecules exhibit greater structural diversity, including more rotatable bonds and multiple stereocenters. The data splits are taken from Ref. [64].

We evaluate our models' ability to generate accurate and diverse conformers using average minimum RMSD (AMR) and coverage (COV), measuring recall (ground-truth coverage) and precision (generation accuracy). A generated conformer is considered valid if it falls within a specified RMSD threshold of any reference conformer ($\delta = 0.5$Å for GEOM-QM9 and $\delta = 0.75$Å for GEOM-DRUGS). Following prior work, we generate $2K$ conformers per test molecule with $K$ reference structures. Appendix G.3 provides further details on the calculation of metrics. Following ET-Flow [32] and GeomMol [64] we also apply chirality correction (see Appendix G.4).

**Ablating Self-Attention and Positional Embedding Strategies** The modular structure of DiTMC allows efficient exploration of the design space through variations in the positional embeddings and associated attention blocks (see Sec. 4). We define three model variants, DiTMC+aPE, DiTMC+rPE, and DiTMC+PE(3), which differ only in the choice of positional embedding and self-attention formulation. Architectural details can be found in Appendix D. On GEOM-QM9, our models produce diverse, high quality samples, outperforming the current state-of-the-art across all AMR-R, AMR-P, and COV-P metrics, demonstrating the broad applicability of our modular design and conditioning (see Tab. 1). We use the harmonic prior introduced in Ref. [65] throughout, which yields improved

Table 2: Ablation of conditioning strategies using DiTMC+aPE-B on GEOM-DRUGS. -R indicates Recall, -P indicates Precision. Best results in **bold**. Our results are averaged over three random seeds. See Appendix Tab. A14 for results on GEOM-QM9.

| Method | COV-R [%]↑ | | AMR-R [Å]↓ | | COV-P [%]↑ | | AMR-P [Å]↓ | |
| --- | --- | --- | --- | --- | --- | --- | --- | --- |
| | Mean | Median | Mean | Median | Mean | Median | Mean | Median |
| No conditioning | 16.6 | 4.7 | 1.132 | 1.110 | 5.15 | 1.2 | 1.849 | 1.845 |
| Atom-wise | 72.8 | 77.5 | 0.555 | 0.565 | 55.6 | 53.3 | 0.762 | 0.716 |
| Atom-wise & Pair-wise | **79.9** | **85.4** | **0.434** | **0.389** | **76.5** | **83.6** | **0.500** | **0.423** |

results compared to the Gaussian prior (see Appendix Tab. A13), and therefore adopt it in all subsequent experiments. Notably, our models maintain competitive performance even when using the Gaussian prior, contrary to the findings in Ref. [32].

**Probing the effect of Euclidean symmetries** Our PEs form a hierarchy based on the extent of Euclidean symmetry incorporated by construction. This enables a systematic evaluation of how incorporating symmetry affects model behavior. We summarize our findings on GEOM-QM9 below.

*Equivariance improves the fidelity of samples.* We analyze COV-R Mean as a function of RMSD threshold $\delta$ for all DiTMC variants (see Fig. 2A). The SO(3)-equivariant DiTMC+PE(3) outperforms the non-equivariant DiTMC+aPE and DiTMC+rPE at low $\delta$, indicating that many of the generated conformers closely match the ground-truth structures. This appears as a leftward shift in the distribution of the minimal RMSD found per generated structure (as shown in Fig. 2B) and aligns with the observation that DiTMC+PE(3) achieves better AMR-R and AMR-P values (as reported in Tab. 1). To better understand this behavior, we examine the loss over time $t$ and find that the non-equivariant models exhibit higher error near the data distribution ($t = 1$) (see Fig. 2C). The increase in error towards the end of the generation trajectory results in noisier structures and reduced fidelity.

*Equivariance increases the computational cost for models of similar size.* The benefit of higher fidelity comes at increased computational cost. During training, the equivariant DiTMC+PE(3) is approximately 3.5 times slower than the non-equivariant DiTMC+aPE and DiTMC+rPE, and about 3 times slower at inference (see Fig. 3B). All models use the same number of layers and differ only in the number of heads per layer in order to match the total parameter count, as discussed in Appendix D.

**Conditioning Strategies** To assess the impact of graph conditioning, we compare different conditioning strategies. As further discussed in Appendix I.2, we compare conditioning solely on atom-wise information (Eq. 6) with an extended scheme that also incorporates pair-wise geodesic graph distances (Eq. 7). In Tab. 2, we show ablation results on GEOM-DRUGS. We also ablate conditioning on pairwise information extracted from our conditioning GNN for each edge corresponding to a chemical bond (see Appendix I.2). We find all variants to be effective, but our proposed combination of geodesic distances and atom-wise information to perform best. This experiment underlines the importance of deriving conditioning tokens for *all* atom pairs, not just those connected by edges in the molecular graph (i.e. by chemical bonds), which lack global information about the graph structure.

**Model Scaling** To investigate model scaling, we define a small ("S"), base ("B"), and large ("L") DiTMC+aPE model variant (see Appendix D). Each model is trained with identical training hyperparameters on the GEOM-DRUGS dataset. We find strong relative improvements for all metrics (up to 54.1% for COV-P Mean) when scaling DiTMC+aPE-S to DiTMC+aPE-B (see Fig. 3A). Scaling DiTMC+aPE-B to DiTMC+aPE-L yields relative improvements, which however might be smaller than expected given the strong increase in performance from the small to the base model. Prior work has shown that to avoid diminishing returns, dataset and model sizes must be scaled together [66, 67]. Scaling only the model yields limited benefits as soon as model size saturates given the amount of data. Strong improvements of DiTMC+aPE-L over DiTMC+aPE-B are observed in terms of sampling fidelity (as indicated by higher COV-R and COV-P Mean values at small thresholds $\delta$) but differences start to vanish at $\delta = 0.75$Å (see Fig. 4A). Moreover, we obtain relative improvement in terms of generalization to larger and previously unseen molecules from the GEOM-XL dataset [46] between 3.1% (for AMR-P Mean) and 7.3% (for AMR-P Median), i.e., DiTMC+aPE-L exhibits significantly stronger out-of-distribution performance compared to the smaller DiTMC+aPE-B (see Tab. 5).

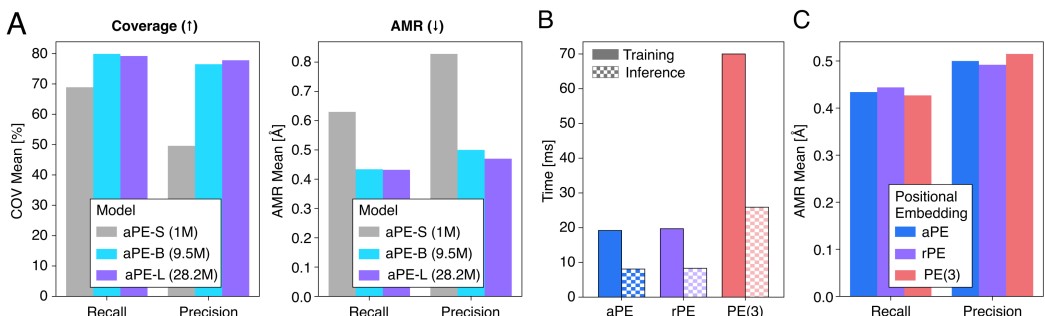

Figure 3: **(A)** Coverage (COV) mean and Absolute Minimum RMSD (AMR) mean for DiTMC+aPE models of increasing model capacity on GEOM-DRUGS. **(B)** Training and inference time for different positional embedding (PE) and associated self-attention strategies. **(C)** Recall and precision AMR mean for the different DiTMC models on GEOM-DRUGS. For panels (B) and (C) we use results from the base ("B") variant of each model.

Table 3: Results on GEOM-DRUGS for different generative models (parameter counts in parentheses). -R indicates Recall, -P indicates Precision. Best results in **bold**, second best underlined; our models are marked with an asterisk (∗). Our results are averaged over three random seeds. See Appendix Tab. A10 for results including standard deviations.

| Method | COV-R [%]↑ | | AMR-R [Å]↓ | | COV-P [%]↑ | | AMR-P [Å]↓ | |
|---|---|---|---|---|---|---|---|---|
| | Mean | Median | Mean | Median | Mean | Median | Mean | Median |
| GeoMol (0.3M) | 44.6 | 41.4 | 0.875 | 0.834 | 43.0 | 36.4 | 0.928 | 0.841 |
| GeoDiff (1.6M) | 42.1 | 37.8 | 0.835 | 0.809 | 24.9 | 14.5 | 1.136 | 1.090 |
| Tors. Diff. (1.6M) | 72.7 | 80.0 | 0.582 | 0.565 | 55.2 | 56.9 | 0.778 | 0.729 |
| MCF-L (242M) | 84.7 | 92.2 | 0.390 | **0.247** | 66.8 | 71.3 | 0.618 | 0.530 |
| DMT-L (150M) | **85.8** | **92.3** | **0.375** | 0.346 | 67.9 | 72.5 | 0.598 | 0.527 |
| ET-Flow - SS (8.3M) | 79.6 | 84.6 | 0.439 | 0.406 | 75.2 | 81.7 | 0.517 | 0.442 |
| ∗DiTMC+aPE-B (9.5M) | 79.9 | 85.4 | 0.434 | 0.389 | 76.5 | 83.6 | 0.500 | 0.423 |
| ∗DiTMC+rPE-B (9.6M) | 79.3 | 84.6 | 0.444 | 0.400 | 77.2 | 84.6 | 0.492 | 0.414 |
| ∗DiTMC+PE(3)-B (8.6M) | 80.8 | 85.6 | 0.427 | 0.396 | 75.3 | 82.0 | 0.515 | 0.437 |
| ∗DiTMC+aPE-L (28.2M) | 79.2 | 84.4 | 0.432 | 0.386 | 77.8 | 85.7 | 0.470 | 0.387 |
| ∗DiTMC+rPE-L (28.3M) | 78.7 | 84.1 | 0.438 | 0.388 | **78.1** | **86.4** | **0.466** | **0.381** |
| ∗DiTMC+PE(3)-L (31.1M) | 80.8 | 85.6 | 0.415 | 0.376 | 76.4 | 82.6 | 0.491 | 0.414 |

**Drug-like Molecules** For the following experiments, we restrict our analysis to the base ("B") and large ("L") DiTMC variants. In contrast to the smaller GEOM-QM9 dataset, the size of GEOM-DRUGS allows us to explore the design space of DiTMC under a more realistic setting, where training is contstrained by a fixed compute budget [67]. Our budget of 9 GPU days allows training DiTMC+aPE-L and DiTMC+rPE-L for 50 epochs, and DiTMC+PE(3)-L for 10 epochs. For consistency, we train the base models for the same number of epochs as their large counterparts. All DiTMC variants achieve state-of-the-art performance on GEOM-DRUGS for COV-P and AMR-P (see Tab. 3). Importantly, even the smaller DiTMC+aPE-B and DiTMC+PE(3)-B models outperform ET-Flow-SS of similar model size across all metrics, underlining the effectiveness of our approach.

*Coverage vs. RMSD Threshold.* We further analyse the COV-R Mean and COV-P Mean as a function of RMSD threshold $\delta$ for DiTMC+aPE-B and DiTMC+aPE-L (see Fig. 4A). For small thresholds ($\delta < 0.4$Å) both DiTMC+aPE models outperform all other methods for coverage recall and precision. For larger thresholds ($\rho \geq 0.4$Å) MCF-L starts to outperform aPE models in terms of COV-R Mean. For COV-P Mean, DiTMC+aPE-B and DiTMC+aPE-L perform better than all other methods for all reasonably small thresholds ($\delta < 1.2$Å). In particular for the most relevant regime where thresholds are small, we see a strong benefit due to model scaling. We find similar results for DiTMC+rPE and DiTMC+PE(3) (see Appendix Fig. A9 and Fig. A10).

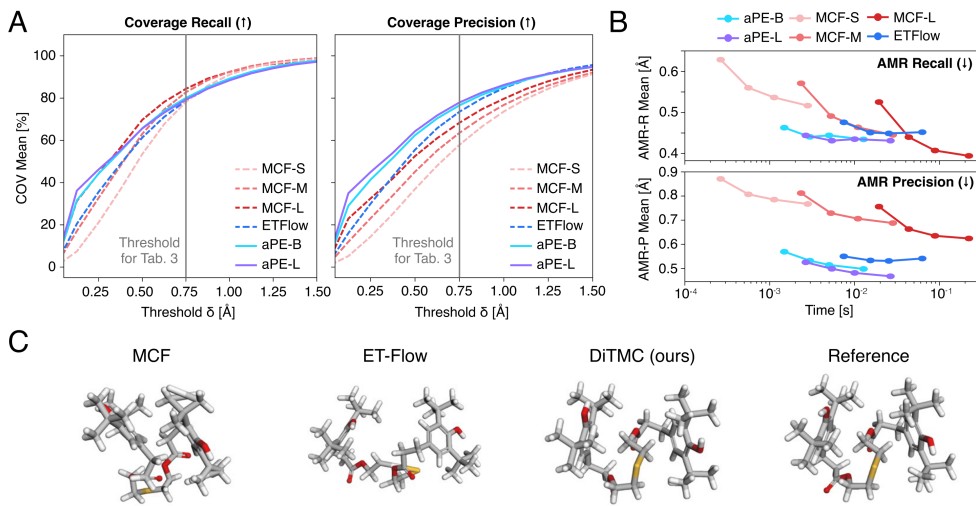

Figure 4: **(A)** Coverage (COV) mean as function of Root Mean Square Deviation (RMSD) threshold $\delta$ and **(B)** average minimum RMSD (AMR) mean vs. time per conformer for DiTMC+aPE and other state-of-the-art models. Per model markers from left to right correspond to 5, 10, 20, and 50 sampler steps following Refs. [32, 47]. Note, that the original MCF paper reports results with two different samplers. Benchmark results (Tab. 3) are obtained with DDPM sampler (1000 steps) and AMR vs. time results are reported for DDIM sampler (5–50 steps). **(C)** Comparison of conformers generated by MCF, ET-Flow, and DiTMC against ground-truth reference conformers from GEOM-XL. Generated conformers are rotationally aligned with their corresponding reference conformer.

*Pareto Front.* Additionally, we investigate the Pareto front of accuracy and computational efficiency, by plotting model accuracy as a function of wall clock time per generated conformer, following Refs. [32, 47] (see Appendix G.5 for details). As measure for accuracy we consider the average minimum RMSD (AMR), since it is independent of the RMSD threshold. We report results of DiTMC+aPE-B and DiTMC+aPE-L for AMR-R Mean and AMR-P Mean in Fig. 4B. For AMR-P Mean, both DiTMC+aPE models shift the whole Pareto front, yielding higher accuracy at lower computational cost. Even higher accuracies (at the cost of compute time) can be obtained by scaling DiTMC+aPE-B to DiTMC+aPE-L. For recall, both DiTMC+aPE models shift the Pareto front for little compute times, but most accurate results at increased cost are obtained by MCF-L. Similar results are obtained for DiTMC+rPE (see Appendix Fig. A12), but benefits for DiTMC+PE(3) are limited due to high computational cost of equivariant operations (see Appendix Fig. A13).

*Ensemble Properties.* To complement the RMSD-based geometric evaluation, we report the median absolute error (MAE) of ensemble properties between generated and reference conformers, following the protocol of MCF [47] and ET-Flow [32] (see Appendix G.6 for details). Our models predict ensemble properties more accurately than all baselines, highlighting the physical validity of our generated structures (see Tab. 4). In particular, MCF, which shows better performance for recall metrics, is outperformed by a large margin (up to a factor of four for energy). The strong performance of the aPE-B and aPE-L models underlines the potential of achieving high physical validity without any geometric priors, which have been hypothesized to be one of the reasons for ET-Flow outperforming MCF in the ensemble property task [32].

**Generalization Performance** Finally, we assess how well our models, trained on GEOM-DRUGS, generalize to larger and previously unseen molecules using the GEOM-XL dataset [46]. It comprises 102 molecules with more than 100 atoms, whereas the molecules in the training set contain only 44 atoms on average. Following MCF [47] we report results for all 102 molecules and a subset of 77 molecules. ET-Flow uses a slightly different subset of 75 molecules. Our aPE models perform on par with or better than the previously best-performing method MCF-L (see Tab. 5), while using approximately 8 times and 25 times fewer parameters for aPE-L and aPE-B, respectively. The other baselines, such as ET-Flow, are outperformed by a larger margin (see Appendix Tab. A12).

Table 4: Median absolute error of ensemble properties between generated and reference conformers for GEOM-DRUGS. Best results in **bold**, second best underlined; our models are marked with an asterisk (∗). Results for MCF, ET-Flow, and ours are averaged over three random seeds. See Appendix Tab. A11 for additional results including standard deviations.

| Method | $E$ [kcal/mol]$\downarrow$ | $\mu$ [D]$\downarrow$ | $\Delta\epsilon$ [kcal/mol]$\downarrow$ | $E_{\min}$ [kcal/mol]$\downarrow$ |
|---|---|---|---|---|
| MCF-L (242M) | 0.68 | 0.28 | 0.63 | 0.04 |
| ET-Flow (8.3M) | 0.18 | 0.18 | 0.35 | 0.02 |
| ∗DiTMC+aPE-B (9.5M) | 0.17 | 0.16 | **0.27** | **0.01** |
| ∗DiTMC+aPE-L (28.2M) | **0.16** | **0.14** | **0.27** | **0.01** |

Table 5: Out-of-distribution generalization results on GEOM-XL for models trained on GEOM-DRUGS. -R indicates Recall, -P indicates Precision. Best results in **bold**, second best underlined; our models are marked with an asterisk (∗). Our results are averaged over three random seeds. See Appendix Tab. A12 for additional results including standard deviations.

| Method | 75 / 77 mols | | | | 102 mols | | | |
|---|---|---|---|---|---|---|---|---|
| | AMR-R [Å]$\downarrow$ | | AMR-P [Å]$\downarrow$ | | AMR-R [Å]$\downarrow$ | | AMR-P [Å]$\downarrow$ | |
| | Mean | Median | Mean | Median | Mean | Median | Mean | Median |
| MCF-L (242M) | 1.64 | 1.51 | 2.57 | 2.26 | 1.97 | 1.60 | 2.94 | 2.43 |
| ET-Flow (8.3M) | 2.00 | 1.80 | 2.96 | 2.63 | 2.31 | 1.93 | 3.31 | 2.84 |
| ∗DiTMC+aPE-B (9.5M) | 1.68 | 1.47 | 2.59 | 2.24 | 1.96 | 1.60 | 2.90 | 2.48 |
| ∗DiTMC+aPE-L (28.2M) | **1.56** | **1.28** | **2.47** | **2.14** | **1.88** | **1.51** | **2.81** | **2.30** |

# 6   Summary and Limitations

We propose a framework for molecular conformer generation that incorporates conditioning strategies tailored to the architectural design principles of DiTs. This modular framework enables a rigorous exploration of different positional embedding and self-attention strategies, allowing us to identify scalable generative architectures that perform competitively with prior methods on standard benchmarks. Our models achieve state-of-the-art results on GEOM-QM9 and GEOM-DRUGS, excelling in both precision and physical validity, and show strong generalization to larger, previously unseen molecules from the GEOM-XL dataset. Exemplary generations for GEOM-XL are shown in Fig. 4C, with additional samples provided in Appendix M.

Through ablation studies, we assess the impact of incorporating Euclidean symmetries into DiTMC. While such symmetries improve performance, they also increase computational cost. Notably, simpler non-equivariant variants remain highly effective. These findings allow us to develop an efficient, accurate, and scalable architecture suitable for large-scale conformer generation.

Nonetheless, some limitations persist. Our evaluation is currently restricted to small and medium-sized molecules, with larger, more flexible compounds left for future work. Moreover, the training process depends on high-quality ground-truth conformers, which may be unavailable in some cases. An interesting direction for future work, is extending the scope of our model beyond conformer generation, for example to de-novo molecular generation [8, 22, 50]. However, this would require a re-design of our proposed conditioning strategies, as those are specifically tailored to molecular graphs as inputs. In terms of computational cost, DiTMC scales quadratic in the number of atoms, resulting in large computational cost for bigger systems. Therefore, integrating recently developed Euclidean fast attention mechanisms [68], poses an interesting direction for further research. Finally, while our analysis advances understanding of equivariance within DiT-based generative models, drawing broader conclusions would require further study.

# 7 Acknowledgements

JTF, WR, KRM, and SC acknowledge support by the German Ministry of Education and Research (BMBF) for BIFOLD (01IS18037A). Further, this work was in part supported by the BMBF under Grants 01IS14013A-E, 01GQ1115, 01GQ0850, 01IS18025A, 031L0207D, and 01IS18037A. KRM was partly supported by the Institute of Information & Communications Technology Planning & Evaluation (IITP) grants funded by the Korea government (MSIT) (No.2019-0-00079, Artificial Intelligence Graduate School Program, Korea University and No. 2022-0-00984, Development of Artificial Intelligence Technology for Personalized Plug-and-Play Explanation and Verification of Explanation). We further want to thank Romuald Elie, Michael Plainer, Adil Kabylda, Khaled Kahouli, Stefan Gugler, Martin Michajlow, Christoph Bornett, Johannes Maeß, Leon Werner and Maximilian Eißler for helpful discussion.

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

## A  From SE(3) to SO(3) Invariance

The target data distribution of molecular conformers $p_1(\boldsymbol{x})$ is SE(3)-invariant, i.e. it does not change under translations and rotations of the input. Following Ref. [69], one can define an SE(3)-invariant measure on $SE(3)^N$ by keeping the center of mass fixed at zero, which can be achieved via the centering operation from Eq. A15. This defines a subgroup $SE(3)_0^N$, called centered SE(3). It can then be shown, that one can define an SE(3)-invariant measure on $SE(3)_0^N$ by constructing an SO(3)-invariant (rotationally invariant) measure on $SE(3)_0^N$.

As a consequence, it is then sufficient to learn an SO(3)-equivariant vector field on the space of centered input positions (see also Ref. [70]). This is achieved by centering $\boldsymbol{x}_0$, $\boldsymbol{x}_1$ and $\boldsymbol{z}$ for the calculation of the interpolant. Moreover, the neural network output (predicted velocities) and the clean target $\boldsymbol{x}_1$ must be centered to have zero center of mass.

We discuss the implications for training in Section B and for sampling in Section C.

## B  Training

Algorithm 1 describes the computation of the training loss for our flow matching objective. We start by sampling from the prior $\boldsymbol{x}_0 \sim p_0(\boldsymbol{x})$, the data distribution $\boldsymbol{x}_1 \sim p_1(\boldsymbol{x})$, and a Gaussian distribution $\boldsymbol{\epsilon} \sim N(\boldsymbol{x}; 0, \boldsymbol{I})$. The interpolant is then defined as

$$\boldsymbol{x}_t = (1 - t) \cdot \boldsymbol{x}_0 + t \cdot \boldsymbol{x}_1 + \sigma \cdot \boldsymbol{\epsilon}\,, \tag{A13}$$

where $\sigma \in \mathbb{R}_{>0}$ is a non-zero noise scaling parameter.

Rather than directly predicting the conditional vector field $\boldsymbol{u}_t(\boldsymbol{x}|\boldsymbol{x}_0, \boldsymbol{x}_1)$, we choose to reparametrize the network such that it predicts the clean sample $\boldsymbol{x}_1$. Similar to Ref. [57], we add a weighting term $1/(1-t)^2$ to encourage the model to accurately capture fine details close to the data distribution. This gives rise to the following loss function

$$\mathcal{L} = \frac{1}{(1-t)^2} \|\boldsymbol{x}_1^\theta(\boldsymbol{x}_t, t, \mathcal{G}) - \boldsymbol{x}_1\|^2\,, \tag{A14}$$

where $t \in (0, 1)$ denotes the timestep in the interpolant $\boldsymbol{x}_t \in \mathbb{R}^{N \times 3}$, $\boldsymbol{x}_1 \in \mathbb{R}^{N \times 3}$ is the clean geometry, and $\mathcal{G} = (\mathcal{V}, \mathcal{E})$ denotes the molecular graph. The full algorithm is summarized in Algorithm 1.

**Geometry Alignment**   Given a set of vectors $\mathcal{U} = \{\vec{u}_1, \ldots, \vec{u}_N \mid \vec{u}_i \in \mathbb{R}^3\}$ associated with a point cloud $\boldsymbol{x} \in \mathbb{R}^{N \times 3}$, we define a centering operation for the $i$-th row

$$\text{Center}(\boldsymbol{x})_i = \vec{u}_i - \frac{1}{N} \sum_{j=1}^N \vec{u}_j\,, \tag{A15}$$

which removes global drift in $\boldsymbol{x}$. Given two point clouds $\boldsymbol{x}_A \in \mathbb{R}^{N \times 3}$ and $\boldsymbol{x}_B \in \mathbb{R}^{N \times 3}$ with positions $\mathcal{R}_A = \{\vec{r}_{1A} \ldots, \vec{r}_{NA}\}$ and $\mathcal{R}_B = \{\vec{r}_{1B} \ldots, \vec{r}_{NB}\}$, we define a rotational alignment operation

$$\text{RotationAlign}(\boldsymbol{x}_A, \boldsymbol{x}_B)_i = \boldsymbol{R}_{\text{opt}} \boldsymbol{x}_{iA}, \tag{A16}$$

where $\boldsymbol{R}_{\text{opt}} \in \mathbb{R}^{3 \times 3}$ is the optimal rotation matrix, minimizing the root mean square deviation (RMSD) between the positions of point clouds A and B, which can be obtained via the Kabsch algorithm [71]. The full geometry alignment operation "GeometryAlign$(\boldsymbol{x}_A, \boldsymbol{x}_B)$", is given by

$$\boldsymbol{x}_A \leftarrow \text{Center}(\boldsymbol{x}_A) \tag{A17}$$
$$\boldsymbol{x}_B \leftarrow \text{Center}(\boldsymbol{x}_B) \tag{A18}$$
$$\boldsymbol{x}_A \leftarrow \text{RotationAlign}(\boldsymbol{x}_A, \boldsymbol{x}_B) \tag{A19}$$

In words, both point clouds are first centered at the origin and then optimally aligned by rotation. This procedure minimizes the path length of a linear interpolation between the point clouds A and B.

---

**Algorithm 1** Conditional Flow Matching Training Loss

---

**Require:** Graph $\mathcal{G}$, target $\boldsymbol{x}_1$, noise level $\sigma$, Model $\boldsymbol{x}_1^\theta$

---

1: $\boldsymbol{x}_0 \sim p_0$, $\boldsymbol{\epsilon} \sim \mathcal{N}(\boldsymbol{0}, \boldsymbol{I})$, $t \sim \mathcal{U}(0, 1)$, $\boldsymbol{R} \sim \mathrm{SO}(3)$
2: $\boldsymbol{x}_0, \boldsymbol{x}_1 \leftarrow \mathrm{GeometryAlign}(\boldsymbol{x}_0, \boldsymbol{x}_1)$ ▷ This centers $\boldsymbol{x}_0$ and $\boldsymbol{x}_1$ and rotation-aligns $\boldsymbol{x}_0$ to $\boldsymbol{x}_1$.
3: $\boldsymbol{\epsilon} \leftarrow \mathrm{Center}(\boldsymbol{\epsilon})$
4: $\boldsymbol{x}_t \leftarrow (1-t)\boldsymbol{x}_0 + t\boldsymbol{x}_1 + \sigma\boldsymbol{\epsilon}$
5: $\boldsymbol{x}_t \leftarrow \mathrm{ApplyRotation}(\boldsymbol{R}, \boldsymbol{x}_t)$
6: $\boldsymbol{x}_1 \leftarrow \mathrm{ApplyRotation}(\boldsymbol{R}, \boldsymbol{x}_1)$
7: $\hat{\boldsymbol{x}}_1 \leftarrow \boldsymbol{x}_1^\theta(\boldsymbol{x}_t, t, \mathcal{G})$
8: $\hat{\boldsymbol{x}}_1 \leftarrow \mathrm{Center}(\hat{\boldsymbol{x}}_1)$

9: **return** $\frac{1}{(1-t)^2}\|\hat{\boldsymbol{x}}_1 - \boldsymbol{x}_1\|^2$

---

**Data Augmentation** The target data distribution of molecular conformers $p_1(\boldsymbol{x})$ is SE(3)-invariant, i.e. it does not change under translations and rotations of the input. One can construct an SE(3)-invariant density by learning an SO(3)-equivariant vector field on centered SE(3) (see Section A). However, only DiTMC+PE(3) is SO(3)-equivariant, whereas aPE and rPE violate SO(3)-equivariance. Therefore, we learn equivariance approximately during training, using data augmentation. Specifically, we randomly sample rotation matrices $\boldsymbol{R}$ (orthogonal matrices with determinant +1) and apply them as

$$\mathrm{ApplyRotation}(\boldsymbol{R}, \boldsymbol{x})_i = \boldsymbol{R}\vec{r}_i, \tag{A20}$$

where $\vec{r}_i$ denotes the positions of the $i$-th atom, i.e., the $i$-th row in the point cloud $\boldsymbol{x} \in \mathbb{R}^{N \times 3}$.

**Noise Scaling Parameter** We ablated the noise scaling parameter $\sigma$ on GEOM-QM9 and GEOM-DRUGS, comparing a larger value of $0.5$ and a smaller value of $0.05$. We set $\sigma$ to the value that empirically worked best for each dataset: $0.05$ for GEOM-QM9 and $0.5$ for GEOM-DRUGS.

**Optimizer and Hyperparameters** We use the AdamW optimizer (weight decay 0.01) with batch size of 128 and learning rate of $\mu_{\mathrm{max}} = 3 \times 10^{-4}$ for GEOM-QM9 and $\mu_{\mathrm{max}} = 1 \times 10^{-4}$ for GEOM-DRUGS. First, we increase the initial learning rate of $\mu_0 = 10^{-5}$ up to $\mu_{\mathrm{max}}$ via a linear learning rate warmup over the first 1% of training steps. Afterwards, the learning rate is decreased via a cosine decay schedule to $\mu_{\mathrm{min}} = 0$ for GEOM-QM9 and $\mu_{\mathrm{min}} = 1 \times 10^{-5}$ for GEOM-DRUGS.

**Compute Budget and Training Times** All models on GEOM-QM9 are trained for 250 epochs, which requires 2 days of training on Nvidia H100 GPU for aPE and rPE models and almost 4 days for PE(3). For GEOM-DRUGS, we fix the total compute budget per model to nine days on a single NVIDIA H100 GPU, due to computational constraints. Within this budget, we can train aPE-L and rPE-L for 50 epochs and PE(3)-L for 10 epochs. To ensure consistency within each PE strategy, the base ("B") model are trained for the same number of epochs as the corresponding large ("L") model.

## C  Sampling

For sampling, we use a simple Euler scheme with 50 steps to sample from the associated ordinary differential equation (ODE) as described in Algorithm 2. Since during training we predict the clean sample $\boldsymbol{x}_1$, we re-parametrize the velocity required for the integration as

$$\boldsymbol{v}_t^\theta(\boldsymbol{x}_t, t, \mathcal{G}) = \frac{\boldsymbol{x}_1^\theta(\boldsymbol{x}_t, t, \mathcal{G}) - \boldsymbol{x}_t}{1 - t}, \tag{A21}$$

where $\boldsymbol{x}_1^\theta(\boldsymbol{x}_t, t, \mathcal{G})$ is the original output of DiTMC.

To ensure SE(3) invariance of the probability path from an (approximately) SO(3)-equivariant velocity predictor, we center the prior $\boldsymbol{x}_0 \sim p_0(\boldsymbol{x})$ and the prediction of DiTMC in each ODE step.

**Algorithm 2** ODE Sampling

**Require:** Model $\boldsymbol{x}_1^\theta$, Graph $\mathcal{G}$, steps $N > 0$

1: $t_n \leftarrow n/N$ for $n \in \{0, \ldots, N\}$
2: $\boldsymbol{x}_0 \sim p_{\text{prior}}(\boldsymbol{x})$ ▷ Sample prior.
3: $\boldsymbol{x}_0 \leftarrow \text{Center}(\boldsymbol{x}_0)$
4: **for** $n \leftarrow 0$ to $N - 1$ **do**
5: $\quad \Delta t \leftarrow t_{n+1} - t_n$ ▷ Compute step size.
6: $\quad \hat{\boldsymbol{x}}_1 \leftarrow \boldsymbol{x}_1^\theta(\boldsymbol{x}_{t_n}, t_n, \mathcal{G})$
7: $\quad \hat{\boldsymbol{x}}_1 \leftarrow \text{Center}(\hat{\boldsymbol{x}}_1)$
8: $\quad \boldsymbol{v} \leftarrow (\hat{\boldsymbol{x}}_1 - \boldsymbol{x}_{t_n})/(1 - t_n)$
9: $\quad \boldsymbol{x}_{t_{n+1}} \leftarrow \boldsymbol{x}_{t_n} + \Delta t \cdot \boldsymbol{v}$ ▷ Euler step.
10: **end for**

11: **return** $\boldsymbol{x}_1$

Table A6: Architectural details for MLPs used in the model. The feature dimension is given as $H = n_{\text{heads}} \cdot d_{\text{head}}$ where $n_{\text{head}}$ is the number of heads and $d_{\text{head}}$ is the number of features per head.

| Name | Layers | Hidden Dim | Out Dim | Activation | Input |
|------|--------|-----------|---------|-----------|-------|
| DiTMC Block | 2 | $4H$ | $H$ | GELU | Tokens |
| SO(3) DiTMC Block | 2 | $4H$ | $H$ | gated GELU | Tokens |
| Time and Atom Cond. | 1 | – | $6H$ | SiLU | $\boldsymbol{c}^t + \boldsymbol{c}_i^\mathcal{G}$ |
| Bond pair | 2 | $H$ | $H$ | SiLU | $\boldsymbol{c}_{ij}^\mathcal{G}$ |
| Time embedding | 2 | $H$ | $H$ | SiLU | $t \in [0, 1]$ |
| Shortest-hop embedding | 2 | $H$ | $H$ | SiLU | Hop distance |
| aPE embedding | 2 | $H$ | $H$ | SiLU | Abs. positions $\vec{r}_i$ |
| rPE embedding | 2 | $H$ | $H$ | SiLU | Rel. positions $\vec{r}_{ij}$ |
| GNN embedding | 2 | $H$ | $H$ | SiLU | Node/edge features |

# D   Architectural Details

In this section, we describe the architecture of DiTMC and the conditioning GNN in more detail.

All DiTMC models rely on conditioning tokens $\mathcal{C}$, for the time $\boldsymbol{c}^t \in \mathbb{R}^H$, per-atom $\boldsymbol{c}_i^\mathcal{G} \in \mathbb{R}^H$, and per atom-pair $\boldsymbol{c}_{ij}^\mathcal{G} \in \mathbb{R}^H$. See main text Section 4.1 for more details. Following Ref. [13], we use adaptive layer norm (AdaLN) and adaptive scale (AdaScale) to include per-atom conditioning tokens based on time $t$ and molecular graph information. To that end, we construct conditioning tokens

$$\boldsymbol{c}_i = \boldsymbol{c}^t + \boldsymbol{c}_i^\mathcal{G}. \tag{A22}$$

Tab. A6 contains details on the MLPs used throughout our architecture, while Tab. A7 summarizes architectural details, as well as training and inference times for all DiTMC variants. Note that the number of attention heads in DiTMC+PE(3) is adjusted to match the total parameter count of the corresponding DiTMC+aPE and DiTMC+rPE variants, while all other hyperparameters are identical.

## D.1   Non-Equivariant DiTMC

In the non-equivariant DiTMC formulations based on aPE and rPE, we have the following set of tokens $\mathcal{H} = \{\boldsymbol{h}_1, \ldots, \boldsymbol{h}_N \mid \boldsymbol{h}_i \in \mathbb{R}^H\}$. Initial token representations are obtained via $\boldsymbol{h}_i = \boldsymbol{e}(z_i) + \boldsymbol{p}_i^{\text{aPE}}$ for aPE and $\boldsymbol{h}_i = \boldsymbol{e}(z_i)$ for rPE, where $\boldsymbol{e}(z_i) \in \mathbb{R}^H$ denotes a learnable embedding based on the atomic number $z_i \in \mathbb{N}_+$ of atom $i$ [35].

Table A7: Architectural details for different PE strategies on GEOM-QM9 and GEOM-DRUGS. Times are measured on GEOM-QM9 with batch size 128 on a single Nvidia H100 GPU. $T$ means number of transformer layers, $n_{\text{heads}}$ number of heads, $d_{\text{head}}$ number of features per head in the attention update, and $T_{MGN}$ number of layers for the conditioning mesh graph net. Thus, total feature dimension is given as $H = n_{\text{heads}} \cdot d_{\text{head}}$.

| Model | $T$ | $n_{\text{heads}}$ | $d_{\text{head}}$ | $T_{MGN}$ | $H$ | Train [ms] | Infer [ms] |
|---|---|---|---|---|---|---|---|
| DiTMC+aPE-S (1M) | 2 | 4 | 32 | 1 | 128 | 5.8 | 2.0 |
| DiTMC+aPE-B (9.5M) | 6 | 8 | 32 | 2 | 256 | 19.2 | 8.1 |
| DiTMC+rPE-B (9.6M) | 6 | 8 | 32 | 2 | 256 | 19.7 | 8.3 |
| DiTMC+PE(3)-B (8.6M) | 6 | 6 | 32 | 2 | 192 | 70.0 | 25.9 |
| DiTMC+aPE-L (28.2M) | 8 | 12 | 32 | 3 | 384 | 32.7 | 9.5 |
| DiTMC+rPE-L (28.3M) | 8 | 12 | 32 | 3 | 384 | 33.5 | 10.1 |
| DiTMC+PE(3)-L (31.1M) | 8 | 10 | 32 | 3 | 320 | 151.6 | 41.8 |

### D.1.1 Self-Attention Operation

For ease of notation, we only describe self-attention with a single head, but employ multi-head attention [29] with $n_{\text{heads}}$ heads in our experiments. All self-attention blocks rely on query, key and value vectors, which are obtained from the input tokens $\mathcal{H} = \{\boldsymbol{h}_1, \dots, \boldsymbol{h}_N \mid \boldsymbol{h}_i \in \mathbb{R}^H\}$ as

$$\boldsymbol{q} = \boldsymbol{W}_q \boldsymbol{h}\,, \qquad \boldsymbol{k} = \boldsymbol{W}_k \boldsymbol{h}\,, \qquad \boldsymbol{v} = \boldsymbol{W}_v \boldsymbol{h}\,, \tag{A23}$$

where $\boldsymbol{W}_q, \boldsymbol{W}_k, \boldsymbol{W}_v \in \mathbb{R}^{H \times H}$ are trainable weight matrices. We define a slightly modified similarity kernel

$$\text{sim}(\boldsymbol{q}, \boldsymbol{k}, \boldsymbol{u}) = \exp\left(\frac{\boldsymbol{q}^\mathsf{T} \cdot (\boldsymbol{k} \odot \boldsymbol{u})}{\sqrt{H}}\right), \tag{A24}$$

where $\boldsymbol{u} \in \mathbb{R}^H$ is used to inject additional information, e.g., conditioning signals and/or positional embeddings, and '$\odot$' denotes element-wise multiplication.

For absolute and relative PEs, we slightly modify standard self-attention to allow injecting pair-wise information into the values in addition to using our modified similarity kernel

$$\text{ATT}(\mathcal{H}, \mathcal{P}, \mathcal{C}_{\text{Pair}}^{\mathcal{G}})_i = \frac{\sum_{j=1}^{N} \text{sim}(\boldsymbol{q}_i, \boldsymbol{k}_j, \boldsymbol{u}_{ij}) \cdot (\boldsymbol{v}_j \odot \boldsymbol{u}_{ij})}{\sum_{j=1}^{N} \text{sim}(\boldsymbol{q}_i, \boldsymbol{k}_j, \boldsymbol{u}_{ij})}\,, \tag{A25}$$

where queries, keys, and values are obtained with Eq. A23 and the injected pair-wise information $\boldsymbol{u}_{ij}$ depends on the positional embedding strategy with

$$\boldsymbol{u}_{ij} = \begin{cases} \boldsymbol{c}_{ij}^{\mathcal{G}} & \text{for absolute PEs}\,, \\ \boldsymbol{c}_{ij}^{\mathcal{G}} + \boldsymbol{p}_{ij}^{\text{rPE}} & \text{for relative PEs}\,. \end{cases} \tag{A26}$$

Here $\boldsymbol{c}_{ij}^{\mathcal{G}} \in \mathbb{R}^H$ are pair-wise graph conditioning tokens (see Eq. 7) and $\boldsymbol{p}_i^{\text{aPE}} \in \mathbb{R}^H$ and $\boldsymbol{p}_{ij}^{\text{rPE}} \in \mathbb{R}^H$ are the absolute and relative PEs described above (see Eqs. 8 and 9).

### D.1.2 Adaptive Layer Normalization and Adaptive Scale

In the standard, non-equivariant setting, we can follow the standard approach of other DiT architectures. We define adaptive layer norm as

$$\text{AdaLN}(\boldsymbol{h}, \boldsymbol{\alpha}, \boldsymbol{\beta}) = \text{LN}(\boldsymbol{h}) \odot (1 + \boldsymbol{\alpha}) + \boldsymbol{\beta}, \tag{A27}$$

where LN is a standard layer normalization without trainable scale and bias, and "$\odot$" denotes entry-wise product.

Adaptive Scale is defined as

$$\text{AdaScale}(\boldsymbol{h}, \boldsymbol{\gamma}) = \boldsymbol{h} \odot \boldsymbol{\gamma}. \tag{A28}$$

In each DiTMC block, we calculate

$$\boldsymbol{\alpha}_{1i}, \boldsymbol{\beta}_{1i}, \boldsymbol{\gamma}_{1i}, \boldsymbol{\alpha}_{2i}, \boldsymbol{\beta}_{2i}, \boldsymbol{\gamma}_{2i} = \boldsymbol{W}\big(\text{SiLU}(\boldsymbol{c}_i)\big), \tag{A29}$$

where $\boldsymbol{W} \in \mathbb{R}^{6H \times H}$ and the output is split into six equally sized vectors $\boldsymbol{\alpha}_{1i}, \boldsymbol{\beta}_{1i}, \boldsymbol{\gamma}_{1i}, \boldsymbol{\alpha}_{2i}, \boldsymbol{\beta}_{2i}, \boldsymbol{\gamma}_{2i} \in \mathbb{R}^H$. The weight matrix $\boldsymbol{W}$ is initialized to all zeros, such that "AdaLN" behaves like identity at initialization. "AdaScale" damps all input tokens to zero at initialization such that the whole DiTMC block behaves like the identity function at initialization.

### D.1.3 Readout

Given final tokens $\boldsymbol{h}$ after performing updates via $T$ DiTMC blocks, we use a readout layer to predict the atomic positions of the clean data sample $\boldsymbol{x}_1$. As in the DiTMC blocks, we employ adaptive LN and therefore calculate

$$\boldsymbol{\alpha}_i, \boldsymbol{\beta}_i = \boldsymbol{W}\big(\text{SiLU}(\boldsymbol{c}_i)\big), \tag{A30}$$

with weight matrix $\boldsymbol{W} \in \mathbb{R}^{2H \times H}$ initialized to all zeros and $\boldsymbol{\alpha}_i, \boldsymbol{\beta}_i \in \mathbb{R}^H$ and do

$$\boldsymbol{h}_i \leftarrow \text{AdaLN}(\boldsymbol{h}_i, \boldsymbol{\alpha}_i, \boldsymbol{\beta}_i),$$
$$\hat{\boldsymbol{x}}_i \leftarrow \boldsymbol{W}_{\text{readout}} \boldsymbol{h}_i,$$

where $\boldsymbol{W}_{\text{readout}} \in \mathbb{R}^{3 \times H}$ is a trainable weight matrix. Thus, we predict a three-dimensional vector per-atom.

## D.2 SO(3) Equivariant DiTMC

Following the notation in Ref. [72], we denote SO(3)-equivariant tokens as $\mathcal{H} = \{\boldsymbol{h}_1, \dots, \boldsymbol{h}_N \mid \boldsymbol{h}_i \in \mathbb{R}^{(L+1)^2 \times H}\}$, where $L$ denotes the maximal degree of the spherical harmonics. We denote the features corresponding to the $\ell$-th degree as $\boldsymbol{h}_i^{(\ell)} \in \mathbb{R}^{(2\ell+1) \times H}$, where the $(2\ell + 1)$ entries corresponds to the orders $m = -\ell, \dots, +\ell$ per degree $\ell$. We refer the reader to Ref. [72] for an in-depth introduction into equivariant features. For the initial token representation $\boldsymbol{h}_i \in \mathbb{R}^{(L+1)^2 \times H}$ we set $\boldsymbol{h}_i^{(0)} = \boldsymbol{e}(z_i)$, where $\boldsymbol{e}(z_i) \in \mathbb{R}^{1 \times H}$ denotes a learnable embedding based on the atomic number $z_i \in \mathbb{N}_+$ for atom $i$ [35]. All higher order features $\boldsymbol{h}_i^{(\ell)}$ with degree $\ell > 0$ are initialized with zero. For all our experiments we use maximal degree of $L = 1$.

### D.2.1 Self-Attention Operation

Our equivariant version of self-attention uses the same transformations for queries, keys and values like the non-equivariant counterpart, as well as the modified similarity kernel (see Appendix sub-subsection D.2.1). However, to preserve all Euclidean symmetries throughout the network, every token must transform equivariantly. One way to achieve this is by separating out the rotational degrees of freedom, encoding them with irreducible representations of the rotation group SO(3). This introduces a "degree-axis" of size $(L + 1)^2$, which encodes angular components of increasing order. The maximum degree $L$ is chosen to ensure high fidelity at a reasonable computational cost. For example, setting $L = 1$ restricts the representation to scalars and vectors, as used in models like PaiNN [60] or TorchMDNet [34]. An SO(3)-equivariant formulation of self-attention is then given as

$$\text{ATT}_{\text{SO(3)}}(\mathcal{H})_i = \frac{\sum_{j=1}^N \text{sim}(\boldsymbol{q}_i, \boldsymbol{k}_j, \boldsymbol{u}_{ij}) \cdot (\hat{\boldsymbol{u}}_{ij} \otimes \boldsymbol{v}_j)}{\sum_{j=1}^N \text{sim}(\boldsymbol{q}_i, \boldsymbol{k}_j, \boldsymbol{u}_{ij})}, \tag{A31}$$

where equivariant queries, keys and values can be calculated similarly to Eq. A23 and '$\otimes$' denotes a Clebsch-Gordan (CG) tensor product contraction [72]. The dot-product in the similarity measure is taken along both feature and degree axes, such that the overall update preserves equivariance (see Appendix Appendix F for details). Tokens and scaling vectors are calculated as

$$\boldsymbol{u}_{ij} = \boldsymbol{\phi}(r_{ij}) \odot \boldsymbol{c}_{ij}^{\mathcal{G}}, \qquad \hat{\boldsymbol{u}}_{ij} = \boldsymbol{p}_{ij}^{\text{PE(3)}} \odot \boldsymbol{c}_{ij}^{\mathcal{G}}, \tag{A32}$$

where $\boldsymbol{\phi}(r_{ij}) \in \mathbb{R}^{(L+1)^2 \times H}$ is a radial filter, and the element-wise products with the pair-wise conditioning tokens $\boldsymbol{c}_{ij}^{\mathcal{G}} \in \mathbb{R}^{1 \times H}$ are broadcast along the degree axis. Importantly, the $2\ell + 1$

subcomponents of the radial filter for degree $\ell$ are obtained by repeating per-degree filter functions $\phi_\ell(r_{ij}) \in \mathbb{R}^{1 \times H}$ along the degree axis to preserve equivariance (see also Eq. 10).

Since, standard MLPs do not preserve SO(3)-equivariance, we use equivariant MLPs for the node-wise refinement after the self-attention calculation via an equivariant formulation for dense layers and so-called gated non-linearities (see e.g. Ref. [72] for more details).

### D.2.2  Adaptive Layer Normalization and Adaptive Scale

In the SO(3)-equivariant case, we define an adapted version of AdaLN and AdaScale, preserving equivariance. Our equivariant formulation of AdaLN is given as

$$
\text{EquivAdaLN}(\boldsymbol{h}, \boldsymbol{\alpha}, \boldsymbol{\beta}) = \begin{cases} \text{LN}(\boldsymbol{h}^{(\ell)}) \odot (1 + \boldsymbol{\alpha}^{(0)}) + \boldsymbol{\beta} & \text{for } \ell = 0, \\ \text{EquivLN}(\boldsymbol{h}^{(\ell)}) \odot (1 + \boldsymbol{\alpha}^{(\ell)}) & \text{for } \ell > 0, \end{cases}
\tag{A33}
$$

where EquivLN is the equivariant formulation of layer normalization following Ref. [36] without trainable per-degree scales and LN is standard layer normalization without trainable scale and bias. Scaling vectors $\boldsymbol{\alpha}^{(\ell)} \in \mathbb{R}^{1 \times H}$ are defined per degree $\ell$, such that input scaling vectors are tensors $\boldsymbol{\alpha} \in \mathbb{R}^{(L+1) \times H}$. Bias vectors are only defined for the invariant ($\ell = 0$) component of the tokens, since adding a non-zero bias to components with $\ell > 0$ would lead to a non-equivariant operation (the bias does not transform under rotations). The element wise multiplication between $(1 + \boldsymbol{\alpha}^{(\ell)}) \in \mathbb{R}^{1 \times H}$ and tokens $h^{(\ell)} \in \mathbb{R}^{(2\ell+1) \times H}$ is "broadcasted" along the degree-axis. For $L = 0$, Eq. A33 reduces to the standard adaptive layer normalization.

Equivariant adaptive scale is defined as

$$
\text{EquivAdaScale}(\boldsymbol{h}, \boldsymbol{\gamma}) = \boldsymbol{h}^{(\ell)} \odot \boldsymbol{\gamma}^{(\ell)}.
\tag{A34}
$$

As for "EquivAdaLN", we define a separate $\boldsymbol{\gamma}^{(\ell)} \in \mathbb{R}^{1 \times H}$ per degree $\ell$, such that $\boldsymbol{\gamma} \in \mathbb{R}^{(L+1) \times H}$. Again, the element wise product is "broadcasted" along the degree-axis. Since no bias is involved, the invariant and equivariant parts in $\boldsymbol{h}$ can be treated equally.

Within each SO(3)-equivariant DiTMC block, we calculate

$$
\boldsymbol{\alpha}_{1i}, \boldsymbol{\beta}_{1i}, \boldsymbol{\gamma}_{1i}, \boldsymbol{\alpha}_{2i}, \boldsymbol{\beta}_{2i}, \boldsymbol{\gamma}_{2i} = \boldsymbol{W}\big(\text{SiLU}(\boldsymbol{c}_i)\big),
\tag{A35}
$$

where $\boldsymbol{\alpha}_{1i}, \boldsymbol{\alpha}_{2i}, \boldsymbol{\gamma}_{1i}, \boldsymbol{\gamma}_{2i} \in \mathbb{R}^{(L+1) \times H}$ and $\boldsymbol{\beta}_{1i}, \boldsymbol{\beta}_{2i} \in \mathbb{R}^H$. Thus, the weight matrix is given as $\boldsymbol{W} \in \mathbb{R}^{(4(L+1)+2) \times H}$ and initialized to all zeros, such that "EquivAdaLN" behaves like identity at initialization and "EquivAdaScale" returns zeros. Thus, also the SO(3)-equivariant DiTMC block behaves like the identity function at initialization.

### D.2.3  Readout

Given final equivariant features $\boldsymbol{h}_i \in \mathbb{R}^{(L+1)^2 \times H}$ we use a readout layer to predict the atomic positions of the clean data sample $\boldsymbol{x}_1$. We employ our equivariant formulation of adaptive layer normalization and calculate

$$
\boldsymbol{\alpha}_i, \boldsymbol{\beta}_i = \boldsymbol{W}\big(\text{SiLU}(\boldsymbol{c}_i)\big),
\tag{A36}
$$

with weight matrix $\boldsymbol{W} \in \mathbb{R}^{2H(L+1) \times H}$ initialized to all zeros and $\boldsymbol{\alpha}_i, \boldsymbol{\beta}_i \in \mathbb{R}^{H(L+1)}$. We then do,

$$
\boldsymbol{h}_i \leftarrow \text{EquivAdaLN}(\boldsymbol{h}_i, \boldsymbol{\alpha}_i, \boldsymbol{\beta}_i),
$$

$$
\boldsymbol{y}_i \leftarrow \boldsymbol{W}_{\text{readout}} \boldsymbol{h}_i^{(\ell=1)},
$$

where $\boldsymbol{W}_{\text{readout}} \in \mathbb{R}^{1 \times H}$ is a trainable weight vector that is applied along the feature axis in $\boldsymbol{h}$. Since $\boldsymbol{h}_i^{(\ell=1)} \in \mathbb{R}^{3 \times H}$ this produces per-atom vectors $\hat{\boldsymbol{y}}_i \in \mathbb{R}^3$. As $\boldsymbol{h}_i$ are rotationally equivariant so is $\hat{\boldsymbol{y}}_i \in \mathbb{R}^3$.

### D.3  Conditioning GNN

Our conditioning GNN directly operates on the molecular graph $\mathcal{G} = (\mathcal{V}, \mathcal{E})$ to obtain graph-based information for atom-wise conditioning. First, we processes the molecular graph $\mathcal{G}$ to derive node

and edge input features, initially represented as one-hot vectors (a full list of features is provided in Tab. A8). These features then are projected into a shared latent space using two-layer multilayer perceptrons (MLPs). Finally, we employ a GNN architecture inspired by the processor described in the MeshGraphNet (MGN) framework [58], which refines the features via message passing.

The conditioning GNN maintains and updates both node and edge representations across multiple layers. Each message passing block consists of two main steps: first, the edge representations are updated based on the current edge representations and the representations of the connected nodes:

$$\boldsymbol{e}'_{ij} \leftarrow f_e(\boldsymbol{e}_{ij}, \boldsymbol{v}_i, \boldsymbol{v}_j) \tag{A37}$$

where $\mathbf{e}_{ij}$ and $\mathbf{v}_i$ denote the input edge and node representations, and $\mathbf{e}'_{ij}$ are the updated edge representations. The learnable function $f_e$ is implemented as a two-layer MLP. Next, node representations $\mathbf{v}_i$ are updated to $\mathbf{v}'_i$ using aggregated messages from neighboring edges:

$$\boldsymbol{v}'_i \leftarrow f_v \left( \boldsymbol{v}_i, \sum_j \boldsymbol{e}'_{ij} \right) \tag{A38}$$

where $f_v$ is also a two-layer MLP, and the summation is over all edges ending at node $i$.

The final output of the described conditioning GNN is a set of node embeddings per atom, and a set of edge embeddings per bond. We use the final node embeddings as atom-wise graph conditioning tokens in DiTMC as detailed in Sec. 4.1. We want to highlight that one can also use the final edge embedding as pair-wise graph conditioning tokens in DiTMC. We discuss this further in Appendix I.2.

# E  Equivariance Proof for Euclidean Positional Embeddings

Given a set of transformations that act on a vector space $\mathbb{A}$ as $S_g : \mathbb{A} \mapsto \mathbb{A}$ to which we associate an abstract group $G$, a function $f : \mathbb{A} \mapsto \mathbb{B}$ is said to be equivariant w.r.t. $G$ if

$$f(S_g x) = T_g f(x) \,, \tag{A39}$$

where $T_g : \mathbb{B} \mapsto \mathbb{B}$ is an equivalent transformation on the output space. Thus, in order to say that $f$ is equivariant, it must hold that under transformation of the input, the output transforms "in the same way".

Let us now recall our definition for the equivariant positional embeddings for a single degree $\ell$

$$\boldsymbol{p}_{ij}^{\mathrm{PE(3)},(\ell)}(\vec{r}_{ij}) = \boldsymbol{\phi}_\ell(||\vec{r}_{ij}||) \odot \boldsymbol{Y}_\ell(\hat{r}_{ij}) \,, \tag{A40}$$

where $\boldsymbol{\phi}_\ell : \mathbb{R} \mapsto \mathbb{R}^{1 \times H}$ is a radial filter function, $\hat{r} = \vec{r}/r$, and $\boldsymbol{Y}_\ell \in \mathbb{R}^{(2\ell+1) \times 1}$ are spherical harmonics of degree $\ell = 0 \dots L$. The element-wise multiplication '$\odot$' between radial filters and spherical harmonics is understood to be "broadcasting" along axes with size 1, such that $(\boldsymbol{\phi}_\ell \odot \boldsymbol{Y}_\ell) \in \mathbb{R}^{(2\ell+1) \times H}$. We have also made the dependence of PE(3) on the pairwise displacement vector $\vec{r}_{ij} = \vec{r}_i - \vec{r}_j$ explicit.

Lets not consider a single feature channel $c$ in PE(3), which is given as

$$\boldsymbol{p}_{ijc}^{\mathrm{PE(3)},(\ell)}(\vec{r}_{ij}) = \boldsymbol{\phi}_{\ell c}(||\vec{r}_{ij}||) \odot \boldsymbol{Y}_\ell(\hat{r}_{ij}) \,. \tag{A41}$$

Rotating the input positions in Eq. A41 leads to

$$\boldsymbol{p}_{ijc}^{\mathrm{PE(3)},(\ell)}(\boldsymbol{R}\vec{r}_{ij}) = \boldsymbol{\phi}_{\ell c}(||\boldsymbol{R}\vec{r}_{ij}||) \odot \boldsymbol{Y}_\ell(\boldsymbol{R}\hat{r}_{ij}) \tag{A42}$$

$$= \boldsymbol{\phi}_{\ell c}(||\vec{r}_{ij}||) \odot \boldsymbol{D}^{(\ell)}(\boldsymbol{R})\boldsymbol{Y}_\ell(\hat{r}_{ij}), \tag{A43}$$

$$= \boldsymbol{D}^{(\ell)}(\boldsymbol{R})\boldsymbol{p}_{ijc}^{\mathrm{PE(3)},(\ell)}(\vec{r}_{ij}) \tag{A44}$$

where $\boldsymbol{D}^{(\ell)} \in \mathbb{R}^{(2\ell+1) \times (2\ell+1)}$ are the Wigner-D matrices for degree $\ell$ and $\boldsymbol{R} \in \mathbb{R}^{3 \times 3}$ is a rotation matrix. According to Eq. A39 and Eq. A44, each channel transforms equivariant and thus $\boldsymbol{p}_{ij}^{\mathrm{PE(3)},(\ell)}(\vec{r}_{ij})$ is also equivariant.

The concatenation of different degrees $\ell$ up to some maximal degree $L$ as given in the main body of the text

$$\boldsymbol{p}_{ij}^{\mathrm{PE(3)}}(\vec{r}_{ij}) = \bigoplus_{\ell=0}^{L} \phi_\ell(r_{ij}) \odot \boldsymbol{Y}_\ell(\hat{r}_{ij}), \tag{A45}$$

transforms under rotation as

$$\boldsymbol{p}_{ij}^{\mathrm{PE(3)}}(\boldsymbol{R}\vec{r}_{ij}) = \boldsymbol{D}(\boldsymbol{R})\, \boldsymbol{p}_{ij}^{\mathrm{PE(3)}}(\vec{r}_{ij}) \tag{A46}$$

with $\boldsymbol{D}(\boldsymbol{R}) = \bigoplus_{\ell=0}^{L} \boldsymbol{D}^{(\ell)}(\boldsymbol{R}) \in \mathbb{R}^{(L+1)^2 \times (L+1)^2}$ being a block-diagonal matrix with the Wigner-D matrices of degree $\boldsymbol{D}^{(\ell)}(\boldsymbol{R}) \in \mathbb{R}^{(2\ell+1) \times (2\ell+1)}$ along the diagonal. Therefore, according to Eq. A39 the proposed positional embeddings $\boldsymbol{p}_{ij}^{\mathrm{PE(3)}}$ are SO(3)-equivariant.

## F    Invariance Proof for the Dot-Product

In the self-attention update, the dot-product between query and key is computed as along the degree and the feature axis. Under rotation, the equivariant features behave as

$$\boldsymbol{h}(\boldsymbol{R}\vec{r}) = \boldsymbol{D}(\boldsymbol{R})\boldsymbol{h}(\vec{r}), \tag{A47}$$

where $\boldsymbol{D}(\boldsymbol{R})$ is the concatenation of Wigner-D matrices from above. The inner product along the degree axis for two features $\boldsymbol{h}$ and $\boldsymbol{g}$ behaves under rotation as

$$\boldsymbol{g}(\boldsymbol{R}\vec{r})^T \cdot \boldsymbol{h}(\boldsymbol{R}\vec{r}) = \boldsymbol{g}(\vec{r})^T \underbrace{\boldsymbol{D}(\boldsymbol{R})^T \boldsymbol{D}(\boldsymbol{R})}_{=\mathrm{Id}} \boldsymbol{h}(\vec{r}) = \boldsymbol{g}(\vec{r})^T \cdot \boldsymbol{h}(\vec{r}), \tag{A48}$$

where we made use of the fact that the Wigner-D matrices are orthogonal matrices. Thus, the dot-product along the degree-axis is invariant and therefore taking the dot-product along the degree and then along the feature axis is also invariant.

## G    Implementation details

### G.1    Data Preprocessing

For both GEOM-QM9 and GEOM-DRUGS, we use the first 30 conformers for each molecule with the lowest energies, i.e., highest Boltzmann weights. We use the train/test/val split from Geomol [64], using the same 1000 molecules for testing.

### G.2    Input Featurization

Tab. A8 defines the features we use for each atom or bond. Each feature is computed using RDKit [73] and one-hot encoded before being passed to the network.

### G.3    Evaluation Metrics

During evaluation, we follow the same procedure as described in Refs. [32, 46, 47, 64]. The root-mean-square deviation (RMSD) metric measures the average distance between atoms of a generated conformer with respect to its reference, while taking into account all possible symmetries. For $L = 2K$ let $\{\hat{C}_l\}_{l \in \{1,\dots,L\}}$ and $\{C_k\}_{k \in \{1,\dots,K\}}$ be the sets of generated conformers and reference conformers respectively. The average minimum RMSD (AMR) and coverage (COV) metrics for both recall (R) and precision (P) are defined as follows, where $\delta > 0$ is the coverage threshold:

Table A8: Atomic and bond features included in DiT-MC. All features are one-hot encoded.

| Atom features | Options |
| --- | --- |
| Chirality | `TETRAHEDRAL_CW, TETRAHEDRAL_CCW, UNSPECIFIED, OTHER` |
| Number of hydrogens | 0, 1, 2, 3, 4 |
| Number of radical electrons | 0, 1, 2, 3, 4 |
| Atom type (QM9) | H, C, N, O, F |
| Atom type (DRUGS) | H, Li, B, C, N, O, F, Na, Mg, Al, Si, P, S, Cl, K, Ca, V, Cr, Mn, Cu, Zn, Ga, Ge, As, Se, Br, Ag, In, Sb, I, Gd, Pt, Au, Hg, Bi |
| Aromaticity | true, false |
| Degree | 0, 1, 2, 3, 4, other |
| Hybridization | $sp, sp^2, sp^3, sp^3d, sp^3d^2$, other |
| Implicit valence | 0, 1, 2, 3, 4, other |
| Formal charge | -5, -4, ..., 5, other |
| Presence in ring of size x | x = 3, 4, 5, 6, 7, 8, other |
| Number of rings atom is in | 0, 1, 2, 3, other |

| Bond features | Options |
| --- | --- |
| Bond type | single, double, triple, aromatic |

$$\text{COV-R}(C, \hat{C}, \delta) := \frac{1}{K} \left| \left\{ k \in \{1, \ldots, K\} \mid \exists_{l \in \{1, \ldots, L\}} \text{RMSD}(\hat{C}_l, C_k) < \delta \right\} \right| \tag{A49}$$

$$\text{COV-P}(C, \hat{C}, \delta) := \frac{1}{L} \left| \left\{ l \in \{1, \ldots, L\} \mid \exists_{k \in \{1, \ldots, K\}} \text{RMSD}(\hat{C}_l, C_k) < \delta \right\} \right| \tag{A50}$$

$$\text{AMR-R}(C, \hat{C}) := \frac{1}{K} \sum_{k \in \{1, \ldots, K\}} \min_{l \in \{1, \ldots, L\}} \text{RMSD}(\hat{C}_l, C_k) \tag{A51}$$

$$\text{AMR-P}(C, \hat{C}) := \frac{1}{L} \sum_{l \in \{1, \ldots, L\}} \min_{k \in \{1, \ldots, K\}} \text{RMSD}(\hat{C}_l, C_k) \tag{A52}$$

### G.4 Chirality Correction

Given the four 3D coordinates around a chirality center denoted as $\mathbf{p_1}, \mathbf{p_2}, \mathbf{p_3}, \mathbf{p_4} \in \mathbb{R}^3$ with $\mathbf{p_i} = (x_i, y_i, z_i)$ for $i = 1, 2, 3, 4$, we can compute the oriented volume $OV$ of the tetrahedron via

$$
\begin{aligned}
OV(\mathbf{p_1}, \mathbf{p_2}, \mathbf{p_3}, \mathbf{p_4}) &= \frac{1}{6} \cdot \det \left( \begin{bmatrix} 1 & 1 & 1 & 1 \\ x_1 & x_2 & x_3 & x_4 \\ y_1 & y_2 & y_3 & y_4 \\ z_1 & z_2 & z_3 & z_4 \end{bmatrix} \right) \\
&= \frac{1}{6} \cdot (\mathbf{p_1} - \mathbf{p_4}) \cdot ((\mathbf{p_2} - \mathbf{p_4}) \times (\mathbf{p_3} - \mathbf{p_4})).
\end{aligned}
\tag{A53}
$$

Following GeomMol [64] and ET-Flow [32], we can then compare the orientation of the volume given by $\text{sign}(OV)$ with the local chirality label produced by RDKit, which corresponds to a certain orientation as well (CW = +1 and CCW = -1) [74]. If the orientation of the volume differs from the RDKit label, we correct the chirality of the conformer by reflecting its positions against the $z$-axis.

### G.5 Pareto Front

For all models, we generate conformers using 5, 10, 20 and 50 sampling steps on a single A100 GPU with a batch size of 128, following Refs. [32, 47]. The wall-clock time per generated sample is obtained by measuring the average time per batch and dividing by the batch size. As done in the original paper, we adopt DDIM sampling for MCF-S, MCF-B and MCF-L.

Table A9: Results on GEOM-QM9 for different generative models (number of parameters in parentheses). -R indicates Recall, -P indicates Precision. Best results in **bold**, second best underlined; our models are marked with an asterisk "∗". Our results are averaged over three random seeds with standard deviation reported below. Other works do not report standard deviations.

| Method | COV-R [%]↑ | | AMR-R [Å]↓ | | COV-P [%]↑ | | AMR-P [Å]↓ | |
|---|---|---|---|---|---|---|---|---|
| | Mean | Median | Mean | Median | Mean | Median | Mean | Median |
| GeoMol (0.3M) | 91.5 | **100.0** | 0.225 | 0.193 | 86.7 | **100.0** | 0.270 | 0.241 |
| GeoDiff (1.6M) | 76.5 | **100.0** | 0.297 | 0.229 | 50.0 | 33.5 | 0.524 | 0.510 |
| Tors. Diff. (1.6M) | 92.8 | **100.0** | 0.178 | 0.147 | 92.7 | **100.0** | 0.221 | 0.195 |
| MCF-B (64M) | 95.0 | **100.0** | 0.103 | 0.044 | 93.7 | **100.0** | 0.119 | 0.055 |
| DMT-B (55M) | 95.2 | **100.0** | 0.090 | 0.036 | 93.8 | **100.0** | 0.108 | 0.049 |
| ET-Flow (8.3M) | **96.5** | **100.0** | 0.073 | 0.030 | 94.1 | **100.0** | 0.098 | 0.039 |
| ∗DiTMC+aPE-B (9.5M) | 96.1 ±0.3 | **100.0** ±0.0 | 0.074 ±0.001 | 0.030 ±0.001 | 95.4 ±0.1 | **100.0** ±0.0 | 0.085 ±0.001 | 0.037 ±0.000 |
| ∗DiTMC+rPE-B (9.6M) | 96.3 ±0.0 | **100.0** ±0.0 | 0.070 ±0.001 | 0.027 ±0.000 | **95.7** ±0.1 | **100.0** ±0.0 | **0.080** ±0.000 | 0.035 ±0.000 |
| ∗DiTMC+PE(3)-B (8.6M) | 95.7 ±0.3 | **100.0** ±0.0 | **0.068** ±0.002 | **0.021** ±0.001 | 93.4 ±0.2 | **100.0** ±0.0 | 0.089 ±0.002 | **0.032** ±0.001 |

## G.6   Ensemble Properties

We adopt the property prediction task setup from MCF [47] and ET-Flow [32], where we draw a subset of 100 randomly sampled molecules from the test set of GEOM-DRUGS and generate $\min(2K, 32)$ conformers for a molecule with $K$ ground truth conformers. Afterwards we relax the conformers using GFN2-xTB [75] and compare the Boltzmann-weighted properties of the generated and ground truth ensembles. More specifically, we employ xTB [75] to calculate the energy $E$, the dipole moment $\mu$, the HOMO–LUMO gap $\Delta\epsilon$ and the minimum energy $E_{\min}$. We repeat this procedure for three subsets each sampled with a different random seed and report the averaged median absolute error and standard deviation of the different ensemble properties.

# H   Additional Experimental Results

## H.1   Results on GEOM-QM9

We report the results on GEOM-QM9 including standard deviations for all DiTMC models in Tab. A9.

## H.2   Results on GEOM-DRUGS

We report the results on GEOM-DRUGS including standard deviations for all DiTMC models in Tab. A10. The median absolute error of ensemble properties on GEOM-DRUGS is shown in Tab. A11.

Additional results for the coverage vs. RMSD threshold analysis, including results for DiTMC+rPE and DiTMC+PE(3), are provided in Fig. A8, Fig. A9, and Fig. A10.

Additional results for the Pareto front analysis, including results for DiTMC+rPE and DiTMC+PE(3), are provided in Fig. A11, Fig. A12, and Fig. A13.

## H.3   Results on GEOM-XL

We report the results on GEOM-XL including standard deviations for all DiTMC models in Tab. A12.

Table A10: Results on GEOM-DRUGS for different generative models (number of parameters in parentheses). -R indicates Recall, -P indicates Precision. Best results in **bold**, second best underlined; our models are marked with an asterisk "∗". Our results are averaged over three random seeds with standard deviation reported below. Other works do not report standard deviations.

| Method | COV-R [%]↑ | | AMR-R [Å]↓ | | COV-P [%]↑ | | AMR-P [Å]↓ | |
|---|---|---|---|---|---|---|---|---|
| | Mean | Median | Mean | Median | Mean | Median | Mean | Median |
| GeoMol (0.3M) | 44.6 | 41.4 | 0.875 | 0.834 | 43.0 | 36.4 | 0.928 | 0.841 |
| GeoDiff (1.6M) | 42.1 | 37.8 | 0.835 | 0.809 | 24.9 | 14.5 | 1.136 | 1.090 |
| Tors. Diff. (1.6M) | 72.7 | 80.0 | 0.582 | 0.565 | 55.2 | 56.9 | 0.778 | 0.729 |
| MCF-S (13M) | 79.4 | 87.5 | 0.512 | 0.492 | 57.4 | 57.6 | 0.761 | 0.715 |
| MCF-B (64M) | 84.0 | 91.5 | 0.427 | 0.402 | 64.0 | 66.2 | 0.667 | 0.605 |
| MCF-L (242M) | 84.7 | 92.2 | 0.390 | **0.247** | 66.8 | 71.3 | 0.618 | 0.530 |
| DMT-L (150M) | **85.8** | **92.3** | **0.375** | 0.346 | 67.9 | 72.5 | 0.598 | 0.527 |
| ET-Flow - SS (8.3M) | 79.6 | 84.6 | 0.439 | 0.406 | 75.2 | 81.7 | 0.517 | 0.442 |
| ∗DiTMC+aPE-B (9.5M) | 79.9 ±0.1 | 85.4 ±0.3 | 0.434 ±0.002 | 0.389 ±0.002 | 76.5 ±0.1 | 83.6 ±0.3 | 0.500 ±0.002 | 0.423 ±0.004 |
| ∗DiTMC+rPE-B (9.6M) | 79.3 ±0.1 | 84.6 ±0.2 | 0.444 ±0.002 | 0.400 ±0.002 | 77.2 ±0.1 | 84.6 ±0.2 | 0.492 ±0.001 | 0.414 ±0.002 |
| ∗DiTMC+PE(3)-B (8.6M) | 80.8 ±0.1 | 85.6 ±0.5 | 0.427 ±0.001 | 0.396 ±0.001 | 75.3 ±0.1 | 82.0 ±0.2 | 0.515 ±0.000 | 0.437 ±0.003 |
| ∗DiTMC+aPE-L (28.2M) | 79.2 ±0.1 | 84.4 ±0.2 | 0.432 ±0.003 | 0.386 ±0.003 | 77.8 ±0.1 | 85.7 ±0.5 | 0.470 ±0.001 | 0.387 ±0.003 |
| ∗DiTMC+rPE-L (28.3M) | 78.7 ±0.1 | 84.1 ±0.4 | 0.438 ±0.002 | 0.388 ±0.005 | **78.1** ±0.1 | **86.4** ±0.3 | **0.466** ±0.001 | **0.381** ±0.003 |
| ∗DiTMC+PE(3)-L (31.1M) | 80.8 ±0.3 | 85.6 ±0.1 | 0.415 ±0.003 | 0.376 ±0.001 | 76.4 ±0.2 | 82.6 ±0.3 | 0.491 ±0.002 | 0.414 ±0.004 |

# I    Additional Ablations

## I.1    Gaussian vs. Harmonic Prior

As shown in Tab. A13, using the harmonic prior improves all metrics slightly for our models on GEOM-QM9. Using the harmonic prior however doesn't seem to be a crucial ingredient for the success of our method, as differences between Gaussian and Harmonic prior appear diminishing. As the results in Tab. A13 verify, our method can also be used with a simple Gaussian prior effectively. For larger molecular graphs the expensive eigendecomposition of the graph Laplacian required for the Harmonic prior could therefore be avoided, which helps scaling our approach more easily.

## I.2    Conditioning Strategies on GEOM-QM9

To evaluate the effectiveness of various graph conditioning strategies in DiTMC , we compare the performance of different conditioning methods against a baseline model without any conditioning. In addition to conditioning strategies discussed in Sec. 4.1, we note that our conditioning GNN also produces edge-level representations, which can be used to define pair-wise graph conditioning tokens as

$$\textbf{bond-pair:} \quad c_{ij}^{\mathcal{G}} = \begin{cases} \text{GNN}_{\text{edge}}(\mathcal{V}, \mathcal{E}) & \forall (i,j) \in \mathcal{E} \\ \bar{c}^{\mathcal{G}} & \forall (i,j) \notin \mathcal{E} . \end{cases} \quad \text{(A54)}$$

These tokens only capture interactions between bonded atoms, i.e., when $(i,j) \in \mathcal{E}$. Conditioning tokens for non-bonded pairs are set to a learnable vector $\bar{c}^{\mathcal{G}}$. Self-attention still operates on all atom pairs $(i,j)$, even if they are not connected by a chemical bond.

Specifically, we ablate the following conditioning strategies:

Table A11: Median absolute error of ensemble properties between generated and reference conformers. Best results in **bold**, second best underlined; our models are marked with an asterisk "∗". Results for MCF, ET-Flow, and ours are averaged over three random seeds with standard deviation reported below. Other works do not report standard deviations.

| Method | $E$ [kcal/mol]↓ | $\mu$ [D]↓ | $\Delta\epsilon$ [kcal/mol]↓ | $E_{\mathbf{min}}$ [kcal/mol]↓ |
|---|---|---|---|---|
| GeoDiff (1.6M) | 0.31 | 0.35 | 0.89 | 0.39 |
| GeoMol (0.3M) | 0.42 | 0.34 | 0.59 | 0.40 |
| Torsional Diff. (1.6M) | 0.22 | 0.35 | 0.54 | 0.13 |
| MCF-L (242M) | 0.68 ±0.06 | 0.28 ±0.05 | 0.63 ±0.05 | 0.04 ±0.00 |
| ET-Flow (8.3M) | 0.18 ±0.01 | 0.18 ±0.01 | 0.35 ±0.06 | 0.02 ±0.00 |
| ∗DiTMC+aPE-B (9.5M) | 0.17 ±0.00 | 0.16 ±0.01 | 0.27 ±0.01 | **0.01** ±0.00 |
| ∗DiTMC+aPE-L (28.2M) | **0.16** ±0.02 | **0.14** ±0.03 | 0.27 ±0.01 | **0.01** ±0.00 |
| ∗DiTMC+rPE-B (9.6M) | **0.16** ±0.03 | 0.16 ±0.03 | 0.29 ±0.02 | 0.02 ±0.00 |
| ∗DiTMC+rPE-L (28.3M) | **0.16** ±0.01 | 0.15 ±0.02 | 0.28 ±0.06 | **0.01** ±0.00 |
| ∗DiTMC+PE(3)-B (8.6M) | 0.18 ±0.01 | 0.18 ±0.01 | 0.27 ±0.03 | 0.02 ±0.00 |
| ∗DiTMC+PE(3)-L (31.1M) | 0.17 ±0.01 | **0.14** ±0.01 | **0.25** ±0.01 | **0.01** ±0.00 |

- **node only** conditioning using only atom-wise graph conditioning tokens $c_i^{\mathcal{G}}$ (Eq. 6).
- **node & bond-pair** conditioning using both atom-wise graph conditioning tokens $c_i^{\mathcal{G}}$ (Eq. 6) and pair-wise graph conditioning tokens $c_{ij}^{\mathcal{G}}$ derived from edge-level representations of the conditioning GNN as discussed above (Eq. A54).
- **node & all-pair** conditioning using both atom-wise graph conditioning tokens $c_i^{\mathcal{G}}$ (Eq. 6) and pair-wise graph conditioning tokens $c_{ij}^{\mathcal{G}}$ based on geodesic graph distances (Eq. 7).

As reported in Tab. A14, all our proposed conditioning strategies significantly reduce the average minimum RMSD (AMR) for both recall (AMR-R) and precision (AMR-P) using DiTMC+aPE-B, compared to the unconditioned baseline.

Notably, the "node & all-pair" strategy achieves the best overall performance, with the lowest AMR values. These results highlight the strength of the all-pair conditioning strategy, which leverages graph geodesics to incorporate information from all atom pairs, rather than restricting conditioning to directly connected nodes or bonded pairs. This comprehensive approach captures more global structural information, thereby improving both precision and recall. See Appendix L for a more in-depth analysis.

### I.3 Index Positional Encoding (iPE)

Tab. A15 compares a variant including index positional encoding (iPE) from classic transformer architectures with DiTMC+aPE-B on GEOM-QM9. Specifically, we use the node index and encode it via sinusoidal encodings into the tokens $\mathcal{H}$ *before* the first DiTMC block, similar to embedding the absolute positions via aPE. Since the molecular graphs are generated from SMILES strings via

Table A12: Out-of-distribution generalization results on GEOM-XL for models trained on GEOM-DRUGS. -R indicates Recall, -P indicates Precision. Best results in **bold**, second best underlined; our models are marked with an asterisk "∗". Our results are averaged over three random seeds with standard deviation reported below. Other works do not report standard deviations.

| Method | 75 / 77 mols | | | | 102 mols | | | |
| --- | --- | --- | --- | --- | --- | --- | --- | --- |
| | AMR-R [Å]↓ | | AMR-P [Å]↓ | | AMR-R [Å]↓ | | AMR-P [Å]↓ | |
| | Mean | Median | Mean | Median | Mean | Median | Mean | Median |
| Tor. Diff. (1.6M) | 1.93 | 1.86 | 2.84 | 2.71 | 2.05 | 1.86 | 2.94 | 2.78 |
| MCF-S (13M) | 2.02 | 1.87 | 2.90 | 2.69 | 2.22 | 1.97 | 3.17 | 2.81 |
| MCF-B (64M) | 1.71 | 1.61 | 2.69 | 2.44 | 2.01 | 1.70 | 3.03 | 2.64 |
| MCF-L (242M) | 1.64 | 1.51 | 2.57 | 2.26 | 1.97 | 1.60 | 2.94 | 2.43 |
| ET-Flow (8.3M) | 2.00 | 1.80 | 2.96 | 2.63 | 2.31 | 1.93 | 3.31 | 2.84 |
| ∗DiTMC+aPE-B (9.5M) | 1.68 ±0.00 | 1.47 ±0.02 | 2.59 ±0.00 | 2.24 ±0.01 | 1.96 ±0.00 | 1.60 ±0.03 | 2.90 ±0.00 | 2.48 ±0.03 |
| ∗DiTMC+aPE-L (28.2M) | **1.56** ±0.01 | **1.28** ±0.01 | **2.47** ±0.00 | 2.14 ±0.01 | 1.88 ±0.01 | **1.51** ±0.02 | **2.81** ±0.00 | **2.30** ±0.02 |
| ∗DiTMC+rPE-B (9.6M) | 1.69 ±0.01 | 1.41 ±0.03 | 2.52 ±0.00 | **2.11** ±0.00 | 1.97 ±0.01 | 1.61 ±0.01 | 2.86 ±0.00 | 2.33 ±0.01 |
| ∗DiTMC+rPE-L (28.3M) | 1.66 ±0.03 | 1.37 ±0.01 | **2.47** ±0.00 | 2.18 ±0.02 | 1.96 ±0.02 | 1.61 ±0.02 | 2.82 ±0.00 | 2.42 ±0.02 |
| ∗DiTMC+PE(3)-B (8.6M) | 1.73 ±0.01 | 1.55 ±0.01 | 2.71 ±0.00 | 2.35 ±0.01 | 1.98 ±0.01 | 1.67 ±0.02 | 3.03 ±0.00 | 2.60 ±0.01 |
| ∗DiTMC+PE(3)-L (31.1M) | 1.57 ±0.01 | 1.46 ±0.01 | 2.60 ±0.00 | 2.27 ±0.02 | **1.85** ±0.02 | 1.58 ±0.03 | 2.93 ±0.00 | 2.53 ±0.03 |

Table A13: Ablation of PE strategies and Gaussian (G) and Harmonic (H) prior on GEOM-QM9. We report mean coverage (COV) at a threshold of 0.5Å, and mean average minimum RMSD (AMR) for Recall -R and Precision -P. Best results in **bold**. All results are averaged over three random seeds.

| Method | COV-R [%]↑ | | AMR-R [Å]↓ | | COV-P [%]↑ | | AMR-P [Å]↓ | |
| --- | --- | --- | --- | --- | --- | --- | --- | --- |
| | G | H | G | H | G | H | G | H |
| DiTMC+aPE-B | 96.2 | 96.1 | 0.074 | 0.073 | 95.2 | 95.4 | 0.087 | 0.085 |
| DiTMC+rPE-B | 96.0 | **96.3** | 0.073 | 0.070 | 95.2 | **95.7** | 0.084 | **0.080** |
| DiTMC+PE(3)-B | 95.7 | 95.7 | 0.069 | **0.068** | 93.5 | 93.4 | 0.090 | 0.089 |

RDKit and RDKit has to some extend a canonical ordering, this information can be used by the transformer architecture. However, index positional encoding breaks permutation equivariance (as we show in Tab. A15). This might be undesirable as permutation equivariance is one of the fundamental symmetries when learning on graphs. Since the ordering of atoms in a SMILES string is not uniquely defined, the trained network depends on the used framework for parsing the SMILES string or even a particular software version. We use RDKit (version 2024.9.5) for parsing SMILES strings to graphs.

Nevertheless, our DiTMC+aPE-B using iPE can effectively exploit the information contained in atom indices assigned by RDKit. A version of DiTMC+aPE-B without atom-pair conditioning but iPE achieves comparable performance to DiTMC+aPE-B using geodesic distances as atom-pair conditioning (pairwise conditioning). As our pairwise conditioning strategy is similarly or more effective than iPE but additionally preserves permutation equivariance, it should be preferred over iPE and we don't use iPE in any of our other experiments.

Table A14: Ablation of conditioning strategies using DiTMC+aPE-B on GEOM-QM9. -R indicates Recall, -P indicates Precision. Best results in **bold**. Our results are averaged over three random seeds with standard deviation reported below.

| Method | COV-R [%]↑ | | AMR-R [Å]↓ | | COV-P [%]↑ | | AMR-P [Å]↓ | |
|---|---|---|---|---|---|---|---|---|
| | Mean | Median | Mean | Median | Mean | Median | Mean | Median |
| DiTMC+aPE-B (no conditioning) | 68.6 ±1.0 | 91.7 ±2.1 | 0.405 ±0.005 | 0.325 ±0.004 | 36.8 ±0.5 | 36.4 ±2.2 | 0.729 ±0.006 | 0.703 ±0.007 |
| DiTMC+aPE-B (node only) | 96.3 ±0.0 | **100.0** ±0.0 | 0.079 ±0.000 | 0.037 ±0.000 | 93.2 ±0.2 | **100.0** ±0.0 | 0.112 ±0.001 | 0.051 ±0.000 |
| DiTMC+aPE-B (node & bond-pair) | **96.5** ±0.1 | **100.0** ±0.0 | 0.077 ±0.001 | 0.035 ±0.001 | 95.3 ±0.2 | **100.0** ±0.0 | 0.092 ±0.001 | 0.046 ±0.002 |
| DiTMC+aPE-B (node & all-pair) | 96.1 ±0.3 | **100.0** ±0.0 | **0.074** ±0.001 | **0.030** ±0.001 | 95.4 ±0.1 | **100.0** ±0.0 | **0.085** ±0.001 | **0.037** ±0.000 |

Table A15: Ablating index positional encoding (iPE) on GEOM-QM9 for different conditioning strategies (in brackets). To show the effect of atom permutations, we include results with randomly permuted atom indices (**perm.**). -R indicates Recall, -P indicates Precision. Best results in **bold**. Our results are averaged over three random seeds with standard deviation reported below.

| Method | COV-R [%]↑ | | AMR-R [Å]↓ | | COV-P [%]↑ | | AMR-P [Å]↓ | |
|---|---|---|---|---|---|---|---|---|
| | Mean | Median | Mean | Median | Mean | Median | Mean | Median |
| DiTMC+aPE-B (node only) | 96.3 ±0.0 | **100.0** ±0.0 | 0.079 ±0.000 | 0.037 ±0.000 | 93.2 ±0.2 | **100.0** ±0.0 | 0.112 ±0.001 | 0.051 ±0.000 |
| DiTMC+aPE-B (node only), **perm.** | 96.3 ±0.0 | **100.0** ±0.0 | 0.079 ±0.000 | 0.037 ±0.000 | 93.2 ±0.2 | **100.0** ±0.0 | 0.112 ±0.001 | 0.051 ±0.000 |
| DiTMC+aPE+iPE-B (node only) | **96.6** ±0.4 | **100.0** ±0.0 | 0.079 ±0.002 | 0.037 ±0.001 | **95.5** ±0.1 | **100.0** ±0.0 | 0.093 ±0.001 | 0.046 ±0.001 |
| DiTMC+aPE+iPE-B (node only), **perm.** | 82.3 ±1.1 | **100.0** ±0.0 | 0.229 ±0.008 | 0.108 ±0.005 | 60.0 ±0.9 | 61.8 ±1.4 | 0.493 ±0.008 | 0.416 ±0.011 |
| DiTMC+aPE-B (node & pairwise) | 96.1 ±0.3 | **100.0** ±0.0 | **0.074** ±0.001 | **0.030** ±0.001 | 95.4 ±0.1 | **100.0** ±0.0 | **0.085** ±0.001 | **0.037** ±0.000 |
| DiTMC+aPE-B (node & pairwise), **perm.** | 96.1 ±0.3 | **100.0** ±0.0 | **0.074** ±0.001 | **0.030** ±0.001 | 95.4 ±0.1 | **100.0** ±0.0 | **0.085** ±0.001 | **0.037** ±0.000 |

## J  Analysis of training loss as a function of latent time

In this section, we provide details for the analysis in Fig. 2C in the main part of the paper. We investigate the effect of the positional embeddings and self-attention formulations on the accuracy of the model. We therefore take pre-trained models on GEOM-QM9 and compute the training loss (as detailed in Algorithm 1) averaged over 1000 samples drawn randomly from the GEOM-QM9 validation set. We compute the loss for 30 logarithmically spaced values of $t_i = 1 - 10^{x_i}$, where $x_i \in [-1.8, 0]$ with uniform spacing. We skip the stochastic term in the loss as is done while sampling from the ODE.

As detailed in Fig. A5, we observe empirically that equivariance leads to a decreased loss close to the data distribution after training. This explains why our equivariant model more often succeeds to produce samples with increased fidelity, as depicted in figure Fig. 2B.

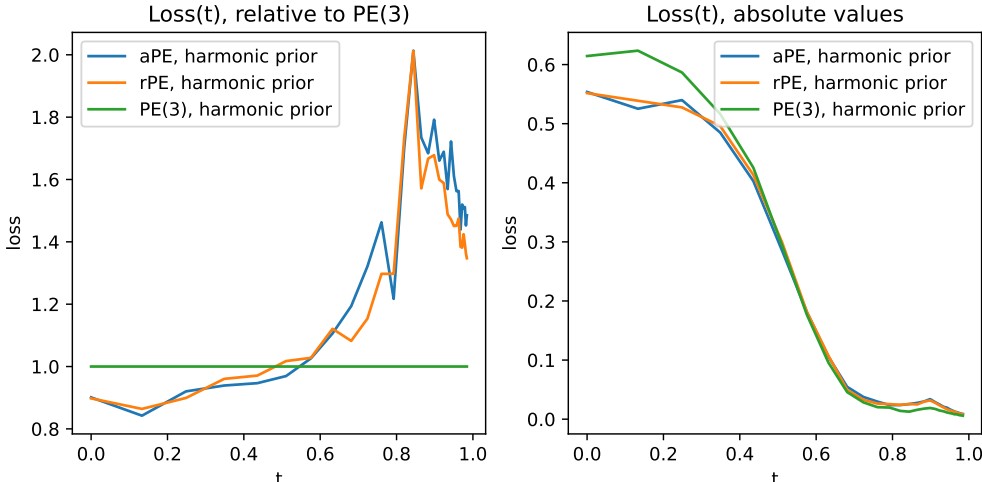

Figure A5: Loss as a function of time comparing different PE strategies. Results averaged over 1000 samples randomly drawn from the GEOM-QM9 validation set. **Left:** loss relative to PE(3) as a baseline. In the important regime close to the data distribution, the model PE(3) has lower loss, yielding higher sample fidelity. **Right:** absolute loss values for all PEs. The loss decreases close to the data distribution for all models.

We further note, that absolute loss values for models trained with all our PE strategies decrease as latent time increases (see Fig. A5). This is expected, as conditional vector fields for each data sample will start to interact more strongly moving away from the data distribution. Our weighted loss (see Appendix B) effectively penalizes errors close to the data distribution during training and helps with keeping the error low in this important regime.

## K    SMILES Classification Experiment via Conditioning GNN

The molecular graphs in our dataset are represented as SMILES strings. A critical requirement of DiTMC is the ability to distinguish distinct molecular graphs or SMILES representations through conditioning. In this section, we investigate whether our conditioning GNN is capable of learning the necessary information to distinguish between different SMILES strings for the generation of matching molecular conformers.

As a proxy evaluation task, we assess whether the conditioning network alone can function as a classifier of SMILES strings. To this end, we construct two training datasets: a toy dataset comprising three specific SMILES strings of a hydroxyl group moving along a carbon chain (`C(O)CCCCCCCC`, `CC(O)CCCCCCC`, `CCC(O)CCCCCC`), and a larger set consisting of 1000 randomly sampled SMILES strings drawn from the GEOM-QM9 validation set. Each SMILES string becomes a seperate class, so for each class there is exactly one example in the training data. The classification task is performed on the graph representations of the molecules, employing the same feature set and GNN architecture utilized in our conditioning GNN (see Appendix D.3 and Appendix G.2) followed by a simple classification head.

We train different models with a batch size of 3 for 5000 epochs on the curated toy dataset and batch size of 64 over 250 epochs on the GEOM-QM9 subset. We report classification accuracy on the training sets directly to evaluate the model's discriminative capacity. Furthermore, we explore whether conditioning weights obtained from an end-to-end trained model retain discriminatory power by freezing them and attaching a linear classification head.

Our results, as shown in Tab. A16, reveal that a simple linear classifier lacking message-passing capabilities fails to distinguish certain SMILES strings. Overall, our results indicate that a simple two-layer GNN effectively captures the necessary conditioning information through end-to-end training. Fig. A6 shows that without GNN layers, isomers will be misclassified.

Table A16: We measure the discriminative power of our conditioning graph network on a training set of 1000 randomly sampled SMILES strings from the GEOM-QM9 validation set, as well as a toy dataset of 3 different SMILES strings. We investigate the required number of message passing layers, as well as using pre-trained weights from an end-to-end trained model.

| GNN layers | Weight init | Trainable | Accuracy (GEOM-QM9) | Accuracy (Toy Data) |
|---|---|---|---|---|
| 0 | random | trainable | 0.887 | 0.333 |
| 1 | random | trainable | 0.999 | 0.666 |
| 2 | random | trainable | **1.000** | **1.000** |
| 2 | random | frozen | 0.980 | **1.000** |
| 2 | pre-trained | frozen | **1.000** | **1.000** |

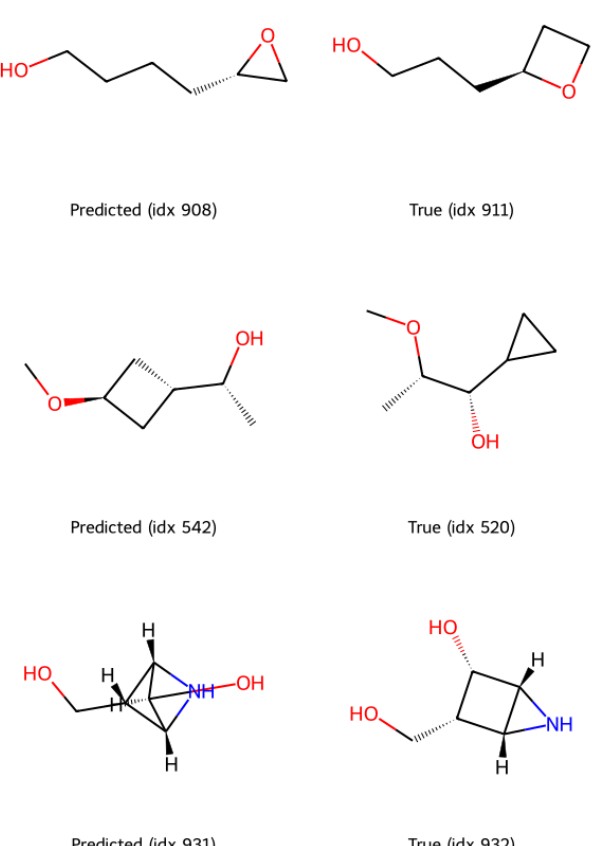

Figure A6: Misclassified exampled for SMILES classification experiment. We randomly pick 3 examples, which are misclassified by a classification head without any GNN layers. We show that GNN layers are essential for correct classification of isomers.

## L  Analysis of Sampling Trajectories

Fig. A7 compares the generative performance of DiTMC models trained on the GEOM-QM9 dataset under two different conditioning strategies: (1) node-only conditioning and (2) node plus pairwise conditioning, where we use the all-pair conditioning based on geodesic graph distances. Each row in the figure corresponds to one example molecule selected from the test set. The molecules are chosen to maximize the root mean squared deviation (RMSD) between the final generated structures of the

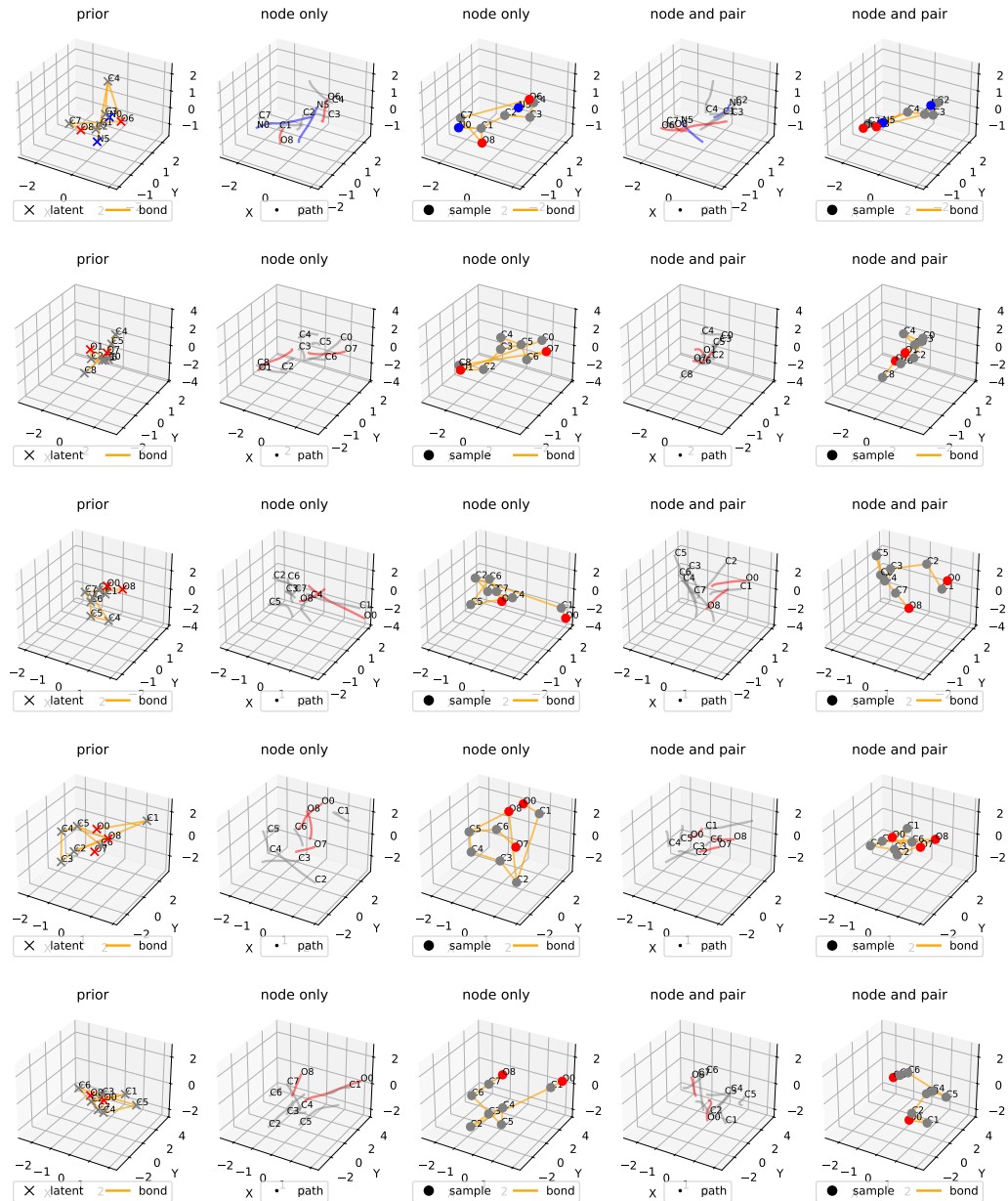

Figure A7: Comparison of molecular generation with node-only versus node- and pair-wise (all-pair) conditioning on GEOM-QM9. Each row shows a prior sample, sampling trajectories, and final generated structure for both models; pairwise conditioning preserves bonds from the conditioning graph $\mathcal{G}$.

two models. This selection highlights cases where the differences between the conditioning schemes are most pronounced.

Within each row, we display a sequence of images: the initial prior sample, followed by the intermediate trajectory of the ODE sampling process over 50 sampling steps for the node-only conditioned model, and the resulting final structure (with predicted bonds rendered in yellow). This sequence is repeated for the node- and pair-wise conditioned model, allowing a side-by-side visual comparison of the generation dynamics and final outputs.

The results reveal a consistent pattern: models trained with node-only conditioning fail to preserve bonding patterns from the conditioning graph $\mathcal{G}$. This manifests as bond stretching or atom permutation in the final structure. In contrast, the model, that is conditioned on pairwise geodesic distances, produces geometries that adhere more closely to expected chemical structure and the given bonds. We note that if atoms are simply permuted by the model using only node conditioning, the generated structure might still be valid in terms of the combination of generated 3D positions and atom types. The degraded performance of node conditioning versus node and pair-conditioning can therefore in part be explained by the used RMSD and Coverage metrics, which are not invariant to permutations of atoms.

Our findings still underscore the importance of incorporating both node-level and pairwise features in molecular generative models, in particular when agreement with the given conditioning on a bond graph is essential.

## M   Visualization

Fig. A14, Fig. A15, and Fig. A16 provide a visual comparison of conformers generated by MCF, ET-Flow, and DiTMC against the corresponding ground-truth reference conformers for the GEOM-QM9, GEOM-DRUGS and GEOM-XL datasets, respectively. For each dataset, we randomly select six reference conformers from the test split and generate conformers using each method. Finally, we apply rotation alignment of the generated conformers with their corresponding reference conformer.

## N   Code and Data Availability

The code and data to reproduce the main results of this paper, can be downloaded from here: `https://doi.org/10.5281/zenodo.15489212`, or via github: `https://github.com/ML4MolSim/dit_mc`.

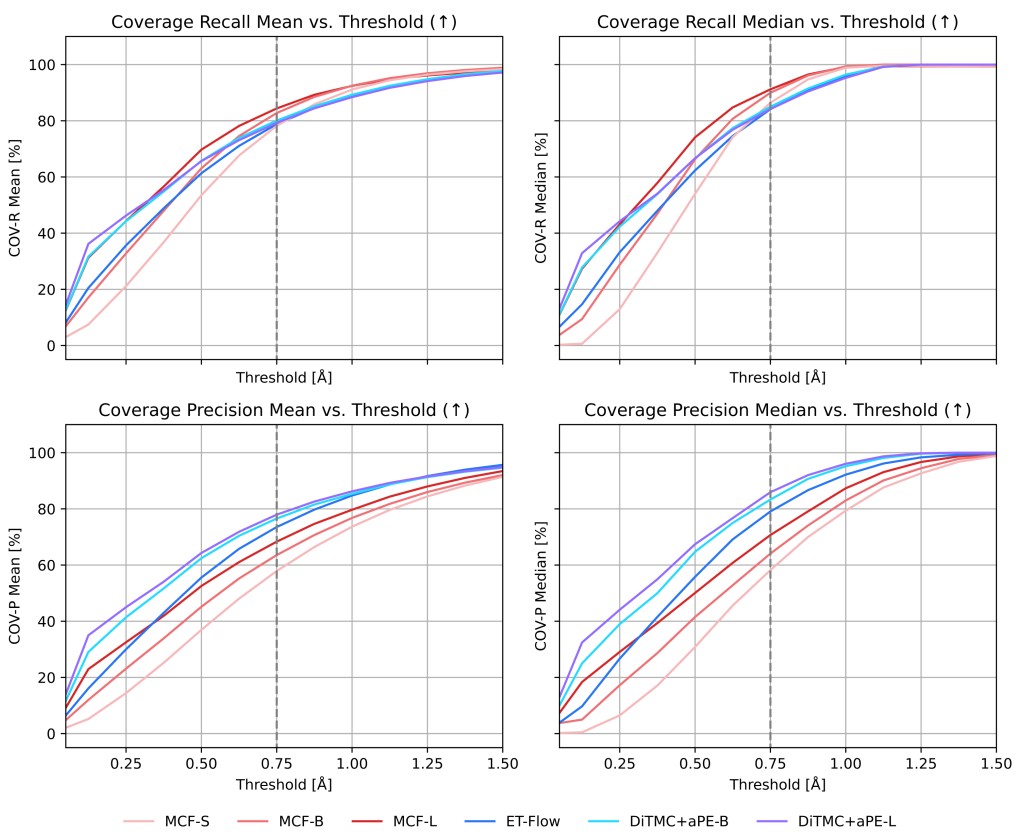

Figure A8: Coverage (COV) for precision ("-P") and recall ("-R") as a function of RMSD threshold $\delta$ for DiTMC+aPE and other state-of-the-art methods on GEOM-DRUGS. The vertical dashed line denotes the RMSD threshold $\delta = 0.75$ commonly used for evaluation on the GEOM-DRUGS dataset.

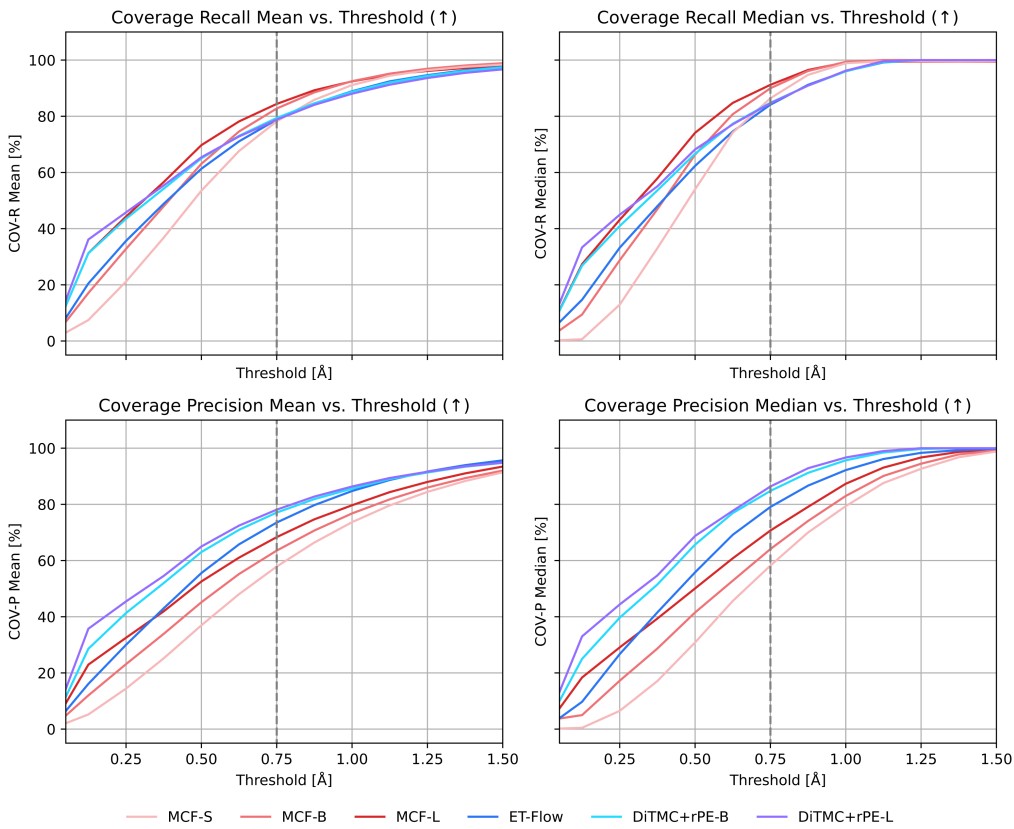

Figure A9: Coverage (COV) for precision ("-P") and recall ("-R") as a function of RMSD threshold $\delta$ for DiTMC+rPE and other state-of-the-art methods on GEOM-DRUGS. The vertical dashed line denotes the RMSD threshold $\delta = 0.75$ commonly used for evaluation on the GEOM-DRUGS dataset.

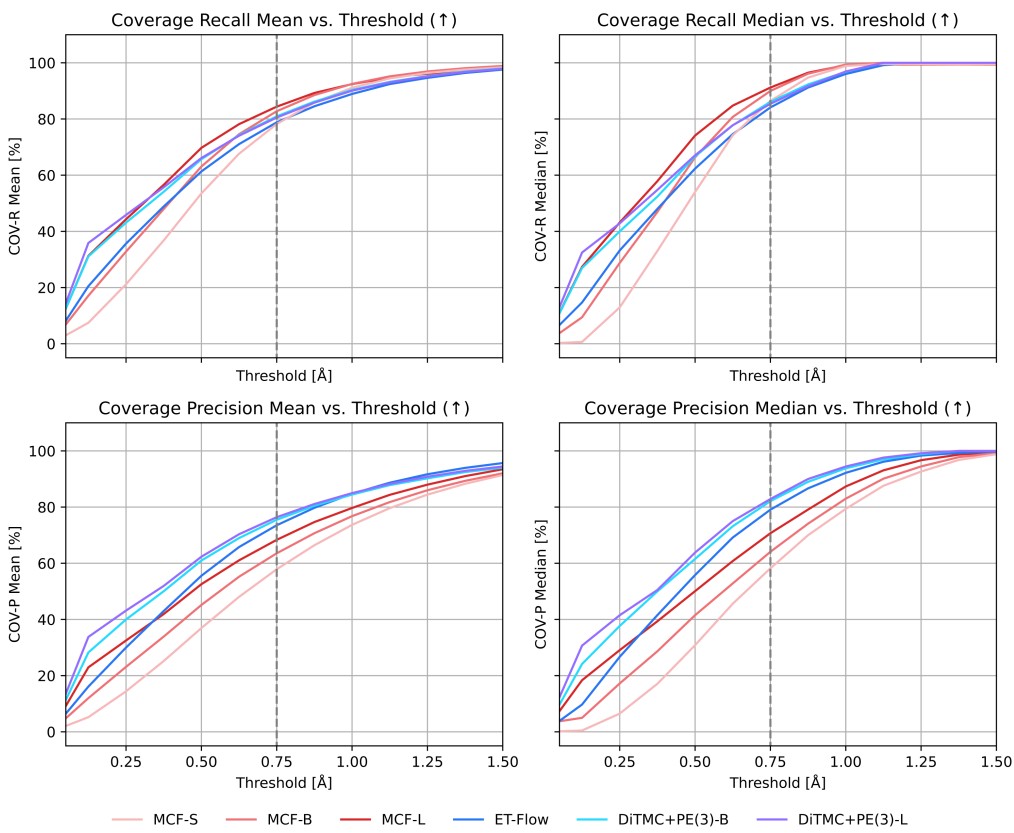

Figure A10: Coverage (COV) for precision ("-P") and recall ("-R") as a function of RMSD threshold $\delta$ for DiTMC+PE(3) and other state-of-the-art methods on GEOM-DRUGS. The vertical dashed line denotes the RMSD threshold $\delta = 0.75$ commonly used for evaluation on the GEOM-DRUGS dataset.

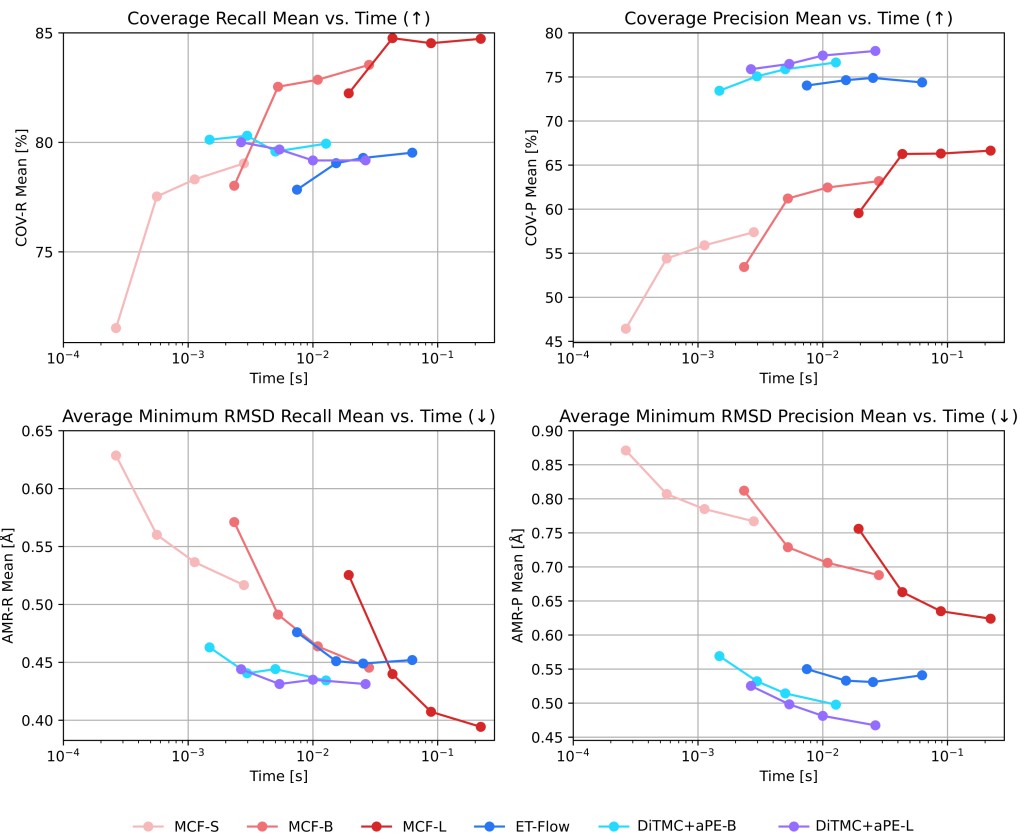

Figure A11: Average minimum RMSD (AMR) for precision ("-P") and recall ("-R") as a function of wall clock time per generated conformer. For each model, markers from left to right correspond to an increasing number of sampling steps during generation. Here, we follow Refs. [32, 47] and use 5, 10, 20, and 50 sampler steps.

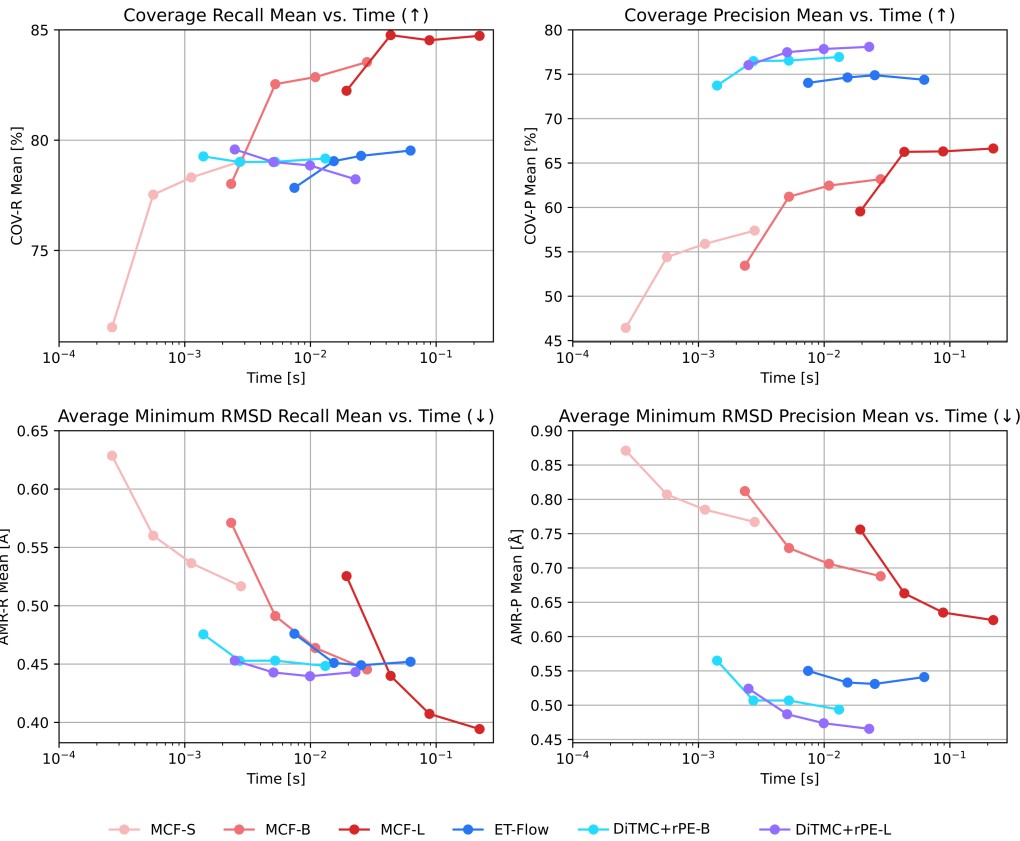

Figure A12: Average minimum RMSD (AMR) for precision ("-P") and recall ("-R") as a function of wall clock time per generated conformer. For each model, markers from left to right correspond to an increasing number of sampling steps during generation. Here, we follow Refs. [32, 47] and use 5, 10, 20, and 50 sampler steps.

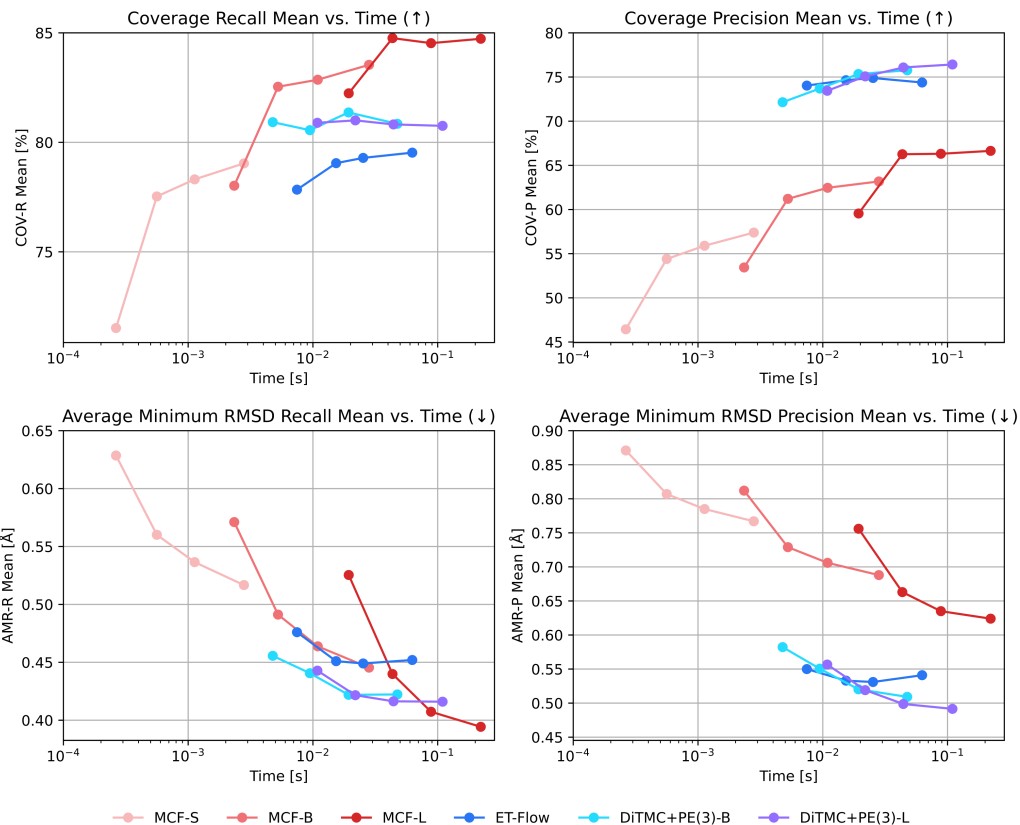

Figure A13: Average minimum RMSD (AMR) for precision ("-P") and recall ("-R") as a function of wall clock time per generated conformer. For each model, markers from left to right correspond to an increasing number of sampling steps during generation. Here, we follow Refs. [32, 47] and use 5, 10, 20, and 50 sampler steps.

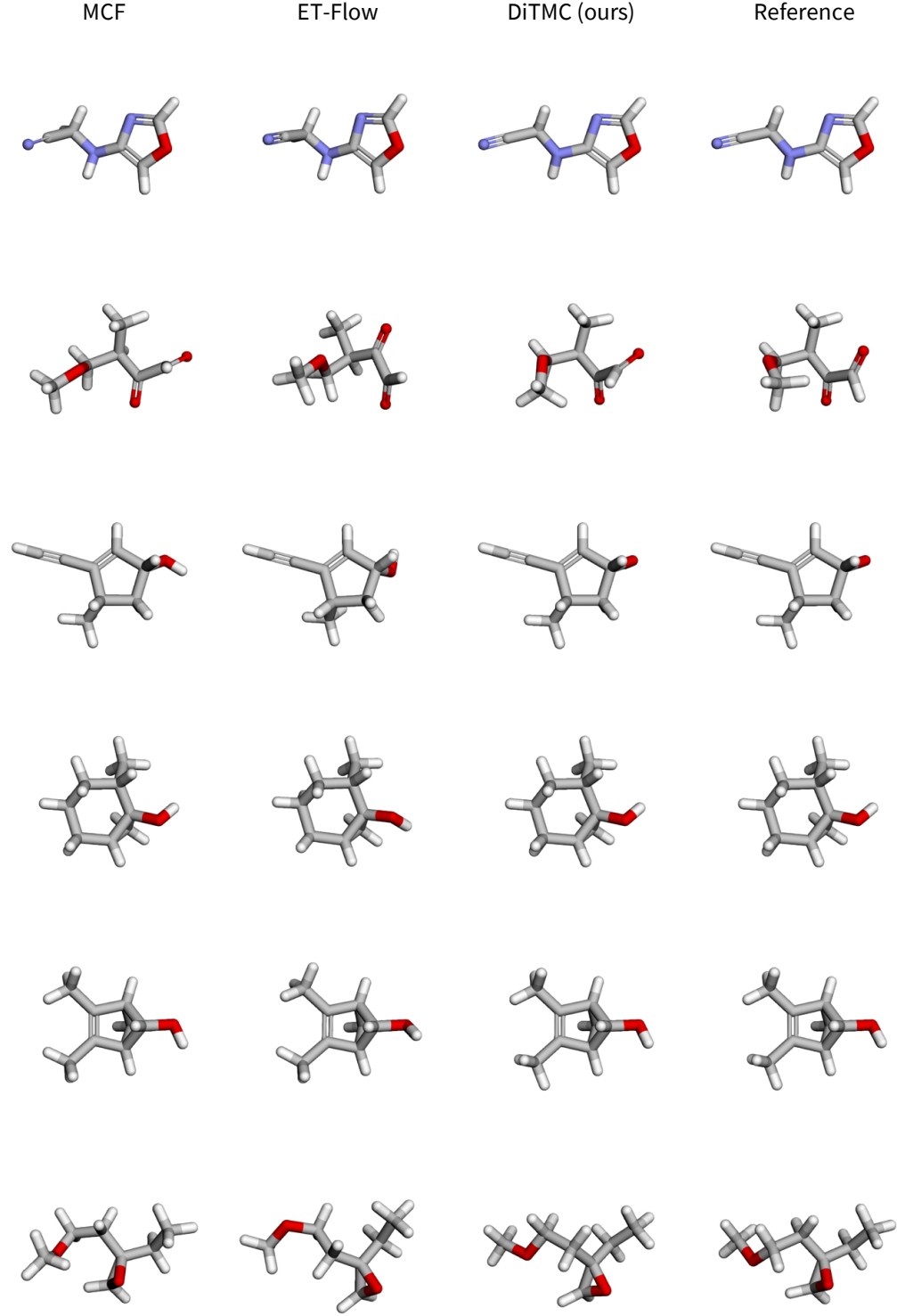

Figure A14: Comparison of conformers generated by MCF, ET-Flow, and DiTMC against ground-truth reference conformers from GEOM-QM9. The generated conformers are rotationally aligned with their corresponding reference conformer to facilitate comparison. **From left to right**: generated conformers from MCF, ET-Flow, DiTMC, and the corresponding ground-truth reference conformers.

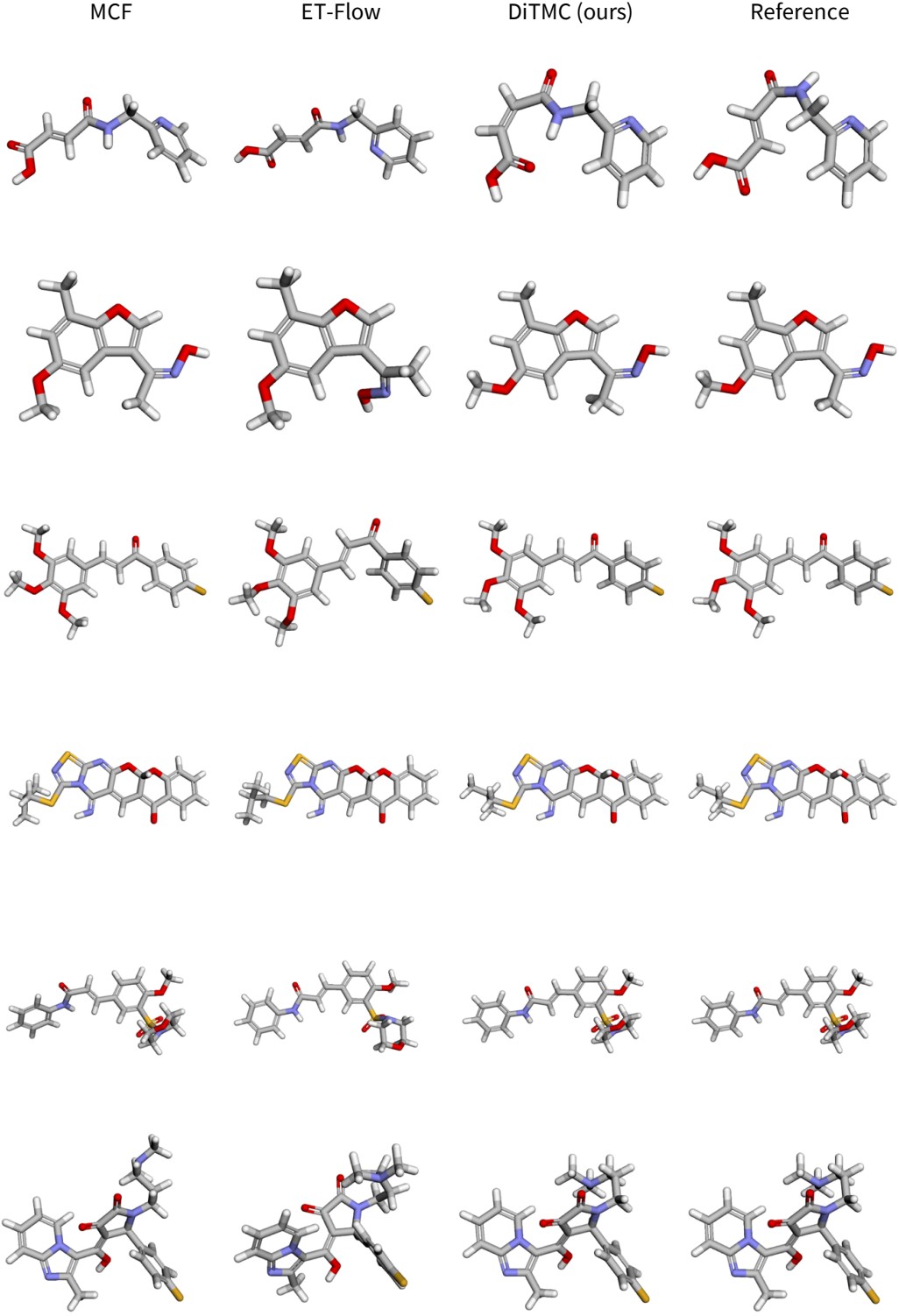

Figure A15: Comparison of conformers generated by MCF, ET-Flow, and DiTMC against ground-truth reference conformers from GEOM-DRUGS. The generated conformers are rotationally aligned with their corresponding reference conformer to facilitate comparison. **From left to right**: generated conformers from MCF, ET-Flow, DiTMC, and the corresponding ground-truth reference conformers.

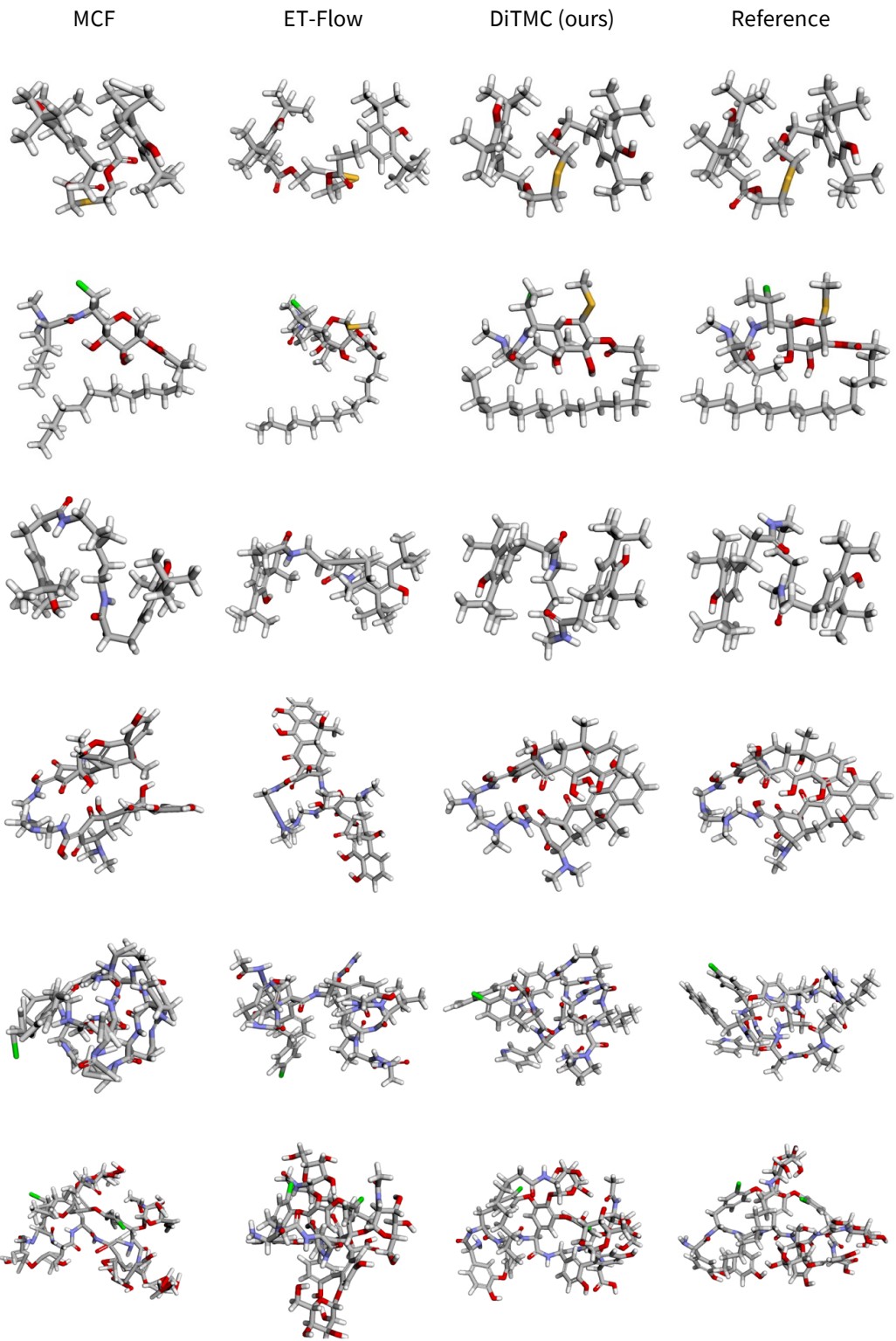

Figure A16: Comparison of conformers generated by MCF, ET-Flow, and DiTMC against ground-truth reference conformers from GEOM-XL. The generated conformers are rotationally aligned with their corresponding reference conformer to facilitate comparison. **From left to right**: generated conformers from MCF, ET-Flow, DiTMC, and the corresponding ground-truth reference conformers.

