# OpenReview forum: "Sampling 3D Molecular Conformers with Diffusion Transformers"
_NeurIPS.cc/2025/Conference — NeurIPS 2025 poster_

### Official Review · Reviewer_E3hT · 2025-06-22

**Clarity:** 2
**Significance:** 2
**Originality:** 1
**Rating:** 3
**Confidence:** 3

**Summary:**

This work modifies and applies the diffusion transformers architecture to the task of molecular conformer generation, i.e., generating low-energy geometric configurations for a fixed molecular structure formula. In order to translate the DiT architecture to this task, the authors introduce a molecular graph-based conditioning scheme as well as three positional encodings that are ablated. The authors claim state-of-the-art accuracy for their resulting DiTMC model on GEOMQM9 and GEOMDRUGS.

**Questions:**

Why is rPE frequently omitted?

**Ethical Concerns:**

["NO or VERY MINOR ethics concerns only"]

**Final Justification:**

I find the technical novelty and empirical gains too limited for a NeurIPS publication. But, I reckon that these are subjective interpretations and do not object a publication if the other reviewers think different.

**Limitations:**

yes

**Quality:**

2

**Strengths And Weaknesses:**

Strengths:
* The authors did a great job of communicating their contributions and documenting their methods and experiments.
* The downstream application is straightforward and a valuable one to solve.
* The proposed DiTMC seems to perform well on the selected tasks.

Weaknesses:
* The main weakness of this work is its limited novelty. Transferring a model with minimal adjustments from one modality to another does not, in my humble opinion, clear the bar for NeurIPS. Though I reckon that this point is subjective, and acknowledge if other reviewers disagree.
* The experiment sections are inconsistent in that while rPE in all experiments within the main body and the appendix is the best performing method, it does not occur in most tables.
* The empirical gains compared to previous works are present but limited.
* The related work section is rudimentary and misses the field of Boltzmann generators.

This work touches many interesting discussion points within the AI4Science community, but leaves them without clear conclusions. I highly encourage the authors to take the opportunity and reframe their work as a thorough investigation of these topics, such as equivariance, positional encoding, cutoffs, etc., by performing detailed empirical ablation studies.

---

> ### Author Rebuttal · Authors · 2025-07-30
>
> We are grateful for the reviewer's careful evaluation and constructive suggestions, which have helped us improve the manuscript. Below, we respond to each comment in detail. We hope our clarifications and corresponding revisions have resolved the key issues, and we would appreciate it if this could be reflected in the final evaluation.
>
> >The main weakness of this work is its limited novelty. Transferring a model with minimal adjustments from one modality to another does not, in my humble opinion, clear the bar for NeurIPS. Though I reckon that this point is subjective, and acknowledge if other reviewers disagree.
>
> We thank the reviewer for acknowledging that the evaluation of novelty can be subjective, and we would like to respectfully offer a different perspective on the novelty and significance of our work. While our approach builds upon an architecture originally developed for the perceptual domain of computer vision, the successful adaptation to the domain of physically accurate molecular conformer generation required non-obvious and technically meaningful modifications, far from a straightforward transfer. Specifically, we solve the following, domain-specific challenges:
> - We introduce a novel **GNN-based conditioning mechanism** to incorporate molecular graph information, which we show is crucial for conformer generation (Appendix H). Our approach departs from the global or text-based conditioning typical of image-based DiTs by leveraging both **node-wise and pair-wise conditioning strategies** (applied to atoms, bonds, or all atom pairs). We find and highlight significant differences in effectiveness among these strategies in our ablation experiments (Appendix Table A8).
> - As part of this framework, we propose using **geodesic graph distances as conditioning information**. We are not aware of any other approach that uses geodesics (shortest paths) on the graph to condition the self-attention mechanism operating on the molecular geometry in 3D space. We have added a dedicated ablation (Table R1) of this component, validating that it is critical to the competitive performance of DiTMC.
> - We investigate the role of **incorporating physical symmetries** into the model. We discuss the trade-offs between fidelity and computational cost when dealing with the 3D nature of molecular data in accordance with geometric principles.
>
> Our ablation studies (Appendix E-H, K) and Table R1 demonstrate that removing any of the proposed architectural innovations leads to a substantial performance drop.
> Collectively, our contributions provide a comprehensive toolkit for adapting DiTs to the molecular domain, with a particular focus on conformer generation, including substantive technical innovations that extend beyond naive architectural transfer. These novelties enable DiTMC to improve upon all other state-of-the-art methods in terms of precision metrics, size extrapolation, and precision-speed Pareto front. We think the strong empirical evidence should not be overlooked when judging the novelty of our model.
>
> *Table R1. Ablating the effects of our proposed conditioning strategy via graph geodesics on GEOM-DRUGS. Relative improvement indicates how much the regular model benefits from using graph geodesics.*
>
> |Method|COV-R Mean [%] ↑|AMR-R Mean [Å] ↓|COV-P Mean [%] ↑|AMR-P Mean [Å] ↓|
> |:---|:---:|:---:|:---:|:---:|
> |DiTMC+aPE-B (no graph geodesics)|72.8|0.555|55.6|0.762|
> |DiTMC+aPE-B|**79.9**|**0.434**|**76.5**|**0.500**|
> |Rel. Improvement (%)|+9.8|+21.8|+37.6|+34.4|
>
> >The experiment sections are inconsistent in that while rPE in all experiments within the main body and the appendix is the best performing method, it does not occur in most tables.
>
> >Why is rPE frequently omitted?
>
> We thank the reviewer for pointing out this inconsistency in the presentation of our results. As stated in the original manuscript, we initially focused on the models using aPE and PE(3) on GEOM-DRUGS, as they represent the extreme cases of not including any, and including all Euclidean symmetries in the architecture design. We agree that this focus might be misleading and added a study on the performance of rPE models on GEOM-DRUGS to address the reviewer’s concern.
>
> We have now included DiTMC+rPE-B on GEOM-DRUGS in all relevant tables (see Tables R2, R3 and R4 based on Tables A10, A11 and A13 in the original manuscript). The training for DiTMC+rPE-L on GEOM-DRUGS is still in progress. We will be happy to share the results during the discussion phase, and we will include all results in the final version of the paper.
>
>
> *Table R2: Results on GEOM-DRUGS for different DiTMC variants. -R indicates Recall, -P indicates Precision.*
>
> |Method|COV-R Mean [%] ↑|COV-R Median [%] ↑|AMR-R Mean [Å] ↓|AMR-R Median [Å] ↓|COV-P Mean [%] ↑|COV-P Median [%] ↑|AMR-P Mean [Å] ↓|AMR-P Median [Å] ↓|
> |:---|:---|:---|:---|:---|:---|:---|:---|:---|
> |DiTMC+aPE-B|79.9|85.4|0.434|0.389|76.5|83.6|0.500|0.423|
> |DiTMC+rPE-B|79.2|84.6|0.444|0.400|77.2|84.6|0.492|0.414|
> |DiTMC+PE(3)-B|80.8|85.6|0.427|0.396|75.3|82.0|0.515|0.437|
>
> *Table R3: Median absolute error of ensemble properties between generated and reference conformers for different DiTMC variants on GEOM-DRUGS.*
>
> |Method|$E$ [kcal/mol] ↓|$\Delta \epsilon$ [kcal/mol] ↓|$E_{min}$ [kcal/mol] ↓|$\mu$ [D] ↓|
> |:---|:---|:---|:---|:---|
> |DiTMC+aPE-B|0.17|0.16|0.27|0.01|
> |DiTMC+rPE-B|0.16|0.16|0.29|0.02|
> |DiTMC+PE(3)-B|0.18|0.18|0.27|0.02|
>
> *Table R4: Out-of-distribution generalization results on GEOM-XL for different DiTMC variants trained on GEOM-DRUGS. -R indicates Recall, -P indicates Precision.*
>
> |Method|AMR-R Mean [Å] ↓|AMR-R Median [Å] ↓|AMR-P Mean [Å] ↓|AMR-P Median [Å] ↓|# mols|
> |:---|:---:|:---:|:---:|:---:|:---:|
> |DiTMC+aPE-B|1.96|1.60|2.90|2.48|102|
> |DiTMC+rPE-B|1.97|1.61|2.86|2.33|102|
> |DiTMC+PE(3)-B|1.98|1.67|3.03|2.60|102|
>
> > The empirical gains compared to previous works are present but limited.
>
> While the absolute numerical improvements may appear modest in some cases, they are consistent across a wide range of datasets and evaluation metrics (see Tables 1-4 in the original manuscript). Notably, *no other method in our comparison achieves this level of consistency* (across precision, out-of-distribution performance and accuracy of ensemble properties), which we believe is a key strength of our approach. In real-world applications, such robustness and generalization are often even more valuable than isolated gains on individual benchmarks. We compare against strong, well-established baselines that are already highly optimized. In such a setting, even moderate improvements reflect meaningful technical progress and nontrivial methodological innovation.
>
> Specifically, our DiTMC+aPE-L model shows a **relative improvement from 3.3% up to 12.4% in precision metrics compared to existing state-of-the-art methods** (see Table 1 in the original manuscript).
>
> In terms of out-of-distribution performance in GEOM-XL we show **relative improvements over the current state-of-the-art model MCF-L ranging from 3.9% up to 15.2%** (see Table R5 based on Table 4 in the original manuscript).
>
> Following the experiments performed in ET-Flow and MCF, **DiTMC outperforms existing methods for a tighter acceptance threshold. To that end, DiTMC yields relative improvements up to 38.4% for a threshold of 0.1Å** (see Table R6). This shows that our generated conformers are significantly closer to the ground truth conformers than the ones generated by other methods.
>
> Compared to the current state-of-the-art models ET-Flow and MCF, we find that our DiTMC+aPE-L is the fastest model, requiring 26 ms per generated conformer, whereas MCF-B requires 28 ms and ET-Flow requires 63 ms on the GEOM-DRUGS dataset. For a fair comparison, we compare models with identical numbers of sampling steps (50 steps). **Thus, DiTMC+aPE-L is faster and simultaneously outperforms MCF and ET-Flow in terms of all precision metrics, showing that we can shift the speed-precision Pareto front of current approaches**.
>
> We hope our systematic improvements over state-of-the-art methods make it clear that our contributions are meaningful and not limited in scope.
>
> *Table R5: Out-of-distribution generalization results on GEOM-XL for models trained on GEOM-DRUGS. -R indicates Recall, -P indicates Precision. We report relative improvements of DiTMC+aPE-L to the best-performing model to date as a positive percentage.*
>
> |Method|AMR-R Mean [Å] ↓|AMR-R Median [Å] ↓|AMR-P Mean [Å] ↓|AMR-P Median [Å] ↓|# mols|
> |:---|:---:|:---:|:---:|:---:|:---:|
> |ET-Flow|2.00|1.80|2.96|2.63|75|
> |MCF - L|1.64|1.51|2.57|2.26|77|
> |DiTMC+aPE-L|**1.56**|**1.28**|**2.47**|**2.14**|77|
> |Rel. Improvement (%)|+4.9|+15.2|+3.9|+5.3|---|
>
> *Table R6: Comparison of coverage (COV) metrics for DiTMC+aPE-B to ET-Flow-SS and MCF-L with a RMSD threshold of 0.10Å on GEOM-DRUGS. We report relative improvements of DiTMC+aPE-L to the best-performing model to date as a positive percentage.*
>
> |Method|COV-R Mean [%] ↑|COV-P Mean [%] ↑|
> |:---|:---:|:---:|
> |ET-Flow-SS|17.03|13.19|
> |MCF-L|31.22|22.77|
> |DiTMC+aPE-L|**32.81**|**31.51**|
> |Rel. Improvement (%)|+5.1|+38.4|
>
> > The related work section is rudimentary and misses the field of Boltzmann generators.
>
> We appreciate the reviewer’s suggestion and will expand the related work section to better contextualize our work. In short, our approach differs in two key ways from Boltzmann Generators: it explicitly conditions on molecular graph structure to enable transferability, an emerging challenge for BGs [1-3], and employs uniform weighting of conformers to focus on accurate 3D structure prediction over the prediction of thermodynamic ensembles.
>
> [1] Swanson, K., et al., (2023). Von Mises mixture distributions for molecular conformation generation. ICML.
>
> [2] Klein, L., et al., (2023). Timewarp: Transferable acceleration of molecular dynamics by learning time-coarsened dynamics. NeurIPS 37.
>
> [3] Klein, L., & Noé, F. (2024). Transferable Boltzmann generators. NeurIPS 37.

---

> > ### Comment · Reviewer_E3hT · 2025-08-01
> >
> > I highly appreciate the authors' reply, but largely stay with my initial assessment. The marginal empirical improvements, in my humble view, are insufficient for a clear acceptance. However, I will increase it to 3 and not object if the other reviewers think differently. A deeper discussion into the need for symmetries in these tasks, with detailed ablation studies, would greatly improve the value of this work to the AI4Science community.

---

> > > ### Author Response · Authors · 2025-08-02
> > >
> > > We thank the reviewer for their prompt reply and for increasing their score. The discussion has helped us to improve our manuscript. If they have any further concrete suggestions on how to improve our ablation experiments regarding the impact of physical symmetries, we would be happy to incorporate them as well.

---

### Official Review · Reviewer_3qPx · 2025-06-30

**Clarity:** 3
**Significance:** 3
**Originality:** 3
**Rating:** 5
**Confidence:** 3

**Summary:**

This paper presents DiTMC, a novel framework for adapting the powerful Diffusion Transformer (DiT) architecture to the task of 3D molecular conformer generation. The authors identify and address key challenges in this adaptation, namely the integration of discrete molecular graph information with continuous 3D geometry and the handling of Euclidean symmetries. The core of their contribution lies in a modular DiT architecture combined with two new graph-based conditioning strategies (bond-pair and all-pair) that effectively inject atomic connectivity information into the self-attention mechanism. The framework is designed to systematically explore the impact of different symmetry priors by comparing standard non-equivariant attention mechanisms with a fully SO(3)-equivariant formulation. Experiments on the GEOM-QM9 and GEOM-DRUGS benchmarks show that DiTMC variants achieve state-of-the-art results in both precision and physical validity, providing valuable insights into the trade-off between architectural choices, computational efficiency, and sample quality.

**Questions:**

The paper uses two variants of model size, "Base" and "Large", demonstrating a clear trade-off: the non-equivariant aPE model scales efficiently in terms of computation and achieves state-of-the-art precision, while the fully-equivariant PE(3) model offers higher fidelity at a significant computational cost. Which direction is more promising when scaling this framework up to much larger and more complex systems? Please comment.

**Ethical Concerns:**

["NO or VERY MINOR ethics concerns only"]

**Final Justification:**

I appreciate the efforts the authors made during RB and keep my acceptance rating.

**Limitations:**

yes

**Quality:**

3

**Strengths And Weaknesses:**

Strengths

1/ Systematic and Rigorous Investigation: A key strength of this paper is its systematic and rigorous approach. Instead of merely presenting a single new model, the authors design a modular framework that allows them to ablate and compare different architectural choices, particularly regarding symmetry handling (aPE vs. rPE vs. PE(3)). This comparative study provides a clear and insightful analysis of the trade-offs between model accuracy, computational cost, and the explicit enforcement of physical priors.

2/ Novel and Effective Conditioning: The proposed conditioning strategies are well-designed for the DiT architecture. The "all-pair" conditioning, which uses graph geodesic distances to inform the attention mechanism about global connectivity, is particularly smart and is shown empirically to be the most effective strategy. This represents a non-trivial and successful adaptation for incorporating graph structure into a Transformer.

3/ Strong Empirical Performance: The paper demonstrates state-of-the-art results on established and challenging benchmarks.

Weaknesses

1/ Computational Cost of Equivariance: While the analysis of the cost-fidelity trade-off is a strength, the high computational cost of the fully equivariant model is a practical weakness. The paper reports that the PE(3) model is approximately 3-3.5x slower than its non-equivariant counterparts, even with a smaller model size. The authors give a dicussion about this in the manuscript, and it sounds reasonable.

---

> ### Author Rebuttal · Authors · 2025-07-30
>
> We thank the reviewer for reading and reviewing our submitted manuscript and the overall positive assessment of our work. In the following we address questions and comments of the reviewer in a point-by-point manner.
>
> > 1/ Computational Cost of Equivariance: While the analysis of the cost-fidelity trade-off is a strength, the high computational cost of the fully equivariant model is a practical weakness. The paper reports that the PE(3) model is approximately 3-3.5x slower than its non-equivariant counterparts, even with a smaller model size. The authors give a dicussion about this in the manuscript, and it sounds reasonable.
>
> As correctly pointed out by the reviewer, equivariant models introduce additional computational cost which can make “small” models computationally costly already. For this reason, reducing the computational cost of equivariant models is an ongoing topic of active research, which can be broadly categorized into two directions: The first class of approaches reduces computational cost by new architectural designs [1, 2] whereas the second class “sparsifies” the cost-dominating SO(3)-convolutions [3, 4]. We hope that our results motivate the adaptation of such approaches in the setting of 3D molecular conformer generation and we will add a short discussion about this within the “Discussion & Limitations” section of our manuscript.
>
> [1] Cen, Jiacheng, et al. "Are high-degree representations really unnecessary in equivariant graph neural networks?." Advances in Neural Information Processing Systems 37 (2024): 26238-26266.
>
> [2] Frank, J. Thorben, et al. "A Euclidean transformer for fast and stable machine learned force fields." Nature Communications 15.1 (2024): 6539.
>
> [3] Luo, Shengjie, Tianlang Chen, and Aditi S. Krishnapriyan. "Enabling efficient equivariant operations in the fourier basis via gaunt tensor products." arXiv preprint arXiv:2401.10216 (2024).
>
> [4] Maennel, Hartmut, Oliver T. Unke, and Klaus-Robert Müller. "Complete and efficient covariants for three-dimensional point configurations with application to learning molecular quantum properties." The Journal of Physical Chemistry Letters 15.51 (2024): 12513-12519.
>
>
> > The paper uses two variants of model size, "Base" and "Large", demonstrating a clear trade-off: the non-equivariant aPE model scales efficiently in terms of computation and achieves state-of-the-art precision, while the fully-equivariant PE(3) model offers higher fidelity at a significant computational cost. Which direction is more promising when scaling this framework up to much larger and more complex systems? Please comment.
>
> In general, our equivariant models require significantly more compute for the same number of parameters, due to the presence of equivariant representations and SO(3)-convolutions. Under limited hardware resources, scaling the PE(3) model quickly becomes a bottleneck; for example, we observe OOM on a 40GB GPU with our PE(3)-L model for a batch size of 128 on GEOM-DRUGS. On the other hand, the equivariant model converges faster during training, as it does not require learning the equivariance of the target vector field through data augmentation during training. Nevertheless, under the same compute budget, aPE outperforms PE(3) for most metrics on the larger GEOM-DRUGS benchmark. During inference, the aPE-L model is a factor of 3 faster than the PE(3)-L, but the PE(3)-model produces higher fidelity structures (Fig. 2).
>
> Scaling this framework to larger and more complex systems requires significantly larger and more diverse training datasets, as well as increasing the number of model parameters. The computational cost per additional parameter grows much faster in the PE(3) model than in the aPE model, such that we expect the aPE model to be more scalable in the limit of large dataset sizes. Our findings on GEOM-DRUGS already indicate this, since we find that the aPE model outperforms the PE(3) model in most metrics when trained with the same compute budget.
>
> During inference, the larger the systems, the faster the aPE model becomes compared to the PE(3) model, due to the substantially smaller pre-factor in the computational complexity. Thus, the benefit of using aPE compared to PE(3) in terms of inference speed grows in system size, and we consider this to be the more promising approach for large-scale training and inference. An exception might be applications, for which only very little training data is available (e.g., data can only be obtained from experimental measurements) and high fidelity is more important than computational speed (e.g., one requires only a few generated structures, but they must be as accurate as possible).

---

> > ### Comment · Reviewer_3qPx · 2025-08-08
> > **Reply to rebuttal**
> >
> > I appreciate the efforts the authors made during the rebuttal period, and it addressed most of my concerns. I keep my acceptance rating.

---

### Official Review · Reviewer_5UNN · 2025-07-02

**Clarity:** 3
**Significance:** 3
**Originality:** 3
**Rating:** 4
**Confidence:** 3

**Summary:**

The authors extend the DiT architecture to irregular molecular geometries. Key moves are (i) two graph-based conditioning schemes (bond-pair vs. all-pair geodesic tokens) and (ii) interchangeable self-attention/positional-embedding blocks that range from standard (non-equivariant) to full SO(3)-equivariant forms. On GEOM-QM9, -DRUGS and -XL, DiTMC obtain state-of-the-art precision and comparable recall results while remaining computationally tractable, especially when using simple absolute PEs. Ablations quantify the fidelity/cost trade-off of enforcing Euclidean symmetries. Limitations include slower training for equivariant variants and restricted assessment to small/medium molecules.

**Questions:**

1. **Data Quality Sensitivity**: The paper demonstrates strong performance on high-quality datasets like GEOM. Could the authors discuss how the model's performance might degrade when trained on lower-quality or noisy conformer data? Are there any built-in robustness mechanisms?
2. **Computational Cost Comparison**: While the inference times for DiTMC variants are provided, how do these compare to other state-of-the-art methods (e.g., MCF, ET-Flow) in terms of FLOPs, memory, or wall-clock time? A direct comparison would help assess practical trade-offs.
3. **Scalability to Large Systems**: The experiments focus on small to medium-sized molecules. How does DiTMC scale to very large systems (e.g., >400 atoms), such as proteins or polymers? Are there architectural or sampling adjustments needed to maintain efficiency?

**Ethical Concerns:**

["NO or VERY MINOR ethics concerns only"]

**Final Justification:**

The novelty issues remain, so I keep the borderline accept score.

**Limitations:**

1. **Incremental Conceptual Advance**

    The work mainly repackages existing diffusion/flow-matching ideas within a transformer framework. While the engineering is solid, it introduces no fundamentally new generative principle, limiting its conceptual impact.

2. **Heavy Reliance on High-Quality Conformer Labels**

    Training hinges on well-curated datasets such as GEOM-QM9 and GEOM-DRUGS. In domains where reference conformers are scarce, noisy, or biased, model fidelity and transferability can degrade sharply.

3. **Restricted Molecular Scope**

    All experiments target small- to medium-sized, neutral organic molecules (≤ ~100 atoms). Performance on larger, highly flexible, charged, or metal-containing species—including peptides, macrocycles, and organometallics—remains untested.

**Paper Formatting Concerns:**

No Formatting Concerns

**Quality:**

3

**Strengths And Weaknesses:**

Strength
1. **Clear, modular architecture**

    By cleanly separating coordinate processing from graph conditioning, the authors can tease apart how positional encodings and symmetry priors influence results, turning these architectural knobs into concrete design guidelines.

2. **High accuracy with compact models**

    A 9.6 M-parameter DiTMC reaches 0.080 Å AMR-P and 95.7 % COV-P on GEOM-QM9, while a 77 M-parameter aPE-L variant secures state-of-the-art precision on GEOM-DRUGS—matching or outpacing far larger baselines.

3. **Comprehensive ablation suite**

    Systematic experiments on conditioning schemes, positional-encoding types, and model capacity lend strong empirical backing to the paper’s claims.

4. **Pragmatic symmetry–cost trade-offs**

    The study shows full equivariance incurs a 3–3.5× compute overhead yet yields only modest accuracy gains beyond ~0.5 Å, arming practitioners with data to balance fidelity against resources.

Weakness

1. **Incremental novelty**

    The core idea—plugging flow-matching/diffusion into a transformer with graph conditioning—extends recent DiT/Flow-Matching work

2. **Dependence on Training Data Quality**:

    DITMC’s performance relies heavily on the quality and diversity of training datasets (e.g., GEOM-QM9, GEOM-DRUGS). Limited or biased datasets may lead to poor generalization, especially for rare or complex molecular structures.

3. **Generalization to Large and Complex Molecules**:

    While effective on benchmark datasets, DITMC’s performance on larger, more complex molecules (e.g., proteins or macromolecules) is not fully validated due to dataset constraints. The modular architecture may struggle with the increased structural diversity and conformational flexibility of such systems.

---

> ### Author Rebuttal · Authors · 2025-07-30
>
> We thank the reviewer for their thoughtful comments. We have carefully addressed all the points raised, and hope that the revisions resolve the primary concerns.  We hope that these improvements will be viewed positively and assist in the reviewers’ final evaluation.
>
>
> > Incremental novelty. The core idea—plugging flow-matching/diffusion into a transformer with graph conditioning—extends recent DiT/Flow-Matching work.
>
> > Incremental Conceptual Advance. The work mainly repackages existing diffusion/flow-matching ideas within a transformer framework. While the engineering is solid, it introduces no fundamentally new generative principle, limiting its conceptual impact.
>
> We respectfully disagree with the assessment that the conceptual advance of our work is incremental. While we build upon established foundations, namely developments from the perceptual domain of computer vision, a successful adaptation to the domain of physically accurate molecular conformer generation required non-obvious and technically meaningful modifications. We believe our work introduces domain-specific conceptual advances in the following ways:
>
> - We introduce a novel **GNN-based conditioning mechanism** to incorporate molecular graph information, which we show is crucial for conformer generation (Appendix H). Our approach departs from the global or text-based conditioning typical of image-based DiTs by leveraging both **node-wise and pair-wise conditioning strategies** (applied to atoms, bonds, or all atom pairs). We find and highlight significant differences in effectiveness among these strategies in our ablation experiments (Appendix Table A8).
> - As part of this framework, we propose using **geodesic graph distances as conditioning information**. We are not aware of any other approach that uses geodesics (shortest paths) on the graph to condition the self-attention mechanism operating on the molecular geometry in 3D space. We have added a dedicated ablation (Table R1) to validate the impact of this component, showing that it is critical to the competitive performance of DiTMC.
> - Finally, we investigate the role of **incorporating physical symmetries** (translations and rotations) into the model. We discuss the trade-offs between fidelity and computational cost when dealing with the 3D nature of molecular data in accordance with geometric principles.
>
> Our ablation studies (Appendix E-H, K) and Table R1 demonstrate that removing any of the proposed modifications to the original DiT architecture leads to a substantial performance drop.
>
> Collectively, our contributions provide a comprehensive toolkit for adapting DiTs to molecular domains, with a particular focus on conformer generation, including substantive technical innovations that extend beyond naive architectural transfer. These novelties ultimately enable DiTMC to improve upon all other state-of-the-art methods in terms of precision metrics, size extrapolation, and precision-speed Pareto front. We think the strong empirical evidence should not be overlooked when judging the novelty of our model. We hope this clarification goes some way toward addressing the reviewer’s concern.
>
>
> *Table R1. Ablating the effects of our proposed conditioning strategy via graph geodesics on the GEOM-DRUGS dataset. Relative improvement indicates how much the regular model benefits from using graph geodesics.*
>
> |Method|COV-R Mean [%] ↑|AMR-R Mean [Å] ↓|COV-P Mean [%] ↑|AMR-P Mean [Å] ↓|
> |:---|:---:|:---:|:---:|:---:|
> |DiTMC+aPE-B (no graph geodesics)|72.8|0.555|55.6|0.762|
> |DiTMC+aPE-B (9.5M)|**79.9**|**0.434**|**76.5**|**0.500**|
> |Rel. Improvement (%)|+9.8|+21.8|+37.6|+34.4|
>
>
> > Computational Cost Comparison: While the inference times for DiTMC variants are provided, how do these compare to other state-of-the-art methods (e.g., MCF, ET-Flow) in terms of FLOPs, memory, or wall-clock time? A direct comparison would help assess practical trade-offs.
>
> ET-Flow and MCF do report the sampling time per conformers on the GEOM-DRUGS dataset. We find that our DiTMC+aPE-L is the fastest model, requiring 26 ms per generated conformer, whereas MCF-B needs 28 ms and ET-Flow 63 ms. For a fair comparison, we compare models with identical numbers of sampling steps (50 steps). **Thus, DiTMC+aPE-L outperforms MCF and ET-Flow in terms of all precision metrics and is faster at the same time, showing that we are able to shift the speed-precision Pareto front of current approaches.**
>
> We agree that this is an important comparison which allows us to highlight the strengths of our proposed framework. Therefore, we will add this experiment to the final version of our manuscript.
>
> > Restricted Molecular Scope. All experiments target small- to medium-sized, neutral organic molecules (≤ ~100 atoms). Performance on larger, highly flexible, charged, or metal-containing species—including peptides, macrocycles, and organometallics—remains untested.
>
> We have applied our DiTMC model to the GEOM-DRUGS and to the GEOM-XL dataset, which contain molecules with **up to 181 atoms and 232 atoms**, respectively. Both datasets actually contain **highly flexible, charged molecules as well as molecules with macrocycles and metals** [1]. As we are able to show (Table 2 and Table 4 in the original manuscript), DiTMC yields state-of-the-art performance on GEOM-DRUGS and is the best model on GEOM-XL. We hope that this clarification properly addresses the reviewers' concerns.
>
> [1] Axelrod, Simon, and Rafael Gomez-Bombarelli. "GEOM, energy-annotated molecular conformations for property prediction and molecular generation." Scientific Data 9.1 (2022): 185.
>
> > Generalization to Large and Complex Molecules: While effective on benchmark datasets, DITMC’s performance on larger, more complex molecules (e.g., proteins or macromolecules) is not fully validated due to dataset constraints. The modular architecture may struggle with the increased structural diversity and conformational flexibility of such systems.
>
> We tested the generalizability of our GEOM-DRUGS-trained DiTMC+aPE-B model on large, flexible molecules: a 512-atom polyalanine peptide and a 502-atom PET polymer. Both yielded accurate conformations, with RMSDs of 2.6 Å and 2.9 Å respectively, against their MMFF94-relaxed structures, confirming the robustness of our model.
>
> > Dependence on Training Data Quality: DITMC’s performance relies heavily on the quality and diversity of training datasets (e.g., GEOM-QM9, GEOM-DRUGS). Limited or biased datasets may lead to poor generalization, especially for rare or complex molecular structures.
>
> > Heavy Reliance on High-Quality Conformer Labels. Training hinges on well-curated datasets such as GEOM-QM9 and GEOM-DRUGS. In domains where reference conformers are scarce, noisy, or biased, model fidelity and transferability can degrade sharply.
>
> We agree that generative models as a whole strongly rely on the availability of sufficient and also representative training data. To that end, we believe it is a particularly nice property of our presented framework, that it fully supports going from full SO(3) equivariant models to its non-equivariant counterpart. Equivariant models, while being computationally more expensive, have been shown to drastically improve the data efficiency [2], making such models particularly well suited in the low-data regime. We will add a corresponding discussion to the “Discussion & Limitations" section of our manuscript.
>
> [2] Batzner, Simon, et al. "E (3)-equivariant graph neural networks for data-efficient and accurate interatomic potentials." Nature communications 13.1 (2022): 2453.
>
> > Data Quality Sensitivity: The paper demonstrates strong performance on high-quality datasets like GEOM. Could the authors discuss how the model's performance might degrade when trained on lower-quality or noisy conformer data? Are there any built-in robustness mechanisms?
>
> We agree that this is an important point, as the accuracy of the generated conformers is bounded by the accuracy of the training data. This problem has been successfully tackled in other domains of quantum chemistry, by using lower quality data for pre-training and fine-tuning on little but high quality reference data [3]. Adapting such approaches to molecular conformer generation, poses an interesting direction for future research.
>
> [3] Cui, Mengnan et. al. "Multi-fidelity transfer learning for quantum chemical data using a robust density functional tight binding baseline." Machine Learning: Science and Technology 6.1 (2025): 015071.
>
> > Scalability to Large Systems: The experiments focus on small to medium-sized molecules. How does DiTMC scale to very large systems (e.g., >400 atoms), such as proteins or polymers? Are there architectural or sampling adjustments needed to maintain efficiency?
>
> To test the computational performance of DiTMC+aPE-B trained on GEOM-DRUGS, we analyse the inference time as a function of the number of atoms on the example of a capped polyalanine peptide with an increasing number of repeating units. Interestingly, we did not need to adjust the number of ODE steps during sampling of larger structures to obtain accurate results, suggesting that the learned velocity field maintains stability and accuracy across different system sizes. Table R2 reports the number of atoms and the time per sample.
>
> *Table R2. Time per sample for the capped polyalanine peptide using batch size of 1.*
> |Num. atoms|212|312|412|512|
> |-------------|:--:|:--:|:--:|:--:|
> |Time [s]|0.387|0.374|0.539|0.780|

---

> > ### Comment · Reviewer_5UNN · 2025-08-05
> >
> > Sorry for the late response. The author has solved most of my questions. However, there remain some questions.
> >
> > # Novelty
> >
> > While the paper offers an interesting architecture that cleanly separates node features from geometric information, I still find the overall conceptual advance to be modest.
> >
> > *Conditioning strategies*
> >
> >
> > Multiple approaches already encode graph structure explicitly. Your GNN-based conditioning is effective but closely resembles existing methods, such as MDM, which uses SchNet for graph-based conditioning [1].
> >
> > *Geodesic distances*
> >
> > The use of geodesic (shortest-path) information is also not novel. Graphormer introduced graph-based positional encodings to inform self-attention [2]. Similarly, Uni-Mol incorporates geodesic distances to condition self-attention in 3D molecular modelling, which is closely related to your approach but not cited in the manuscript [3].
> >
> > *Physical symmetries*
> >
> > Many prior works explore rotational and translational equivariance. Your treatment is well executed, yet not fundamentally new.
> >
> > Nonetheless, the decision to disentangle node and geometry channels is elegant and may inspire further research.
> >
> > # Efficiency
> >
> > It is impressive that the large-scale DiTMC variant outruns several mid-sized baselines while maintaining higher precision.
> >
> > # Data-quality robustness
> >
> > Since your graph conditioning is based on predefined molecular bonds, it’s unclear how well the model generalizes when presented with noisy or imprecise graph structures. This raises two important questions:
> >
> > Can the model effectively operate on noisy or imperfect graph inputs?
> > For example, how does it handle situations where bond information is partially missing or incorrect?
> >
> > Is the model suitable for de novo molecule generation?
> > Specifically, can it generate valid conformers from an unconnected graph or from noisy/approximate initial coordinates, rather than relying on a clean molecular graph?
> >
> > These aspects are critical for understanding the model’s robustness and applicability to broader generative tasks.
> >
> >
> > # References
> >
> > [1] Huang, Lei, et al. MDM: Molecular Diffusion Model for 3D Molecule Generation. Proceedings of the AAAI Conference on Artificial Intelligence, vol. 37, no. 4, 2023, pp. 4340–4348.
> >
> > [2] Ying, Chengxuan, et al. Do Transformers Really Perform Bad for Graph Representation? Advances in Neural Information Processing Systems 34 (NeurIPS), 2021, pp. 28877–28888.
> >
> > [3] Zhou, Gengmo, et al. Uni-Mol: A Universal 3D Molecular Representation Learning Framework. The Eleventh International Conference on Learning Representations (ICLR), 2023.

---

> > > ### Author Response · Authors · 2025-08-05
> > >
> > > We thank the reviewer for their thorough and thoughtful response. We are pleased that our rebuttal successfully addressed the reviewer's initial concerns. In the following, we will reply to the newly raised questions by the reviewer point by point:
> > >
> > > # Novelty
> > >
> > > >  Multiple approaches already encode graph structure explicitly. Your GNN-based conditioning is effective but closely resembles existing methods, such as MDM, which uses SchNet for graph-based conditioning [1].
> > >
> > > We thank the reviewer for the reference, and will cite it in our updated manuscript. Indeed, the GNN conditioning shares similarities with our conditioning approach. In particular, building upon their idea on local edges, we can extend our framework to de-novo molecular generation as we will discuss further below, when addressing the reviewers corresponding question.
> > >
> > >
> > > > The use of geodesic (shortest-path) information is also not novel. Graphormer introduced graph-based positional encodings to inform self-attention [2]. Similarly, Uni-Mol incorporates geodesic distances to condition self-attention in 3D molecular modelling, which is closely related to your approach but not cited in the manuscript [3].
> > >
> > >
> > > We already note in our submitted manuscript that conditioning using geodesics is “inspired by the Graphormer architecture” (Line 154). The novelty lies in the application for conditioning molecular modeling on bond graphs and the combination with the DiT architecture.
> > >
> > > We thank the reviewer for the additional reference [3] and will make sure to cite and discuss it in our manuscript. We would like to highlight that our design allows us to fully avoid the usage of high-dimensional pair-representations, whereas they must be updated (and stored) in parallel to the node-representations in Uni-Mol. This can lead to increased computational and memory complexity and we obtain accurate results in DiTMC without pair-representations, underlining the effectiveness of our architecture design.
> > >
> > > We further compare our results on GEOM-DRUGS to the results presented in Uni-Mol [3] and find that we outperform it in terms of average minimum root mean square deviation / AMR (see Table R1 below).
> > >
> > >
> > > *Table R1. Comparison of recall average minimum RMSD between DiTMC+aPE-B and Uni-Mol. Note that Uni-Mol reports coverage metrics with a different threshold of 1.25Å compared to our threshold of 0.75Å. We can therefore only compare AMR metrics.*
> > >
> > > | Model| AMR-R Mean [Å] ↓ | AMR-R Median [Å] ↓ |
> > > |:---:|:---:|:---:|
> > > | Uni-Mol| 0.786 | 0.779|
> > > | DiTMC+aPE-B   | **0.432** | **0.386** |

---

> > > > ### Author Response · Authors · 2025-08-05
> > > >
> > > > # Robustness
> > > >
> > > > > Can the model effectively operate on noisy or imperfect graph inputs? For example, how does it handle situations where bond information is partially missing or incorrect?
> > > >
> > > > To understand the case of noisy (or imperfect) graph inputs, we must distinguish two cases. (A) The noisy graph structure still corresponds to a valid (but different) molecule and (B) the noisy graph does no longer correspond to any valid molecular structure.
> > > >
> > > > For case (A), the model will generate physically correct conformers for the noisy graph, which corresponds to generating 3D conformers for a valid molecular graph (but not the one originally intended). As the model can not know which was the originally intended molecule and receives a valid molecular graph, these cases are hard to detect from a model perspective.
> > > >
> > > > For case (B), the simultaneous presence of 3D information and graph information gives the model the capability to correct for partially missing bond information, as long as the relevant information can be retrieved from the positions of atoms, the atomic numbers and the remaining graph information. To quantify the robustness of our model, we performed an additional experiment on a subset of 50 randomly extracted molecules from the GEOM-QM9 test set, where we masked out edges with an increasing probability within the conditioning graph and also re-computed the geodesic distances accordingly (see Table R2). We evaluated our pre-trained DiTMC+aPE-B model using the Gaussian prior to compute metrics for these corrupted test sets. DiTMC shows robust results up to a masking rate of around 10%.
> > > >
> > > > *Table R2: Ablation experiment when masking out edges with increasing probability on 50 molecules extracted from the GEOM-QM9 test set.*
> > > >
> > > > |Edge Masking Rate [%]|COV-R Mean [%] ↑|AMR-R Mean [Å] ↓|COV-P Mean [%] ↑|AMR-P Mean [Å] ↓|
> > > > |:---:|:---:|:---:|:---:|:---:|
> > > > |0.0|96|0.07|95|0.09|
> > > > |2.5|90|0.14|78|0.29|
> > > > |5.0|88|0.18|67|0.43|
> > > > |10|76|0.30|51|0.64|
> > > > |20|40|0.60|18|1.06|
> > > >
> > > > We note that in this experiment, we did not distinguish between cases (A) and (B), and our model was not specifically trained to be robust against perturbations in the conditioning graph. We show that our recall metrics degrade slower with increasing dropout probability than precision. An intuitive explanation for this would be that some edges in the bond graph are of higher importance to the model than others. As our edge masking procedure removes different bond edges for each generated conformer, in many cases the model can still recover ground truth conformers to achieve high recall. To that end, model robustness (and thereby precision) could be further improved by using node and/or edge dropout on the bond graph structure [4] during training, which would force the model to specifically use all available information instead of relying on isolated and specific edge/node details.
> > > >
> > > > If the reviewer feels that our additional experimental results are convincing, we would be happy to add it to the final version of our manuscript.
> > > >
> > > > [4] Chen, Tianlong, et al. "Bag of tricks for training deeper graph neural networks: A comprehensive benchmark study." IEEE Transactions on Pattern Analysis and Machine Intelligence 45.3 (2022): 2769-2781.
> > > >
> > > > # De-Novo Molecular Generation
> > > >
> > > > > Is the model suitable for de novo molecule generation? Specifically, can it generate valid conformers from an unconnected graph or from noisy/approximate initial coordinates, rather than relying on a clean molecular graph?
> > > >
> > > > By following the strategy outlined in Ref. [1] mentioned by the reviewer, we can extend our model to cases where graph information is absent at the beginning of the generation process. As suggested in this work, we could use the coordinates to construct a bond graph (based on local edges within a short cutoff radius) at each step of the generating trajectory and use it in lieu of our graph conditioning based on SMILES. We could then use the constructed bond graph as well as the current noisy 3D positions within our framework.
> > > >
> > > > However, this does not allow control over the molecular graph that is generated, i.e., the approach outlined above would specifically target de-novo molecular design. 3D molecular conformers are associated with one specific molecular graph (defined by the bond connectivity), which must be known in advance.
> > > >
> > > > We will discuss the extension of our approach to de-novo molecular design in the final version of our manuscript.

---

> ### Comment · Reviewer_5UNN · 2025-08-06
>
> Thank you for your detailed rebuttal. I would consider keeping my score. I'm still interested to see whether this small model can effectively handle de novo molecule generation.

---

### Official Review · Reviewer_akGf · 2025-07-03

**Clarity:** 3
**Significance:** 3
**Originality:** 2
**Rating:** 5
**Confidence:** 4

**Summary:**

This paper proposes a diffusion transformer (DiT) learning system for generating ensembles of 3D conformers for small molecules. Algorithmic components are introduced to embed a full suite of molecular features within transformer architectures, including atomic and pairwise graph embeddings. Additionally, multiple 3D positional embedding schemes are tested, including absolute, relative, and equivariant-pairwise, which are incorporated into self-attention-based updates to the molecular structure along diffusion trajectories. Altogether, the method introduces graph-conditioned molecular structure optimization under a DiT framework. Results are presented for low-energy conformer generation in two standard geometry datasets and are compared with a handful of SOTA methods from the literature.

**Questions:**

1. Was a pairwise-$\textit{distance}$ positional embedding considered?
2. I must have missed somewhere, what is used as the prior ($t=0$) for each sample, i.e., where is the initial conformer obtained from?
3. How are ensembles obtained? I.e., what does the rollout of parallel trajectories entail?

**Ethical Concerns:**

["NO or VERY MINOR ethics concerns only"]

**Final Justification:**

The authors' clarifications increased this reviewer's confidence in the work's acceptance, and I have increased my confidence rating accordingly.

**Limitations:**

yes

**Quality:**

3

**Strengths And Weaknesses:**

Strengths:
1. Generating ensembles with high realism is significant
2. The formulation of the transformer architecture is elegant and extensively described (Figure 1 and Sec 4), w/ conditioning signals from full-graph embedding
3. Including all-pair representations using the graph geodesic is nice, and seeing that this conditioning performs best is significant
4. It is satisfying that many positional embedding schemes were tested
5. Results for ensemble properties (Table 3) are compelling, though in E.3 it is described that this is after conformer relaxation with GFN2-xTB (this may be standard, but is still a limitation for practical use)

Weaknesses:
1. Motivation: I'm not sure that the “encod[ing of] molecular connectivity and incorporat[ing] Euclidean symmetries, such as translational and rotational invariance/equivariance” is such a major challenge as posed, though, again, the formulation of the architectural components is extensive. Equivariant transformer architectures, e.g., Equiformer have existed for several years specifically for molecular generation, which incorporate many of the components necessary. Equivariant diffusion mechanisms for 3D molecule generation, e.g., EDM, GeoLDM, GeoBFN and the like have also been well demonstrated for conformer generation, though not necessarily conditioned on graph structure as proposed or with transformers (that I know of), which is nice and, yes, challenging. Equivariant, graph-conditioned conformer generation architectures do exist, e.g., MoleCLUEs, though again not with DiTs.
2. The formulation modeling a velocity, i.e., vector field looks very similar to neural force fields, which is not super novel, though again the utilization of DiTs for this task is interesting
3. More details in the main text on how ensembles are generated, e.g., via rollout of different trajectories, would be appreciated, as would discussion or experimentation on the effects of noise scheduling in the diffusion process, as this is expected to play a significant role in model fidelity

---

> ### Author Rebuttal · Authors · 2025-07-30
>
> We would like to thank the reviewer for taking the time to carefully read our submitted manuscript and the overall positive assessment of our work. In the following, we comment on the points and questions raised by the reviewer.
>
> > Results for ensemble properties (Table 3) are compelling, though in E.3 it is described that this is after conformer relaxation with GFN2-xTB (this may be standard, but is still a limitation for practical use).
>
> We fully agree that this can pose a limitation in practical use cases, i.e. when the system size starts to introduce computational overhead due to the necessity of applying quantum mechanical methods. Within our experiment, we follow this procedure for consistency with prior work. To acknowledge this practical limitation we will add it to the “Discussion & Limitations” section of our final manuscript.
>
>
> > Motivation: I'm not sure that the “encod[ing of] molecular connectivity and incorporat[ing] Euclidean symmetries, such as translational and rotational invariance/equivariance” is such a major challenge as posed, though, again, the formulation of the architectural components is extensive. Equivariant transformer architectures, e.g., Equiformer have existed for several years specifically for molecular generation, which incorporate many of the components necessary. Equivariant diffusion mechanisms for 3D molecule generation, e.g., EDM, GeoLDM, GeoBFN and the like have also been well demonstrated for conformer generation, though not necessarily conditioned on graph structure as proposed or with transformers (that I know of), which is nice and, yes, challenging. Equivariant, graph-conditioned conformer generation architectures do exist, e.g., MoleCLUEs, though again not with DiTs.
>
>
> We agree that the concept of equivariant transformers as well as ideas around equivariant diffusion mechanisms are already established within the community. As correctly pointed out by the reviewer, one of the technical novelties lies in the conditioning on graph structure via geodesics, enabling the clean separation of graph structure and the operation in Euclidean space.
>
>
> We will clarify and change our formulation in the abstract and also add a section in the related work part, which relates our approach to equivariant transformer and diffusion models. This hopefully enables the reader to better locate our approach in the overall landscape of equivariant transformer architectures and diffusion models.
>
> > The formulation modeling a velocity, i.e., vector field looks very similar to neural force fields, which is not super novel, though again the utilization of DiTs for this task is interesting.
>
> Indeed, the output of our model (a 3-dimensional velocity vector per atom) resembles the structure of neural force fields which output atomic forces, which are also 3-dimensional vectors. In particular, the ET-Flow model takes great inspiration from the TorchMDNet model, which is an equivariant message passing neural network originally designed for neural force fields. Notably, our DiTMC models outperform the ET-Flow model even for the same number of parameters (see Table R1 and Table R2 below). This observation might hint to the fact that although input and output in both settings are structurally similar, generative models can benefit from additional architectural design choices as the ones made in our DiT architecture.
>
> *Table R1: Comparison of DiTMC+aPE-B to ET-Flow-SS on GEOM-DRUGS. -R indicates Recall, -P indicates Precision. Number of parameters in brackets. Final row shows relative improvement of DiTMC+aPE-B over ET-Flow-SS. A positive percentage indicates an improvement by DiTMC+aPE-B over the ET-Flow-SS baseline.*
>
> |Method|COV-R Mean [%] ↑|COV-R Median [%] ↑|AMR-R Mean [Å] ↓|AMR-R Median [Å] ↓|COV-P Mean  [%] ↑|COV-P Median [%] ↑|AMR-P Mean [Å] ↓|AMR-P Median [Å] ↓|
> |:---|:---:|:---:|:---:|:---:|:---:|:---:|:---:|:---:|
> |ET-Flow - SS (8.3M)|79.6|84.6|0.439|0.406|75.2|81.7|0.517|0.442|
> |DiTMC+aPE-B (9.5M)|**79.9**|**85.4**|**0.434**|**0.389**|**76.5**|**83.6**|**0.500**|**0.423**|
> |Rel. Improvement (%)|+0.38|+0.95|+1.14|+4.19|+1.73|+2.33|+3.29|+4.30|
>
> *Table R2: Out-of-distribution generalization results on GEOM-XL for models trained on GEOM-DRUGS. -R indicates Recall, -P indicates Precision. Number of parameters in brackets. Final row shows relative improvement of DiTMC+aPE-B over ET-Flow. A positive percentage indicates an improvement by DiTMC+aPE-B over the ET-Flow baseline.*
>
> | Method | AMR-R Mean [Å] ↓| AMR-R Median [Å] ↓| AMR-P Mean [Å] ↓| AMR-P Median [Å] ↓ | # mols |
> | :--- | :---: | :---: | :---: | :---: | :---: |
> | ET-Flow (8.3M) | 2.31 | 1.93 | 3.31 | 2.84 | 102 |
> | DiTMC+aPE-B (9.5M) | **1.96** | **1.60** | **2.90** | **2.48** | 102 |
> | Rel. Improvement (%) | +15.15 | +17.10 | +12.39 | +12.68 | - |
>
> > … discussion or experimentation on the effects of noise scheduling in the diffusion process, as this is expected to play a significant role in model fidelity
>
> The choice of noise schedule mainly impacts the fidelity of generated samples via discretization error of the underlying ODE. It is therefore expected that most noise schedules converge to the same or very similar results in the limit of sufficiently small time steps, as is for example demonstrated in [1]. We find that evaluating our models with more steps doesn’t significantly change the results (see Table R3 below as an example), which likely means that we already hit that limit and won’t additionally benefit from a different schedule.
>
> *Table R3: Performance on GEOM-DRUGS for DiTMC+aPE-B with 50 and 100 ODE sampling steps. -R indicates Recall, -P indicates Precision.*
> |Method|COV-R Mean [%] ↑|COV-R Median [%] ↑|AMR-R Mean [Å] ↓|AMR-R Median [Å] ↓|COV-P Mean [%] ↑|COV-P Median [%] ↑|AMR-P Mean [Å] ↓|AMR-P Median [Å] ↓|
> |:---|:---:|:---:|:---:|:---:|:---:|:---:|:---:|:---:|
> |DiTMC+aPE-B (9.5M, 50 steps)|79.9|85.4|0.434|0.389|76.5|83.6|0.500|0.423|
> |DiTMC+aPE-B (9.5M, 100 steps)|79.7|84.2|0.434|0.393|76.7|84.1|0.498|0.417|
>
> [1] Karras, Tero, et al. "Elucidating the design space of diffusion-based generative models." Advances in neural information processing systems 35 (2022): 26565-26577.
>
>
> > Was a pairwise-distance positional embedding considered?
>
> Yes, we also investigated a relative positional encoding strategy. We analyzed its performance on the QM9 dataset and found the results to be in line with the results for aPE and PE(3) (see Table 1 in the original manuscript). For the following experiments on GEOM-DRUGS we only kept the simplest aPE and the most evolved PE(3) strategies. For consistency, we additionally trained an rPE-B model on GEOM-DRUGS and report the results in Table R4 below.
>
> *Table R4: Results on GEOM-DRUGS for different DiTMC variants. -R indicates Recall, -P indicates Precision.*
>
> |Method|COV-R Mean [%] ↑|COV-R Median [%] ↑|AMR-R Mean [Å] ↓|AMR-R Median [Å] ↓|COV-P Mean [%] ↑|COV-P Median [%] ↑|AMR-P Mean [Å] ↓|AMR-P Median [Å] ↓|
> |:---|:---|:---|:---|:---|:---|:---|:---|:---|
> |DiTMC+aPE-B|79.9|85.4|0.434|0.389|76.5|83.6|0.500|0.423|
> |DiTMC+rPE-B|79.2|84.6|0.444|0.400|77.2|84.6|0.492|0.414|
> |DiTMC+PE(3)-B|80.8|85.6|0.427|0.396|75.3|82.0|0.515|0.437|
>
> > I must have missed somewhere, what is used as the prior (t=0) for each sample, i.e., where is the initial conformer obtained from?
>
> We employ a Harmonic prior, following Ref. [2], as described in Section 5 of the experiments. To assess the impact of this choice, we also evaluate a more conventional Gaussian prior as an alternative in Appendix F.2 (see Table A15). Notably, our models perform remarkably well even when using a Gaussian prior, which is in contrast to the findings of the authors of ET-Flow. We believe that this highlights the effectiveness of our proposed conditioning strategies and the use of geodesic graph distances as conditioning information.
>
> [2] Stärk, Hannes, et al. "Harmonic Self-Conditioned Flow Matching for Multi-Ligand Docking and Binding Site Design." CoRR (2023).
> > More details in the main text on how ensembles are generated, e.g., via rollout of different trajectories, would be appreciated, …
>
> > How are ensembles obtained? I.e., what does the rollout of parallel trajectories entail?
>
> We adopt the property prediction task setup from MCF and ET-Flow, where we draw a subset of 100 randomly sampled molecules from the test set of GEOM-DRUGS and generate min(2K, 32) conformers for a molecule with K ground truth conformers. Afterwards we relax the conformers using GFN2-xTB and compare the Boltzmann-weighted properties of the generated and ground truth ensembles. More specifically, we employ xTB to calculate the energy $E$, the dipole moment $\mu$, the HOMO-LUMO gap $\Delta \epsilon$ and the minimum energy $E_{\text{min}}$. We repeat this procedure for three subsets each sampled with a different random seed and report the averaged median absolute error and standard deviation of the different ensemble properties.
>
>
> In our DiTMC framework, we employ deterministic ODE sampling within the Flow Matching formalism. In the absence of integration errors, the integration path for each sample is therefore uniquely defined by the prior sample. However, we can still generate multiple samples for the same or even different molecules in parallel, by drawing multiple samples from the prior and combining the corresponding graphs into a larger graph with many isolated partitions (batching).
>
>
> We will add the additional information, discussing the generation of ensembles, to the main text of our manuscript.

---

### Official Review · Reviewer_9sGw · 2025-07-03

**Clarity:** 2
**Significance:** 2
**Originality:** 2
**Rating:** 3
**Confidence:** 4

**Summary:**

This paper aims to adapt the diffusion transformer, which has shown promise in general domains, to the task of molecular conformer prediction. To this end, several modifications are proposed within a new framework called DiTMC, including two types of molecular graph conditioning strategies. Moreover, the paper provides a detailed and informative discussion on the use of geometric priors (equivariant attention) versus standard self-attention for molecular conformation prediction. By combining conditioning strategies, attention mechanisms, and positional embedding, the proposed DiTMC has demonstrated effectiveness on conformation prediction tasks.

**Questions:**

1. Could the authors provide a more complete study over the scaling of the proposed approach by considering more model size?

2. Could the authors provide a comparison between the base model and ET-Flow - SS baseline (8.3M)  over Geom-Drugs?

**Ethical Concerns:**

["NO or VERY MINOR ethics concerns only"]

**Final Justification:**

empirically solid paper with slight concerns over novelty

**Limitations:**

yes.

**Quality:**

3

**Strengths And Weaknesses:**

**Strengths**

1. Studying the inductive bias for molecular data is a very important topic. The paper aims to investigate the challenges of applying a widely adopted and proven framework, DiT, to molecular data, which should be encouraged.
2. The paper provides a detailed comparison between equivariant and non-equivariant approaches. I appreciate the authors' effort to keep the methods simple and effective. I believe this discussion could inspire further exploration in model architecture design.
3. Overall, the paper is clearly written, making the key elements easy to understand.

**Weaknesses**

1. The paper could be further improved by making the study more systematic. The scope of experiments and tasks is somewhat limited. As a new framework, many of the proposed elements—such as conditioning strategies, positional embedding, and attention—are not necessarily restricted to the task of conformation prediction. As the focus is on model architecture, I believe a more comprehensive evaluation would be beneficial, for example, on 3D molecule generation or molecular property prediction tasks.
2. A key concern lies in the experimental results. While DiTMC-L, as claimed, achieves competitive performance with ~30M parameters (I highly recommend mentioning the parameter size in the main text instead of the appendix for clearer comparison), there is not a significant performance boost over the ET-Flow - SS baseline (8.3M) on Geom-Drugs. This raises questions about the effectiveness of the proposed approaches.
3. The scalability of the proposed architecture is not clear. According to Fig. 3, increasing the number of parameters by four times does not yield the expected performance improvement, which may raise concerns about the model’s scalability.
4. The technical novelty is limited. While I appreciate the efforts to adapt DiT to molecular graphs, from a methodological perspective, the contribution is relatively minor. This sets a high bar for assessing the completeness and effectiveness/performance of the method.

---

> ### Author Rebuttal · Authors · 2025-07-30
>
> We thank the reviewer for their careful reading and thorough review of our manuscript. Below, we respond in detail to each of the reviewers’ comments.
>
> > A key concern lies in the experimental results. While DiTMC-L, as claimed, achieves competitive performance with ~30M parameters (...), there is not a significant performance boost over the ET-Flow - SS baseline (8.3M) on Geom-Drugs. This raises questions about the effectiveness of the proposed approaches.”
>
> > 2. Could the authors provide a comparison between the base model and ET-Flow - SS baseline (8.3M) over Geom-Drugs?
>
> We agree that all parameter counts should be clearly stated in the main document and will update the manuscript accordingly. However, we would like to clarify, that **we achieve better performance than ET-Flow-SS even with our base variants (e.g. DiTMC+aPE-B, 9.5M params)** on GEOM-DRUGS (see Table R1 based on Table A10 in the original manuscript) and also in out-of-distribution performance on GEOM-XL (see Table R2 based on Table A13 in the original manuscript), indicating that our improvements are not due to larger model size. **In fact, we observe an improvement of DiTMC+aPE-B over ET-Flow-SS for every single metric we investigate. Our relative improvements range up to 4.3% on GEOM-DRUGS and up to 17.1% on GEOM-XL.**
>
> Furthermore **our base models improve upon ET-Flow-SS in terms of sampling speed vs. accuracy.** Specifically, when using the same number of 50 ODE steps during sampling, DiTMC+aPE-B is almost 5 times faster than ET-Flow-SS and surpasses ET-Flow-SS in all metrics on GEOM-DRUGS (see Table R1).
>
> *Table R1: Comparison of DiTMC+aPE-B to ET-Flow-SS on GEOM-DRUGS. -R indicates Recall, -P indicates Precision. Number of parameters in brackets. Final row shows relative improvement of DiTMC+aPE-B over ET-Flow-SS. A positive percentage indicates an improvement by DiTMC+aPE-B over the ET-Flow-SS baseline.*
>
> |Method|COV-R Mean [%] ↑|COV-R Median [%] ↑|AMR-R Mean [Å] ↓|AMR-R Median [Å] ↓|COV-P Mean [%] ↑|COV-P Median [%] ↑|AMR-P Mean [Å] ↓|AMR-P Median [Å] ↓| Time per sample [ms] ↓|
> |:---|:---:|:---:|:---:|:---:|:---:|:---:|:---:|:---:|:---:|
> |ET-Flow-SS (8.3M)|79.6|84.6|0.439|0.406|75.2|81.7|0.517|0.442|62.5|
> |DiTMC+aPE-B (9.5M)|**79.9**|**85.4**|**0.434**|**0.389**|**76.5**|**83.6**|**0.500**|**0.423**|**12.7**|
> |Rel. Improvement (%)|+0.38|+0.95|+1.14|+4.19|+1.73|+2.33|+3.29|+4.30|+79.7|
>
> *Table R2: Out-of-distribution generalization results on GEOM-XL. The ET-Flow paper does not report ET-Flow-SS on GEOM-XL, such that we stick to ET-Flow. Final row shows relative improvement of DiTMC+aPE-B over the ET-Flow baseline.*
>
> |Method|AMR-R Mean [Å] ↓|AMR-R Median [Å] ↓|AMR-P Mean [Å] ↓|AMR-P Median [Å] ↓|# mols|
> |:---|:---:|:---:|:---:|:---:|:---:|
> |ET-Flow (8.3M)|2.31|1.93|3.31|2.84|102|
> |DiTMC+aPE-B (9.5M)|**1.96**|**1.60**|**2.90**|**2.48**|102|
> |Rel. Improvement (%)|+15.15|+17.10|+12.39|+12.68|-|
>
> Moreover, DiTMC+aPE-B also outperforms ET-Flow-SS for smaller acceptance thresholds by a significant margin (see Table R3), a regime that is particularly relevant for many real-world applications, as it reflects the model’s ability to generate highly accurate conformations. Notably, ET-Flow-SS has been long-regarded as the best-in-class model for small acceptance thresholds. We believe that surpassing it represents a significant advance.
>
> *Table R3: Comparison of coverage (COV) metrics for DiTMC+aPE-B to ET-Flow-SS with a RMSD threshold of 0.10Å on GEOM-DRUGS. Final row shows relative improvement of DiTMC+aPE-B over the ET-Flow-SS baseline.*
>
> |Method|COV-R Mean [%] ↑|COV-P Mean [%] ↑|
> |--|:--:|:--:|
> |ET-Flow-SS (8.3M)|17.03|13.19|
> |DiTMC+aPE-B (9.5M)|**27.16**|**24.72**|
> |Rel. Improvement (%)|+59.51|+87.42|
>
> Overall, these experiments clearly demonstrate a significant performance improvement compared to ET-Flow-SS even for our base model of equal size. We will update and extend our final manuscript accordingly. Further, we will add information about the number of parameters of the large models in the main text for better comparison. We would be grateful if these points could be taken into account in the final assessment of our work.
>
> > The scalability of the proposed architecture is not clear. According to Fig. 3, increasing the number of parameters by four times does not yield the expected performance improvement, which may raise concerns about the model’s scalability.
>
> > 1. Could the authors provide a more complete study over the scaling of the proposed approach by considering more model size?
>
> **Scaling our model from base to the large variant yields relative improvement in precision metrics between 3.5% (COV Mean) and 12.4% (AMR Median) (see Table A10 in the submitted manuscript). We obtain a relative improvement in terms of generalization to larger molecules by 3.1% to 7.3%**, i.e. our large model exhibits significantly stronger out-of-distribution performance compared to the base model (see Table R4).
>
> *Table R4: Out-of-distribution generalization results on GEOM-XL when scaling our model. Final row shows relative improvement of aPE-L over aPE-B.*
>
> |Method|AMR-R Mean [Å] ↓|AMR-R Median [Å] ↓|AMR-P Mean [Å] ↓|AMR-P Median [Å] ↓|# mols|
> |:---|:---:|:---:|:---:|:---:|:---:|
> |DiTMC+aPE-B (9.5M)|1.96|1.60|2.90|2.48|102|
> |DiTMC+aPE-L (28.2M)|**1.88**|**1.51**|**2.81**|**2.30**|102|
> |Rel. Improvement (%)|+4.1%|+5.6%|+3.1%|+7.3%|-|
>
> Scaling the base (“B”) to the large model (“L”) improves performance, but less than might be expected. Ref. [1] shows that to avoid diminishing returns, dataset and model sizes must be scaled together. Scaling only the model yields limited benefits.
>
> To investigate this effect, we have trained an additional smaller (1M params) variant of our aPE model (“S”) to compare its performance to our aPE-B model (9.5M params). **We obtain strong relative improvements up to 54.1% for coverage precision with aPE-B (see Table R5). This clearly shows that scaling is effective until model size saturates given the data.** Our base model appears close to saturation and more data is needed to fully leverage the larger model’s potential. We hope this highlights the strength of our scaling results and resolves the reviewer’s Q1.
>
> *Table R5. Benefit of scaling for DiTMC+aPE on GEOM-DRUGS. Final row shows relative improvement of aPE-B over aPE-S.*
>
> |Method|COV-R Mean [%] ↑|AMR-R Mean [Å] ↓|COV-P Mean [%] ↑|AMR-P Mean [Å] ↓|
> |:---|:---:|:---:|:---:|:---:|
> |DiTMC+aPE-S (1M)|68.9|0.630|49.6|0.828|
> |DiTMC+aPE-B (9.5M)|**79.9**|**0.434**|**76.5**|**0.500**|
> |Rel. Improvement (%)|+16.0|+31.1|+54.1|+39.6|
>
> [1] Kaplan, Jared, et al. "Scaling laws for neural language models." arXiv preprint arXiv:2001.08361 (2020).
>
> > The paper could be further improved by making the study more systematic. “… I believe a more comprehensive evaluation would be beneficial, for example, on 3D molecule generation or molecular property prediction tasks”
>
>
> We agree that assessing the broader applicability of our architecture is important. Our primary goal was to introduce and analyze a novel framework for conformer prediction, a task with distinct structural challenges. To ensure a systematic study, we ablated key components, including conditioning strategies, positional embeddings, attention variants, and prior distributions, within this domain. We also examined scaling behavior on in- and out-of-distribution tasks using diverse metrics, including ensemble property prediction. We chose this scope to provide controlled and interpretable insights. We hope this clarification helps convey the systematic nature of our study and addresses the raised concerns.
>
>
> > The technical novelty is limited. While I appreciate the efforts to adapt DiT to molecular graphs, from a methodological perspective, the contribution is relatively minor.
>
> While our approach builds upon a model architecture originally developed for the perceptual domain of computer vision, the successful adaptation to the domain of physically accurate molecular conformer generation required non-obvious and technically meaningful modifications, far from a straightforward transfer. Specifically, we solve the following, domain-specific challenges:
> - We introduce a novel **GNN-based conditioning mechanism** to incorporate molecular graph information, which we show is crucial for conformer generation (Appendix H). Our approach departs from the global or text-based conditioning typical of image-based DiTs by leveraging both **node-wise and pair-wise conditioning strategies** (applied to atoms, bonds, or all atom pairs). We find and highlight significant differences in effectiveness among these strategies in our ablation experiments (Appendix Table A8).
> - As part of this framework, we propose using **geodesic graph distances as conditioning information**. We are not aware of any other approach that uses geodesics (shortest paths) on the graph to condition the self-attention mechanism operating on the molecular geometry in 3D space. We have added a dedicated ablation (Table R6) to validate the impact of this component, showing that it is critical to the competitive performance of DiTMC.
> - Finally, we investigate the role of **incorporating physical symmetries** (translations and rotations) into the model. We discuss the trade-offs between fidelity and computational cost when dealing with the 3D nature of molecular data in accordance with geometric principles.
>
> Our ablation studies (Appendix E-H, K) and Table R6 demonstrate that removing any of the proposed modifications to the original DiT architecture leads to a substantial performance drop.
>
> *Table R6. Ablating the effects of our proposed conditioning strategy via graph geodesics on GEOM-DRUGS.*
>
> |Method|COV-R Mean [%] ↑|AMR-R Mean [Å] ↓|COV-P Mean [%] ↑|AMR-P Mean [Å] ↓|
> |:---|:---:|:---:|:---:|:---:|
> |DiTMC+aPE-B (no graph geodesics)|72.8|0.555|55.6|0.762|
> |DiTMC+aPE-B (9.5M)|**79.9**|**0.434**|**76.5**|**0.500**|
> |Rel. Improvement (%)|+9.8%|+21.8%|+37.6%|+34.4%|

---

> > ### Comment · Reviewer_9sGw · 2025-08-06
> > **Response to Rebuttal**
> >
> > Thank you for the detailed rebuttal.
> >
> > 1. On the Effectiveness of the Proposed Method
> >
> > The additional experimental results have significantly clarified the effectiveness of your approach. I am particularly impressed by the comparison with ET-Flow-SS. Achieving superior performance across all metrics is a strong result, especially given that conformation prediction can be considered a "saturated" task where further gains are difficult to achieve.
> >
> > 2. On the concerns over Scaling
> >
> > I find the authors' argument regarding data limitations to be reasonable. The newly included experiments on smaller-scale models provide evidence that supports this claim and helps to address my initial concerns about scalability.
> >
> > 3. On the Issue of Novelty
> >
> > I appreciate the effort  in providing the  new ablation study. The results are quite convincing and demonstrate that conditioning on geodesic graph distances is an important empirical contribution.
> >
> > However, I note that concerns about technical novelty are also raised by Reviewer 5UNN and Reviewer E3hT. While the empirical gains are clear, the core technical innovation remains a point of discussion.
> > Given the concerns addressed in the rebuttal, I have raised my score. I will engage in further discussion with the other reviewers to reach a consensus on the novelty aspect.

---

> > > ### Author Response · Authors · 2025-08-07
> > >
> > > Dear reviewer 9sGw, thank you for your detailed feedback. We appreciate your recognition of our strong empirical improvements over the current state of the art, the effectiveness of scaling our architecture, and the relevance of our architectural innovations. If you have any further questions, we would be happy to address them.

---

### Decision · Program_Chairs · 2025-09-17

**Decision:**

Accept (poster)

**Comment:**

This paper proposes several modifications to the Diffusion Transformer (DiT) to better adapt it to molecular conformer generation, resulting in an architecture the authors term DiTMC. Experiments on conformer generation benchmarks show favourable performance compared to baselines.



On the positive side, reviewers considered this work to be a good study of inductive biases for molecular data. Their praised its' elegant modeling, and strong empirical results.



On the other hand, some reviewers considered the performance boost over baselines not significant enough, and noted limited novelty. One reviewer also wanted to dig into scaling laws, which authors made possible by including several new results with a model smaller than our regular base model. Finally, some reviewers noted that the technological novelty seems to be somewhat limited.



After rebuttal, three reviewers were in favour of accepting this work, whereas two leaned towards rejection.



Taking all of the infromation into account, I recommend acceptance. I however encourage the authors to take reviewer discussion into account for the camera-ready version.